# Beneficiation of High-Density Tantalum Ore in the REFLUX™ Concentrating Classifier Analysed Using Batch Fractionation Assay and Density Data

**Simon M. Iveson** [1,*] **, Nicolas Boonzaier** [2] **and Kevin P. Galvin** [1]

1  ARC Centre of Excellence for Enabling Eco-Efficient Beneficiation of Minerals, University of Newcastle, Callaghan, NSW 2308, Australia; kevin.galvin@newcastle.edu.au
2  FLSmidth, Product Line Manager—Global—REFLUX™ Classifiers, Pinkenba, QLD 4008, Australia; nicolas.boonzaier@flsmidth.com
*  Correspondence: simon.iveson@newcastle.edu.au; Tel.: +61-2-4033-9079

**Abstract:** A laboratory-scale REFLUX™ Concentrating Classifier was operated in continuous mode to beneficiate a sub 0.100 mm tantalum ore with a head grade of 0.56 wt.% Ta. The unit incorporated a lower section with a reduced diameter to accommodate a low yield. At a yield to underflow product of 4.0 wt.%, a product grade of 13.3 wt.% was achieved (23.7 upgrade) at a recovery of 88.3%. Samples of the feed, product and reject were then fractionated in a batch REFLUX™ Classifier unit using dense lithium heteropolytungstate (LST) solution into 11 fractions. Each of these fractions was then screened into seven size intervals and analysed by pycnometry and X-ray fluorescence (XRF). Most of the material was found to reside in four relatively narrow density bands. A new analysis based on the recovery of selected tracer elements showed that the partition curve had good closure at both ends and that the density cut point and $Ep$ both increased with decreasing particle size. For the +0.045 mm material, the density cut point was estimated to be around 3952 kg/m$^3$ with an $Ep$ of 317 kg/m$^3$, but it was expected that this new method could overestimate $Ep$. An alternative novel approach for estimating the partition performance was developed. This method estimated the cut point and $Ep$ values to be 3764 kg/m$^3$ and 107 kg/m$^3$, respectively. However, sensitivity analysis found that due to the near total absence of material in the density range from 3400 kg/m$^3$ to 4700 kg/m$^3$, the $Ep$ could likely lie anywhere in the range from 0 to 250 kg/m$^3$. The methodology proved useful in establishing these limitations in the analysis.

**Keywords:** beneficiation; gravity separation; density separation; REFLUX™ Classifier; REFLUX™ Concentrating Classifier; tantalum; partition curve; sink–float method





## 1. Introduction

Assessing the performance of density-based separation processes is an ongoing challenge in the minerals processing industry, particularly for high-density minerals, including many of the so-called critical minerals required for the transition to a green circular economy. The traditional sink–float method requires multiple baths of varying density liquids. However, the organic heavy liquids only cover densities up to 3.3 RD (diiodo methane) or with Clerici solution (aqueous solution of thallium malonate and formate) up to 5.0 RD. Unfortunately, many of these liquids have serious toxicity concerns. Solutions of lithium heteropolytungstates (LST) in water are more benign but, even at elevated temperatures, can only reach up to around 3.5 RD [1,2]. An alternative method for measuring partition performance is to use tracer particles of varying but known densities, but commercial tracer particles have a lower size limit of about 1 mm [3], and often, the particle size range of industrial interest is much finer. Mineral liberation analysis using 2D scanning electron microscopy and 3D X-ray measurement is also used to infer density distributions and,

hence, partition performance, but these methods also struggle with fine particles, and the densities are inferred values that require calibration rather than being directly measured [4].

The REFLUX™ Classifier (RC™) consists of a vertical fluidised bed section with a set of parallel inclined channels mounted above. Through the Boycott effect [5], the inclined channels create a large effective surface area for settling, thus permitting a much larger throughput than a traditional teetered or fluidised-bed separator [6,7]. Closely spaced channels lead to laminar flow with a high shear rate and, in turn, shear-induced lift, leading to a very powerful density-based separation [8]. The technology is being used increasingly for dense mineral beneficiation. The published literature covers iron ore [9–11], mineral sands [12], zircon [13], antimony oxide [14] and chromite ore [15].

The REFLUX™ Classifier has also been used as a laboratory tool to perform batch fractionation experiments to measure the density distribution of particles. Early work with fine 0.038–2.00 mm coal using water produced yield–ash curves that closely matched those obtained by the sink–float method [16]. Later work using glycerol solutions extended this range of close agreement up to coal particles as large as 16 mm [17]. Water-based batch fractionation has also been used to accurately measure the density distributions and partition curves for a chromite ore covering a maximum particle density of ~4.5 RD with a nominal top size of 0.300 mm [18]. More recent batch fractionation of a nominal −2.00 mm gold-bearing sulfide ore with maximum particle density ~3.8 RD showed strong performance using a dense LST solution (~2.4 RD) as the fluidising liquid. The greater density of the LST solution increases the buoyancy force on the particles and, thus, makes it easier for shear-induced lift to occur, which enhances the sensitivity of the separation to particle density [19].

This fractionation technique has also been applied to sets of feed, product and reject samples taken from a continuous separation system to determine the partition performance (partition curve) of the separator. Unlike the sink–float method, the batch RC™ fractionation method does not produce three sets of products with identical density intervals that can be directly compared to calculate the partition value for each density interval. Instead, an algorithm was developed which interpolates the density distributions, assumes a given form for the partition surface and then obtains the best-fit value for the partition curve parameters by minimising the sum of the square of the errors between the predicted density distributions and the interpolated raw fractionation data [20].

This paper is the first to focus on the application of a modified REFLUX™ Classifier, referred to as the REFLUX™ Concentrating Classifier (RCC™), developed to concentrate low-grade ores of high density. This device has an extended lower section of reduced diameter, ideal for targeting low yields. This first application focused on tantalum ore, with a head grade of ~0.5 wt.% $Ta_2O_5$ and 1.4 wt.% tin (Sn). Tantalum oxide ($Ta_2O_5$, Tantalite) and tin oxide ($SnO_2$, Cassiterite) have densities in the range 5.3–7.3 RD and 6.8–7.1 RD, respectively [21], significantly higher than the dense mineral beneficiation examples mentioned earlier. Therefore, it is of interest to assess the performance of the new concentrator. Feed, product and reject samples were collected from a laboratory-scale unit operated in continuous mode.

To determine the partition curve of this continuous system, the steady state feed, product and reject samples were batch fractionated in a second RC™ unit using LST solution, with each flow fraction screened into multiple size intervals. These were then assayed and had their density measured. For reasons that are explained in Section 3.1, the extremely low yield of the high-density product meant that the algorithm of Galvin et al. [20] was unable to extract a sensible partition curve from this fractionation data. This finding prompted entirely new ways to approach the assessment of the separation. The paper provides a comprehensive consideration of the multi-species data available for evaluating this separation. We anticipate simpler approaches will emerge from this work.

## 2. Materials and Methods

A tantalum ore sample was beneficiated in a continuous laboratory REFLUX™ Concentrating Classifier (RCC™) (FLSmidth, Gauteng, South Africa), and feed, product and reject samples were collected. The +0.045 mm material from each of these samples was then fractionated into 11 flow fractions using a batch RC™ with LST solution. Each of these flow fractions was then screened into 7 size intervals (see Figure 1). Each of the resulting flow × size portions was then assayed by XRF (X-ray fluorescence), and their density was measured using gas pycnometry. From the distribution of elements in these data, the overall partition curve was estimated. The experimental methods and the data analysis are explained in this section, and the results are discussed in Section 3. The experiments, XRF assays and pycnometry were performed under instruction by Nagrom (Perth, Australia), the XRD analysis was performed by Microanalysis Australia (Perth, Australia), and the authors then analysed the data.

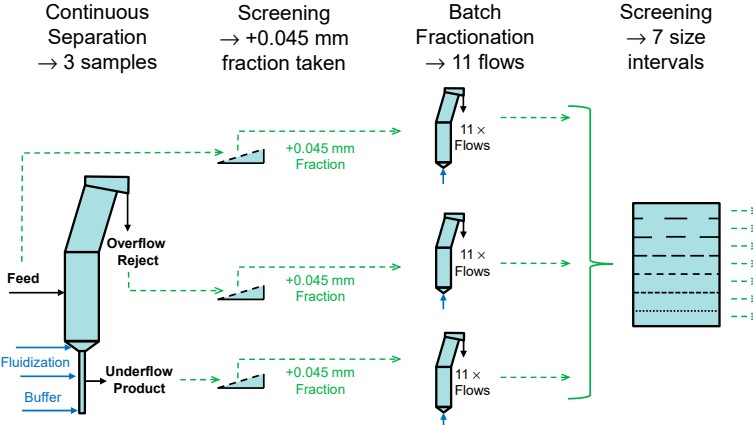

**Figure 1.** Schematic summary of the processing steps. The original continuous experiment furnished three samples—feed, product and reject. The +0.045 mm material from each of these samples was then batch fractioned into 11 flow fractions. Each of these fractions was then screened into 7 size intervals, thus resulting in a total of 3 × 11 × 7 = 231 portions.

### 2.1. The Tantalum Ore Feed

The feed sample was sourced from the tailings of a gravity circuit used to process a Western Australian tantalum deposit. The feed had a head grade of 0.56% $Ta_2O_5$. The nominal top size was 0.100 mm. The cumulative mass size distribution is shown in Figure 2. Further details on the feed composition are presented and discussed in Section 3.

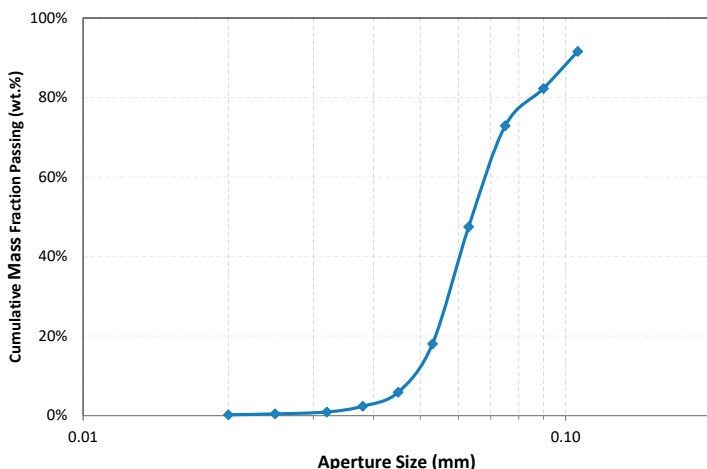

**Figure 2.** Cumulative mass fraction less than size distribution of the tantalum ore feed.

### 2.2. Continuous Separation Experiment—Source of Primary Separation Samples

The tantalum feed ore was processed under continuous steady-state conditions using an RCC™100-40 system (Figure 3). This system has a 340 mm tall main vertical RC™100 chamber with cross-section of 100 mm × 100 mm. This interfaced above with a ~1000 mm long inclined lamella section containing 23 plates that divided it into 24 channels with 3 mm channel spacing. Below, the unit connected with a 40 mm diameter pipe section that was 1039 mm tall (this is the RC™40 "concentrator" section installed for low yield separations). The underflow exited via a valve at the base of the RC™40 section into a short length of 25 mm NB clear hose beneath. A "buffer" upwards flow was added to keep the underflow stream fluidised whilst the actual underflow stream exited to the side. The underflow valve was automatically controlled to maintain a specified set point average bed density in the RC™ 40 section, as measured by the pressure difference between two pressure transducers located 130 mm and 910 mm, respectively, above the RC™40 base plate.

A 40 wt.% slurry of the feed solids was prepared in a 1000 L tank. During the approximate 3 h run time, wet filter cake and water in the correct ratio were periodically added to this tank to maintain a homogenous feed. The feed slurry was pumped into the unit at 5.30 L/min. This is equivalent to a solid feed flux of 17 t/(m$^2$ h) based on the 0.1 m × 0.1 m cross-sectional footprint area of the main vertical section. A flow of 0.12 L/min of primary fluidisation water was introduced at the base of the RC™40 section, and an additional 0.467 L/min of secondary fluidisation water entered around the base of the RC™100 section. The underflow buffer, which does not enter the separator, was set at 0.90 L/min. The density control set point in the RC™40 section was set at 2050 kg/m$^3$. The average exit underflow rate at these conditions was about 0.9 L/min, meaning that the average overflow rate was about 5.9 L/min.

Full-stream samples of the feed, underflow concentrate product and overflow tailing reject were collected at those conditions. The head assays of these three samples were measured using XRF. The mass concentrations of 20 species are reported in Table B1 (Appendix B). The species are reported as either pure elements or their oxides, and there is also "LOI" (=loss on ignition at 1000 °C). Note that when XRF reports elements as oxides, this does not imply that those mineral species are actually present in the sample (for instance, as discussed in Section 3.2, much of the iron was in the form FeS$_2$ rather than Fe$_2$O$_3$). This is why the raw compositions may often not sum to 100%. For instance, in samples with large amounts of Sn, the summation is often below 90%, likely due to the unaccounted-for presence of additional O associated with the Sn as cassiterite, SnO$_2$. It is further noted that data on Cl, As and Sb were also included in the original XRF reports, but they were present at such trace levels as to have no analytical value and so are not reported here. Later, head samples of the product and reject also had XRD (X-ray diffraction) analysis to check their actual mineralogy, as shown in Appendix C.

The overall mass yield to product ($Y$) and the recovery ($R_i$) of each component to product can be calculated from these raw assays using the two-product formula [22]:

$$Y = \frac{m_C}{m_F} = \frac{x_{i,F} - x_{i,T}}{x_{i,C} - x_{i,T}} \tag{1}$$

$$R_i = \frac{x_{i,C} m_C}{x_{i,F} m_F} = \frac{x_{i,C}}{x_{i,F}} \left( \frac{x_{i,F} - x_{i,T}}{x_{i,C} - x_{i,T}} \right) \tag{2}$$

where $m_j$ and $x_{i,j}$ are, respectively, the mass rate of stream $j$ and the mass fraction of species $i$ in stream $j$, where $j$ = F, C and T indicate the feed, underflow concentrate product and overflow tailings, respectively. These equations assume steady-state conditions, with no leaks or other losses, and when applied to data for size or density intervals, also assume no attrition, agglomeration or other phenomena that might cause material to move between different intervals. These equations are very sensitive to noise in the data when the concentrations in any two streams are similar. Hence, in situations like this study, where

there is a very low product yield resulting in a tailings composition that is often very similar to the feed composition, spurious zero or negative values can often occur.

The mass yield to product ($Y$) should, in theory, be independent of which species' assay values are used in Equation (1). Invariably, there will always be some variation due to random sampling and measurement errors, but the consistency of these data gives some indication of whether the samples were representative and the system was truly at a steady state. These calculated yield values are shown in Table 1.

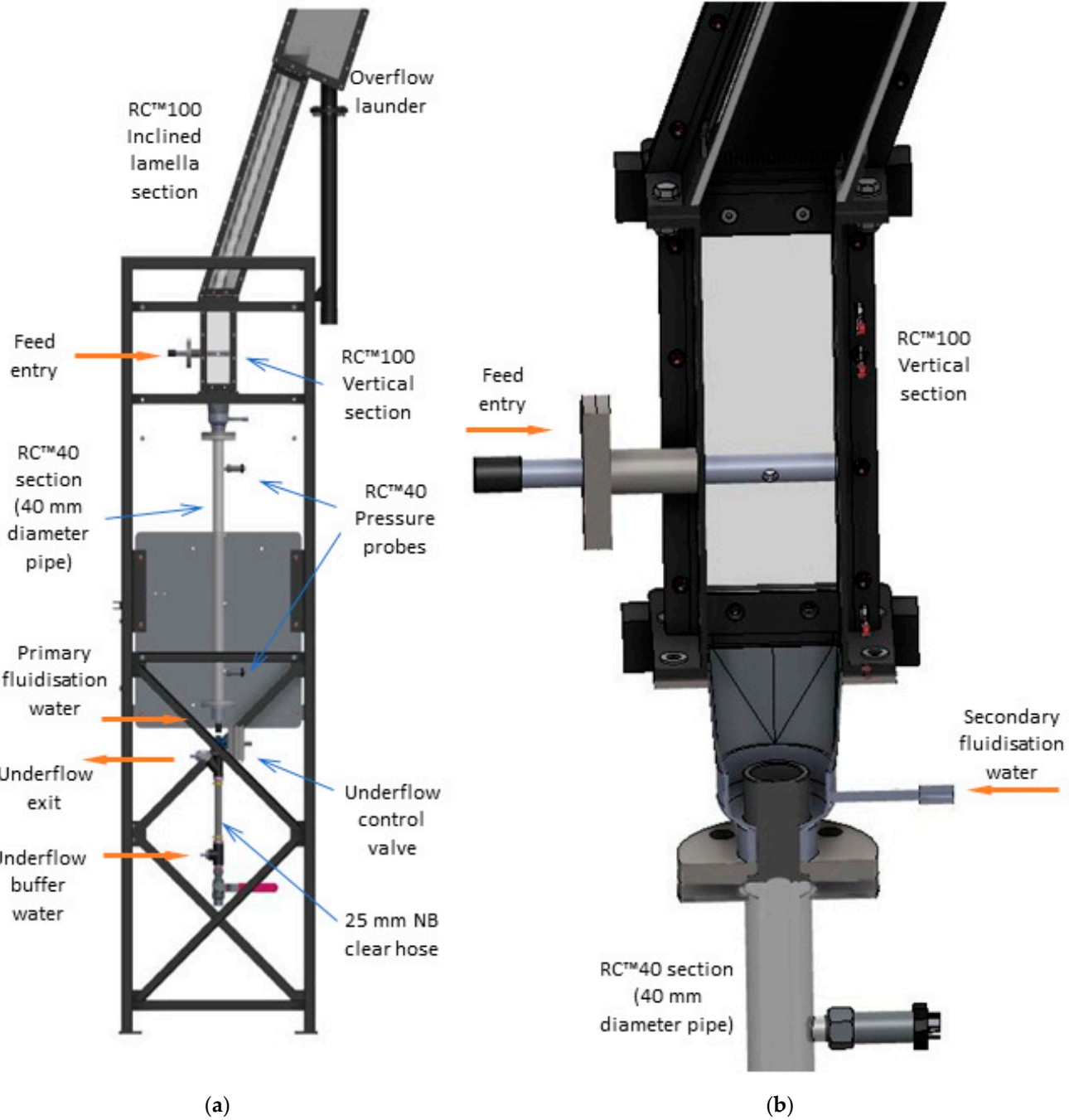

**Figure 3.** Rendered images of the continuous RC™100-40 system: (**a**) overall structure and (**b**) close-up details of the midsection.

**Table 1.** Raw XRF assay data for the head and wet split +0.045 mm and −0.045 mm fractions of the feed, underflow product and overflow reject streams from the continuous separation experiment. Mass yield (wt.%) and recovery (%) based on the raw assays of each species were calculated using the two-product formulae (Equations (1) and (2)). Grey font is used in cases where spurious negative or zero results occur. "Balance Yield" is the yield estimated by standard mass balancing of that set of raw assay data seeking to minimise the sum of the squares of the relative differences between the experimental and balanced assays over all species (Table A1). Yield and recovery error amplifications were estimated using the equations presented by Iveson and Galvin [23]. Bold is used to highlight the three densest species which are discussed in Section 3.

| Sample | Mass Fraction (wt.%) | Balance Yield (wt.%) | Density (RD) | Li$_2$O (wt.%) | Fe$_2$O$_3$ (wt.%) | Al$_2$O$_3$ (wt.%) | SiO$_2$ (wt.%) | TiO$_2$ (wt.%) | Mn (wt.%) | S (wt.%) | P (wt.%) | Sn (wt.%) | Ta$_2$O$_5$ (wt.%) | Nb$_2$O$_5$ (wt.%) | Na$_2$O (wt.%) | PbO (wt.%) | CaO (wt.%) | MgO (wt.%) | K$_2$O (wt.%) | Rb (wt.%) | U$_3$O$_8$ (wt.%) | ThO$_2$ (wt.%) | LOI (wt.%) |
|---|---|---|---|---|---|---|---|---|---|---|---|---|---|---|---|---|---|---|---|---|---|---|---|
| **Head samples:** | | | | | | | | | | | | | | | | | | | | | | | |
| Feed | | | 2.8102 | 0.828 | 1.880 | 15.130 | 66.240 | 0.105 | 0.118 | 0.808 | 0.801 | **1.296** | **0.561** | **0.199** | 4.040 | 0.015 | 3.320 | 0.500 | 1.436 | 0.094 | 0.007 | 0.002 | 1.29 |
| Product | | | | 0.255 | 16.400 | 1.750 | 4.280 | 0.324 | 1.326 | 12.258 | 0.222 | **34.553** | **13.305** | **4.359** | 0.160 | 0.214 | 1.200 | 0.140 | 0.141 | 0.006 | 0.163 | 0.014 | 7.78 |
| Reject | | | | 0.858 | 1.260 | 15.780 | 68.550 | 0.098 | 0.062 | 0.283 | 0.827 | **0.065** | **0.068** | **0.027** | 4.250 | 0.004 | 3.320 | 0.530 | 1.467 | 0.094 | 0.002 | 0.001 | 0.98 |
| Yield (wt.%) | | 4.0 | | 5.0 | 4.1 | 4.6 | 3.6 | 3.1 | 4.4 | 4.4 | 4.3 | **3.6** | **3.7** | **4.0** | 5.1 | 5.2 | 0.00 | 7.7 | 2.3 | −0.8 | 3.1 | 4.5 | 4.6 |
| Recovery (%) | | | | 1.5 | 35.7 | 0.5 | 0.2 | 9.6 | 49.8 | 66.5 | 1.2 | **95.2** | **88.3** | **87.0** | 0.2 | 74.7 | 0.00 | 2.2 | 0.2 | −0.1 | 69.5 | 31.5 | 27.5 |
| Upgrade (-) | | | | 0.3 | 8.7 | 0.1 | 0.1 | 3.1 | 11.2 | 15.2 | 0.3 | **26.7** | **23.7** | **21.9** | 0.0 | 14.3 | 0.4 | 0.3 | 0.1 | 0.1 | 22.3 | 6.9 | 6.03 |
| Yield Error Amplification, $A_Y$ (-) | | | | 39.0 | 4.3 | 32.9 | 40.6 | 21.2 | 3.0 | 2.2 | 43.6 | 1.5 | 1.6 | 1.6 | 27.2 | 1.9 | Error | 23.6 | 65.5 | 190.7 | 2.1 | 5.0 | 5.9 |
| Recovery Error Amplification, $A_R$ (-) | | | | 40.4 | 2.9 | 34.3 | 42.0 | 19.8 | 1.6 | 0.8 | 45.0 | 0.1 | 0.2 | 0.2 | 28.6 | 0.5 | Error | 25.0 | 66.9 | 189.3 | 0.7 | 3.6 | 4.5 |
| **+0.045 mm wet-screened samples:** | | | | | | | | | | | | | | | | | | | | | | | |
| Feed | 93.1 | | 2.8102 | 0.777 | 1.81 | 15.11 | 66.62 | 0.104 | 0.103 | 0.761 | 0.792 | **1.257** | **0.527** | **0.176** | 4.11 | 0.011 | 3.27 | 0.50 | 1.416 | 0.091 | 0.008 | 0.003 | 1.22 |
| Product | 93.8 | | 5.4846 | 0.327 | 18.16 | 2.11 | 5.28 | 0.338 | 1.205 | 11.534 | 0.266 | **32.182** | **12.486** | **4.017** | 0.19 | 0.200 | 1.35 | 0.17 | 0.153 | 0.006 | 0.132 | 0.012 | 8.65 |
| Reject | 86.1 | | 2.7609 | 0.829 | 1.08 | 15.64 | 69.56 | 0.096 | 0.054 | 0.236 | 0.801 | **0.048** | **0.045** | **0.019** | 4.33 | 0.004 | 3.29 | 0.49 | 1.430 | 0.096 | 0.002 | 0.001 | 0.86 |
| Yield (wt.%) | | 4.1 | | 10.4 | 4.3 | 3.9 | 4.6 | 3.3 | 4.3 | 4.6 | 1.7 | **3.8** | **3.9** | **3.9** | 5.3 | 3.6 | 1.0 | −3.1 | 1.1 | 4.9 | 4.8 | 10.5 | 4.6 |
| Recovery (%) | | | | 4.4 | 42.9 | 0.5 | 0.4 | 10.7 | 49.8 | 70.4 | 0.6 | **96.3** | **91.8** | **89.6** | 0.2 | 64.9 | 0.4 | −1.1 | 0.1 | 0.3 | 80.6 | 49.1 | 32.8 |
| Upgrade (-) | | | | 0.4 | 10.0 | 0.1 | 0.1 | 3.3 | 11.7 | 15.2 | 0.3 | **25.6** | **23.7** | **22.8** | 0.0 | 18.2 | 0.4 | 0.3 | 0.1 | 0.1 | 16.8 | 4.7 | 7.1 |
| Yield Error Amplification, $A_Y$ (-) | | | | 21.1 | 3.5 | 40.3 | 32.0 | 18.4 | 3.0 | 2.0 | 124.5 | 1.5 | 1.5 | 1.6 | 26.4 | 2.2 | 231.2 | 70.7 | 143.0 | 29.3 | 1.8 | 3.3 | 4.8 |
| Recovery Error Amplification, $A_R$ (-) | | | | 22.5 | 2.1 | 41.7 | 33.5 | 17.0 | 1.6 | 0.6 | 125.9 | 0.1 | 0.1 | 0.2 | 27.8 | 0.8 | 232.6 | 69.3 | 144.5 | 30.7 | 0.4 | 1.9 | 3.4 |
| **−0.045 mm wet-screened samples:** | | | | | | | | | | | | | | | | | | | | | | | |
| Feed | 6.9 | | | 0.776 | 3.78 | 15.12 | 57.66 | 0.189 | 0.311 | 0.911 | 0.988 | **3.658** | **1.793** | **0.584** | 3.74 | 0.049 | 4.10 | 0.73 | 1.399 | 0.082 | 0.031 | 0.005 | 1.71 |
| Product | 6.2 | | | 0.014 | 4.63 | 0.31 | 1.21 | 0.260 | 1.944 | 1.810 | 0.023 | **50.691** | **18.271** | **5.339** | 0.05 | 0.385 | 0.47 | 0.03 | 0.080 | 0.003 | 0.341 | 0.025 | 1.36 |
| Reject | 13.9 | | | 0.880 | 3.71 | 17.06 | 59.63 | 0.192 | 0.193 | 0.632 | 1.112 | **0.812** | **0.607** | **0.216** | 3.79 | 0.033 | 4.46 | 0.91 | 1.752 | 0.096 | 0.016 | 0.004 | 1.84 |
| Yield (wt.%) | | 6.0 | | 12.0 | 7.6 | 11.6 | 3.4 | −4.4 | 6.7 | 23.7 | 11.4 | **5.7** | **6.7** | **7.2** | 1.3 | 4.5 | 9.0 | 20.5 | 21.1 | 15.5 | 4.4 | 2.0 | 27.1 |
| Recovery (%) | | | | 0.2 | 9.3 | 0.2 | 0.1 | −6.1 | 42.1 | 47.1 | 0.3 | **79.1** | **68.4** | **65.7** | 0.02 | 35.7 | 1.0 | 0.8 | 1.2 | 0.6 | 49.2 | 10.7 | 21.5 |
| Upgrade (-) | | | | 0.0 | 1.2 | 0.0 | 0.0 | 1.4 | 6.3 | 2.0 | 0.0 | **13.9** | **10.2** | **9.1** | 0.01 | 7.9 | 0.1 | 0.0 | 0.1 | 0.0 | 11.1 | 5.5 | 0.8 |
| Yield Error Amplification, $A_Y$ (-) | | | | 10.6 | 76.4 | 11.0 | 41.4 | 89.1 | 3.7 | 4.6 | 11.3 | 1.8 | 2.1 | 2.2 | 105.8 | 4.3 | 16.1 | 5.7 | 5.6 | 8.0 | 3.0 | 15.9 | 18.6 |
| Recovery Error Amplification, $A_R$ (-) | | | | 12.0 | 75.0 | 12.4 | 42.8 | 90.5 | 2.3 | 3.2 | 12.7 | 0.4 | 0.7 | 0.8 | 107.2 | 2.9 | 17.5 | 7.1 | 7.0 | 9.5 | 1.6 | 14.5 | 20.0 |

Samples of each stream were then wet-screened at 0.045 mm, and both the +0.045 mm and −0.045 mm fractions were again assayed. These mass splits and assays are also shown in Table 1, together with the calculated yields and recoveries. This paper is focused on the analysis of the wet-screened +0.045 mm material unless otherwise specified.

### 2.3. Batch Fractionation Experiments—Source of Flow Fractionation Samples

The wet-screened +0.045 mm fraction of the feed, product and reject samples from the continuous experiment were each then flow fractionated using a batch RC™ (60 mm × 100 mm cross-section). The vertical lower section was 1000 mm tall. The inclined section was 1000 mm long and was split into 22 parallel channels with 2 mm spacing. LST solution was used which had a density of 2550 kg/m³. A 38 μm screen was used to collect the solids in the overflow for each flow fraction, with the LST solution being recirculated (Figure 4).

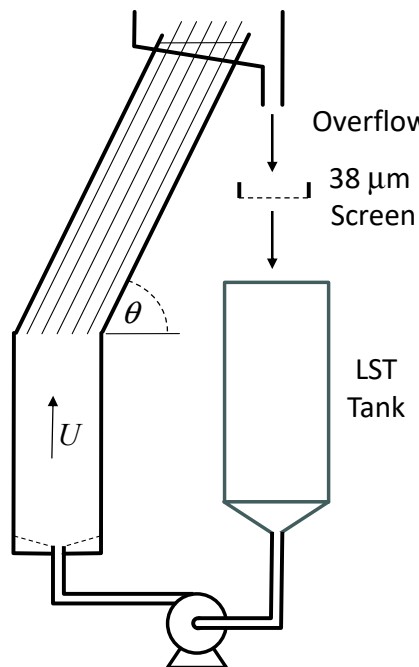

**Figure 4.** Schematic of batch fractionation setup.

A batch of about 3.0 kg of solids was placed in the unit and initially fluidised at a low flow rate. Each sample was then fractionated into 10 flow fractions by increasing the fluidisation rate in steps, waiting each time until there was negligible further mass reporting to the overflow before making the next step increase the flow rate. The solids remaining inside the unit at the end of the fractionation experiment, referred to as "Flow 11", were also collected and analysed. Thus, there were 11 "flow" fractions in total for each of the three streams. Note that flows with the same label number from different streams cannot be directly compared since they may have been collected at quite different flow rates, after different lengths of time and with different levels of hindered settling inside the unit. Hence, they may have quite different average particle densities. Each of these flow fractions then had its density measured by gas (helium) pycnometry and was assayed using XRF. These raw data are presented in Table B1.

Each of the 11 flow fractions from the 3 streams was then screened into 7 size intervals (+0.106 mm, +0.090 mm, +0.075 mm, +0.063 mm, +0.053 mm, +0.045 mm, −0.045 mm). This gave 11 × 7 = 77 portions for each stream. Note, to help clarify what is being referred to, in this paper, the term "fraction" is used to refer to one of the 11 flow fractions, "interval" is used to refer to one of the 7 size intervals, and "portion" refers to one of the resulting 77 flow × size portions. Where sufficient material was present (35–40 g), these portions were

also assayed by XRF, and their densities were measured using gas pycnometry. These raw data are presented in Tables B2–B4.

The mass fraction, density and assay of material in each size interval of the original wet-screened +0.045 mm samples prior to flow fractionation were back-calculated by recombining the data from each flow portion for that size interval (see Table B5).

*2.4. Data Analysis*

When the algorithm developed by Galvin et al. [20] was applied to the flow fraction-ation data, the algorithm did not give sensible results due to the extremely low yield of material to the product and a lack of data about the distribution of high-density material in the feed as noted in the discussion in Section 3.1. Instead, an alternative approach was developed to estimate the partition curve based on the distribution of species in the feed, product and reject samples.

The data from all the portions from a given stream were arranged in order of decreasing density. In cases where there was insufficient mass in an interval to measure its properties, it was assumed to have the same properties as the nearest adjacent size interval in that flow fraction whose properties had been measured. Thus, in this analysis, the masses of size Intervals 6 and 7 were combined into one mass assumed to have the same properties as Interval 6, and sometimes the masses of size Intervals 1 and 2 and possibly even Interval 3 were also considered to be a single mass with the properties of Interval 2 or 3 as applicable. Then, the cumulative mass fraction greater than the average density of a particular portion was calculated by summing the masses of all the higher-density portions plus half of the mass of the portion of interest. This value was then plotted versus the measured average density of that portion. This approach assumes that the mass of the portion is split evenly on either side of its average density. This assumption may break down if there is a long tail of either high- or low-density material in a portion. This approach also assumes that there is no material in any portion with a density outside of the range between the average density of the two adjacent higher- and lower-density portions. The accuracy of this assumption depends on how well the batch flow fractionation method separated particles based only on their density.

To highlight the occurrence of material in specific density bands, it is desirable to plot the data in a frequency form, which involves a plot of the negative slope of the cumulative mass versus density distribution. Due to the uneven density increment widths between adjacent portions, calculating this slope based on a linear fit between two adjacent points generated a large amount of variability. Instead, the slope at the density of portion $i$ was based on a quadratic fit through that point and the two adjacent data points $i - 1$ and $i + 1$. It can be easily shown that the fit of the parabolic function $y = ax^2 + bx + c$ through a set of three data points $(x_1, y_1)$, $(x_2, y_2)$ and $(x_3, y_3)$, is given by:

$$a = \frac{y_1(x_3 - x_2) + y_2(x_1 - x_3) + y_3(x_2 - x_1)}{(x_1 - x_2)(x_2 - x_3)(x_3 - x_1)} \tag{3a}$$

$$b = \frac{y_1(x_2^2 - x_3^2) + y_2(x_3^2 - x_1^2) + y_3(x_1^2 - x_2^2)}{(x_1 - x_2)(x_2 - x_3)(x_3 - x_1)} \tag{3b}$$

$$c = \frac{y_1 x_2 x_3(x_3 - x_2) + y_2 x_3 x_1(x_1 - x_3) + y_3 x_1 x_2(x_2 - x_1)}{(x_1 - x_2)(x_2 - x_3)(x_3 - x_1)} \tag{3c}$$

and so the slope at the middle point $(x_2, y_2)$ is given by:

$$\left.\frac{dy}{dx}\right|_{x_2} = \frac{y_3(x_1 - x_2)^2 + y_2(x_3^2 - 2x_2x_3 + 2x_1x_2 - x_1^2) - y_1(x_2 - x_3)^2}{(x_1 - x_2)(x_2 - x_3)(x_3 - x_1)} \tag{4}$$

where we set $1 = i - 1$, $2 = i$ and $3 = i + 1$.

When a partition curve was fitted through experimental data, the following two-parameter model was used [22]:

$$P(D) = \frac{1}{1 + \exp\left[\ln(3)\frac{D_{50}-D}{Ep}\right]} \tag{5}$$

where $P(D)$ is the probability of a particle of density $D$ reporting to the high-density underflow product, $D_{50}$ is the cut point density and $Ep = (D_{75} - D_{25})/2$ is the *Ecart Probable* (probable error), where $D_{75}$, $D_{50}$ and $D_{25}$ are, respectively, the densities of particles with 75%, 50% and 25% chances of reporting to the underflow product. This curve is symmetrical and has complete closure at both ends.

There were too many assays to simultaneously perform a global mass balance using MS® Excel® (Version 2308, 64-bit). So, instead, mass balancing was performed three times in order to estimate the mass yield $Y$ of various parts of the feed to the product. The first set of mass balancing was on the 60 head assays (20 each for feed, product and reject), the second was on the 60 assays of the wet-screened +0.045 mm material and the third mass balancing was on the 60 assays of the −0.045 mm wet-screened material. In each case, the objective function $G$ was defined as

$$G = \sum_{i=1}^{60}\left(\frac{x_i^{\mathrm{B}} - x_i^{\mathrm{E}}}{x_i^{\mathrm{E}}}\right)^2 \tag{6}$$

where $x_i^{\mathrm{E}}$ is the $i$th experimental assay value, $x_i^{\mathrm{B}}$ is the $i$th balanced value. The Solver function in MS Excel was then used to vary the guess yield values and 60 guess assays in such a way that the objective function was minimised subject to the 20 species steady-state mass balance constraints being satisfied. The results are shown in Table A1 (Appendix A), which also shows the relative percentage adjustment of each raw assay value required to balance the data.

## 3. Results and Discussion

Table 1 shows the raw XRF (X-ray fluorescence) data for 20 species (reported as pure elements or their oxides, plus also LOI, "Loss on Ignition" at 1000 °C) in the head feed, product and reject samples. The feed with a head grade of 1.296% Sn, 0.56% Ta and 0.199% Nb was upgraded to a product with 34.553% Sn (upgrade of 26.7), 13.305% Ta (23.7 fold upgrade) and 4.359% Nb (21.9 fold upgrade). Using these raw assay values in Equation (2) suggests that the recoveries were 95.2% for Sn, 88.3% for Ta and 87.0% for Nb, respectively. Table 1 also shows the raw XRF assays for the wet-screened +0.045 mm and −0.045 mm fractions of all three stream samples, together with the relative masses in each size fraction. For the +0.045 mm fraction, the recoveries based on applying Equation (2) to these raw data were 96.3%, 91.8% and 89.6% for the Sn, Ta and Nb, respectively, at 25.6-, 23.7- and 22.8-fold upgrades. Respectable recoveries and upgrades were also obtained for the much harder to beneficiate −0.045 mm material.

As mentioned in Section 2.2, a feature of Equations (1) and (2) is that they can give spurious zero or negative results for species with similar concentrations in both feed and tailings streams. This is evident in Table 1. Iveson and Galvin [23] present simple expressions for estimating the error amplification in the calculated yield ($A_{\mathrm{Y}}$) and recovery ($A_{\mathrm{R}}$), which are included in Table 1. The spurious negative values only occur for species with large (>60 fold) estimated error amplification. Conversely, it can be seen that for the three dense species of interest (Sn, Ta, Nb), the calculated yields should have similar relative errors to the raw assays (since $A_{\mathrm{Y}}$ ~1), and the calculated recoveries are expected to have much less relative error that the raw assay data ($A_{\mathrm{R}}$ ~0.1) and so should be quite accurate.

Mass balancing was independently performed on each of these three sets of raw assay data. This gave similar recovery and upgrade results (Table A1). The average magnitude

of the mass balancing relative adjustments to the raw assay data was 1.8% for the head assays, 2.1% for the +0.045 mm assays and 3.4% for the −0.045 mm assays. This high level of consistency suggests that the system was indeed operating close to a steady state and that the data set is reliable.

Mass balancing of the +0.045 mm assays (93% of the feed) gave a yield of 4.1%, whilst mass balancing the −0.045 mm assays (7% of the feed) gave a yield of 6.0%. These give a combined overall yield of 0.93 (4.1%) + 0.07 (6%) = 4.2%, which was in close agreement with the value of 4.0% found mass balancing the head assays (Table 1). Given that the +0.045 mm size fraction contributed 93% of the feed mass and given the need to restrict the LST flow fractionation to the +0.045 mm fraction, it was decided to base the yield in subsequent analysis on the mass-balanced value of 4.1% obtained directly from the +0.045 mm assay data.

The wet-split +0.045 mm fractions of each stream were then batch-fractionated into 11 flow fractions, each screened into seven size intervals, to give 77 portions for each stream. Where there was enough material present, these portions were then XRF assayed, and their densities were measured using gas pycnometry. Where there was insufficient mass for these measurements, the material was assumed to share the same density and assay as the nearest adjacent size interval in that flow whose properties were able to be measured. In the end, 56 feed portions, 60 product portions and 58 reject portions were sufficiently large to have their density and composition measured. Full tables of these flow fractions by size increment portion data are presented in Tables B2–B4.

The data from the same size fractions of each flow were combined to reconstitute and calculate the mass, average density and assay of each of the size intervals in the parent +0.045 mm wet-screened samples (Table B5). From these data, the recovery was calculated using Equation (2), together with the upgrade. Figure 5 shows these recovery and upgrade values for the three high-density components (Sn, Ta and Nb) as a function of particle size. Excellent performance was achieved. Stable recoveries were obtained down to 0.053 mm with average values of 95%, 92% and 90%, respectively, with a slow drop-off after that. There was a more than 20-fold upgrade for all three species in the 0.053–0.106 mm size range, with a drop in performance at each end.

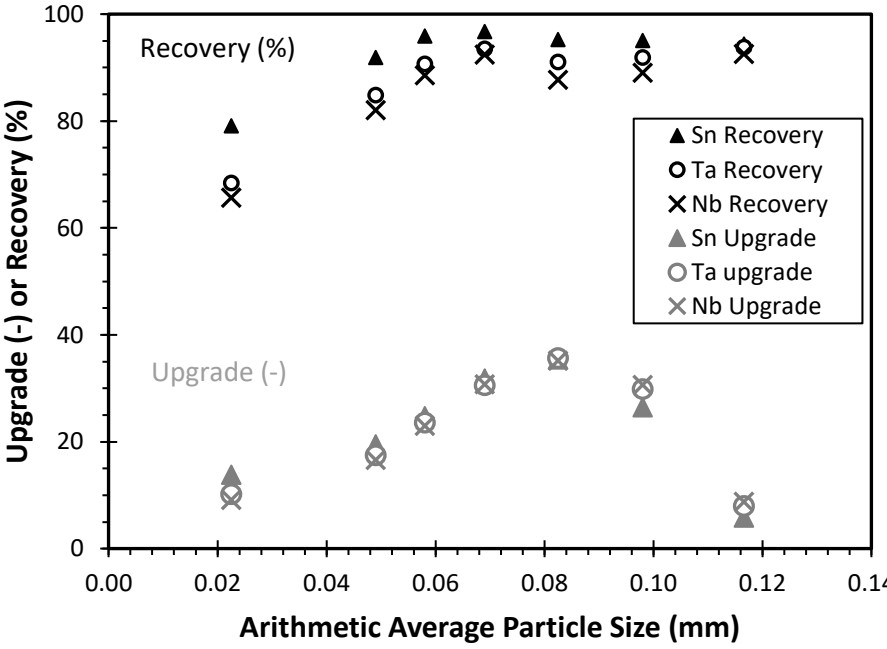

**Figure 5.** Upgrade and recovery of Sn, Ta and Nb plotted as a function of average particle size. Recovery calculated from Equation (2) applied to the reconstituted assays in each size interval (Table B5), except for the 0–0.045 mm size interval, which is based on the raw assays of the −0.045 mm wet-screened material (Table 1).

### 3.1. Application of the Algorithm of Galvin et al. [20]

Figure 6 shows the cumulative mass fraction plotted versus density for the +0.045 mm wet-split samples. Data were arranged in descending order according to density, with the cumulative mass at the average density of a given portion being calculated as the sum of all higher-density portion masses plus half of the mass of the portion of interest (see Section 2.4). Hence, reading from high to low density, each curve starts slightly above 0% (since half the mass of the densest portion is assumed to be denser than that portion's average density but with an unknown upper-density limit) and rises to slightly below 100% (since half the mass of the least dense portion is less dense than that portion's average density and has an unknown lower-density limit).

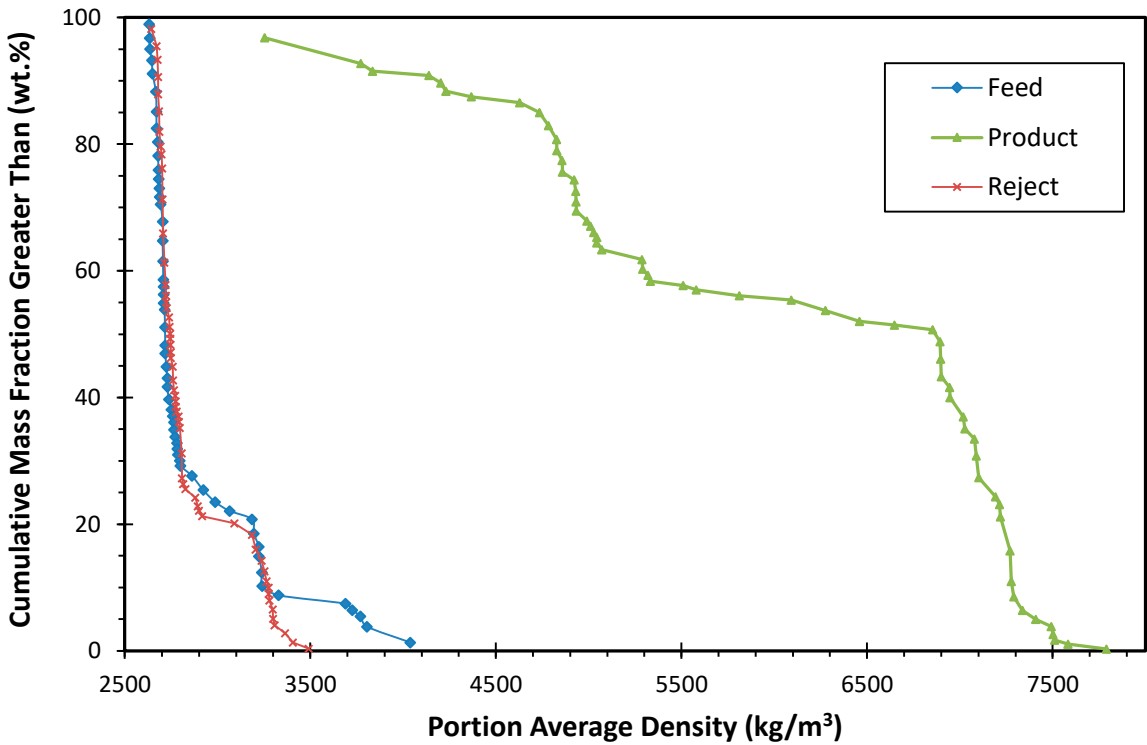

**Figure 6.** Cumulative mass fraction plotted as a function of average density for the combined flow fraction × size interval portions of the +0.045 mm wet-screened feed, product and reject samples. Portions with insufficient mass for pycnometry were assigned the same properties as the portion in the nearest adjacent size interval in that same flow fraction.

The product density distribution in Figure 6 shows that there was material in the system with densities at least as high as 7790 kg/m$^3$. However, the highest average density measured in any of the flow × size portions of the feed was only 4038 kg/m$^3$ for the 71 g of −0.063 + 0.053 mm material in Flow 11, as shown in Table B2. Clearly, this 71 g portion of material must have contained some particles with densities as high as 7790 kg/m$^3$. However, due to the low numbers of these high-density particles (yield ~4.1%), the batch flow fractionation of the feed sample was terminated before this high-density material was split into more discrete density intervals.

As we shall see in Section 3.3, the cut point of the continuous separator is estimated to have been around 4000 kg/m$^3$. So this means that the flow fractionation did not provide any information about the distribution of near-density material in the feed sample from which to sensibly infer the separation performance. So, when the algorithm of Galvin et al. [20] is applied to the data, both for the narrow size fractions and for the overall +0.045 mm material, due to the absence of any near-density material, it frequently converges to a solution with zero *Ep* value, which is clearly implausible.

This example highlights an important aspect of how to conduct the batch flow fractionation experiments that was mentioned previously [20] but bears stating again, namely, that the objective is not to split each feed, product and reject sample into roughly equally sized flow fractions. Rather, to accurately estimate the partition curve, the objective must be to obtain a high resolution near the density cut point. So, for cases with extremely low yields of a high-density product, this means taking multiple small samples by using small steps in flow rate towards the end of fractionating the feed. For the reject samples, multiple flow rates are required at the end of the fractionation when only high-density material remains. For the product sample, multiple small steps in flow rate are needed at the commencement of the fractionation.

### 3.2. Approximate Partition Analysis Using the "Tracer Species Recovery" Method

The cumulative density distributions in Figure 6 show regions of very steep rise, indicating the presence of large numbers of particles in certain discrete narrow ranges of density. This can be shown more clearly by plotting the negative slope of the cumulative mass versus density curve, which we herein refer to as the "density frequency" (wt.%/$\Delta$(kg/m$^3$)). The slope at a particular density value was calculated using Equation (4), which fits a parabola through a data point and its two neighboring data points. These density frequency distributions are shown in Figure 7 for the +0.045 mm wet-screened feed, product and reject samples. The area under each curve equals 100% (unity).

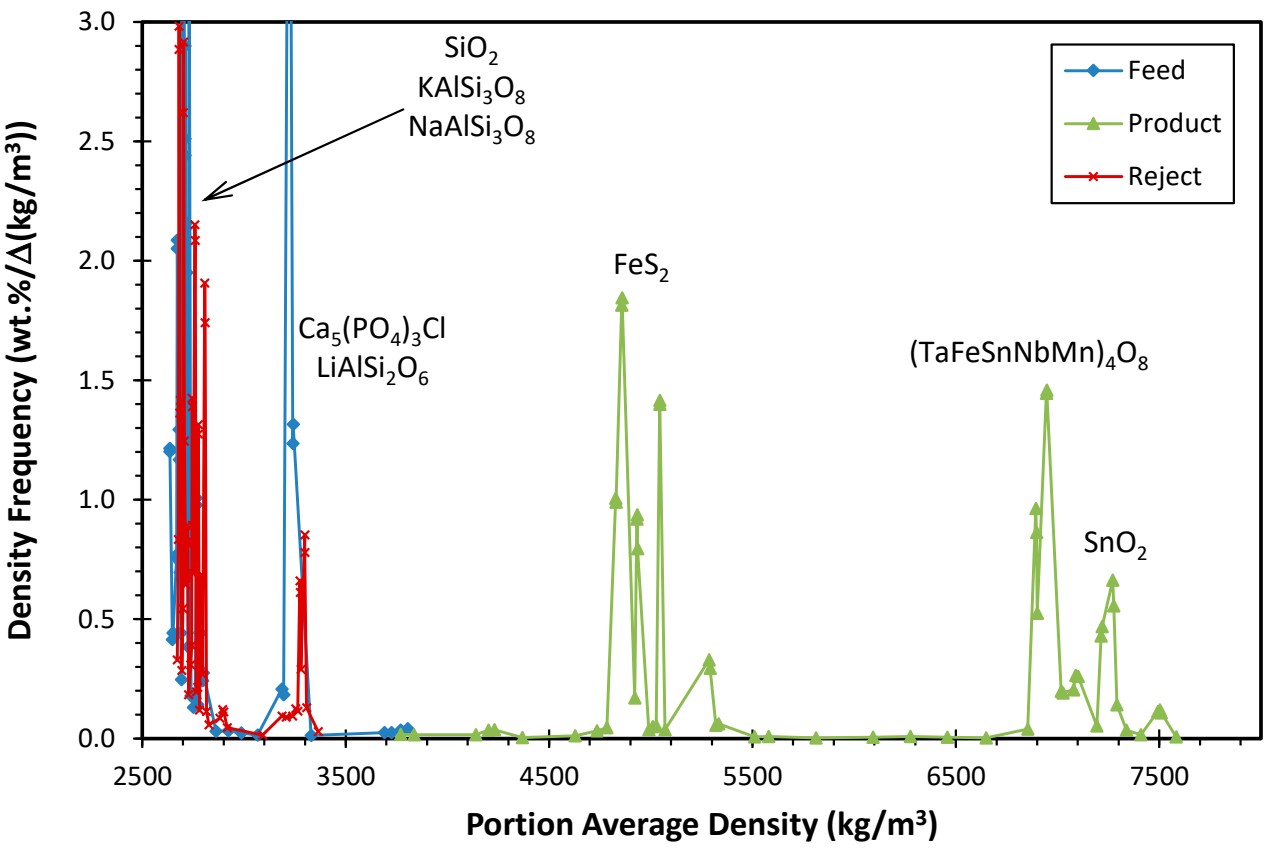

**Figure 7.** Density frequency distributions of the combined flow fractionation × size interval portions of the +0.045 mm wet-screened feed, product and reject stream samples. Labels indicate the major minerals most likely responsible for each peak (see Table 2 and discussion of the XRD data below). Portions with insufficient mass for pycnometry were assigned the same properties as the portion in the nearest adjacent size interval in that same flow fraction.

**Table 2.** Average density associated with the 10 potential tracer elements (shown in bold) compared with literature values. Peak densities from Figure 11 are reported for highest peaks in a $\pm 50$ kg/m$^3$ density interval provided density frequency is above 1 wt.%/$\Delta$(kg/m$^3$). Average densities calculated using Equation (7), based on product data for Sn (cassiterite), Ta and Nb (ixiolite), reconstituted composite feed data for Fe and S and reject data for the other five species.

| XRF Label | Likely Actual Mineral (Based on XRD) | Peak Densities in Figure 11 (kg/m$^3$) | Average Density, $r_{av,i}$ (kg/m$^3$) | Literature Density (kg/m$^3$) | Literature Density Source |
|---|---|---|---|---|---|
| Sn | Cassiterite **Sn**O$_2$ | 7270, 6947, 6893 | 6936 UF Product | 6800–7100 | [21] |
| Ta$_2$O$_5$ Nb$_2$O$_5$ | Ixiolite (**Ta**FeSn**Nb**Mn)$_4$O$_8$ | Ta 6947, 3253 Nb 6947, 6893, 4931, 3253 | Ta 6559 Nb 6387 UF Product | 6940–7230 | [23] |
| Fe$_2$O$_3$ S | Pyrite **FeS**$_2$ | 5044, 4859, 3298, 3253 | Fe 3783 S 4272 | 5000 | [21] |
| P | Chlorapatite Ca$_5$(**P**O$_4$)$_3$Cl | 3253 | OF Reject 3239 | 3100–3200 | [23] |
| Li$_2$O | Spodumene **Li**AlSi$_2$O$_6$ | 3253, 2807 | OF Reject 3191 | 3100–3200 | [21] |
| SiO$_2$ | Quartz **Si**O$_2$ | 2759, 2703 | OF Reject 2794 | 2650–2660 | [21] |
| K$_2$O | Microcline **K**AlSi$_3$O$_8$ | 2807, 2749, 2680 | OF Reject 2770 | 2560 | [23] |
| Na$_2$O | Albite **Na**AlSi$_3$O$_8$ | 2745, 2681 | OF Reject 2743 | 2600–2700 | [21] |

The density frequency distribution plot clearly shows that particle densities are not uniformly spread across the entire density range but rather are concentrated in narrow bands. A particular point to note is the near total absence of material in the density range of 3400–4700 kg/m$^3$, which will have a significant bearing on the attempts below to determine the partition curve. These narrow bands of density likely indicate the presence of well-liberated particles of a single mineral phase and, hence, single density. The fact that particles are found with such narrow bands of density shows that the batch fractionation method was effectively separating particles on the basis of their density; the presence of misplaced particles would have broadened these bands out. The dominant species in each of these density bands can be identified by plotting the cumulative distribution of each element versus density. This is shown for the product sample in Figure 8 and for the reject sample in Figure 9 (the feed distribution looks very similar to the reject distribution).

Figures 8 and 9 reveal that some elements were strongly concentrated in certain ranges of density. Species that exclusively reside in only a very narrow range of density can potentially be treated as if they are density tracer particles. Clear candidates to be density tracer species are Sn, Ta and Nb, which are mostly found at densities above 6850 kg/m$^3$, and Si, K and Na, which are found almost entirely at densities below 2800 kg/m$^3$. There is also a band of material with densities in the range of 3200–3400 kg/m$^3$ that contains a large proportion of the total Li and P (Figure 9). Some 72% of both the Fe and S in the product are found in the density range of 4600 to 5510 kg/m$^3$. However, there are also significant amounts of Fe and S in the reject sample, appearing in the density range of 3200–3400 kg/m$^3$. So, given that the product represents only about 4.1% of the total feed mass, we must be cautious in interpreting the Fe and S data from only a single output stream.

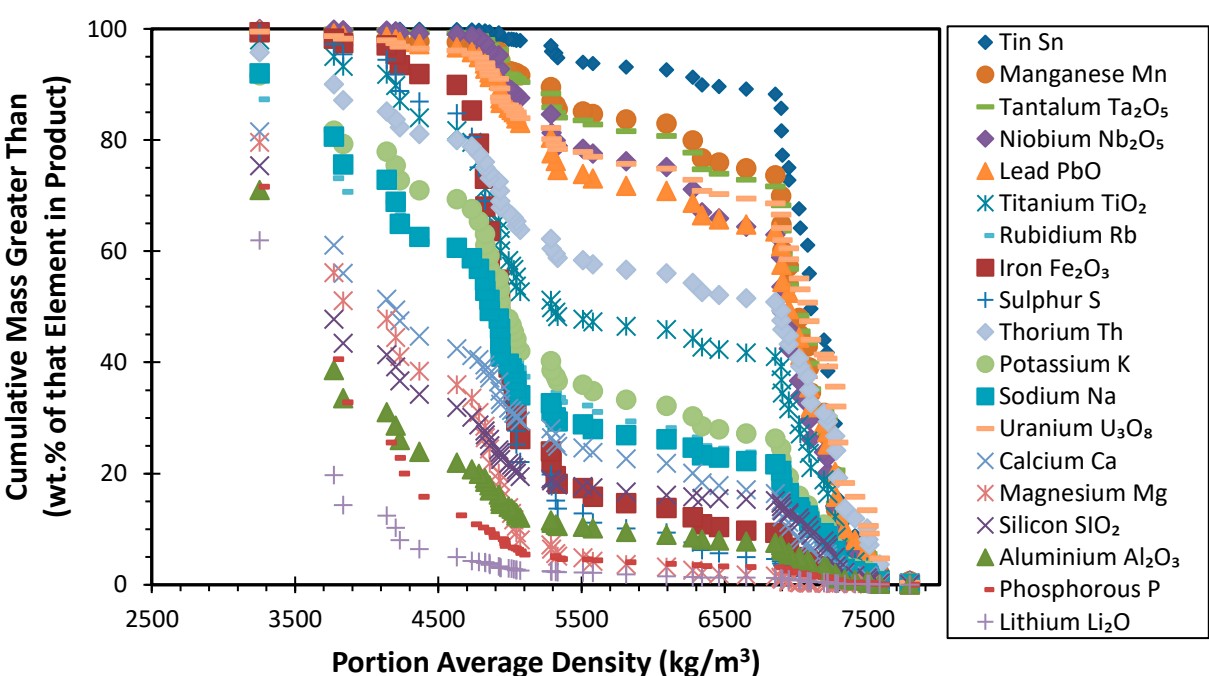

**Figure 8.** Product cumulative mass plotted as a function of density for each of the 19 species (elements or their oxides) plotted versus the average portion density for the flow × size portions of the +0.045 mm wet-screened sample. Percentages are based on the total amount of that species in the sample and so all 19 sets of data rise towards 100% at the lowest density. Portions with insufficient mass for assay and pycnometry were assigned the same properties as the portion in the nearest adjacent size interval in that same flow fraction.

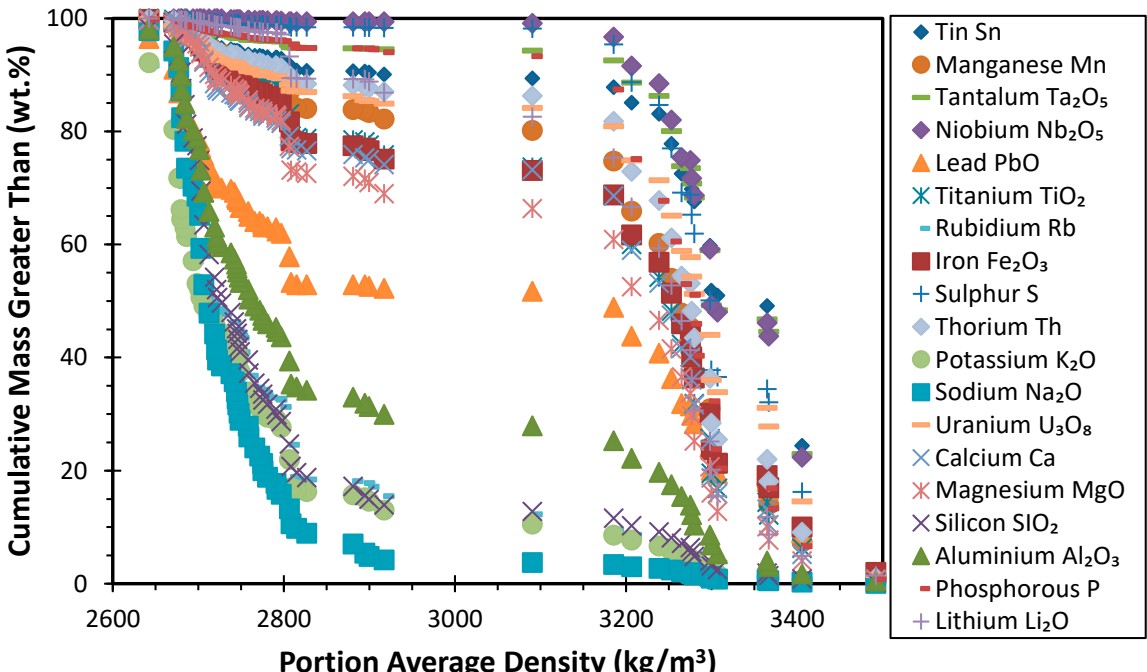

**Figure 9.** Reject cumulative mass plotted as a function of density for each of the 19 species (elements or their oxides) versus average portion density for the flow × size portions of the +0.045 mm wet-screened sample. Percentages are based on the total amount of each species in the sample, and so all 19 sets of data rise to 100% at the lowest density. Portions with insufficient mass for assay and pycnometry were assigned the same properties as the portion in the nearest adjacent size interval in that same flow fraction.

To better understand the distribution of these candidate tracer species, it is desirable to plot their distributions in the feed. However, the raw feed data cannot be used since there is a very poor resolution of the high-density material. So, instead, the raw feed distributions of each species were back-calculated by combining the product and rejecting data in the ratio 0.041:0.959 to reflect the mass yield to the product of 4.1% (Table 1). Figures 10 and 11, respectively, show the cumulative mass and the density frequency distributions of the ten (10) candidate tracer species discussed above in this reconstituted feed sample. These figures highlight the suitability of Sn, P, Li, Si, K and Na as tracer particles, with over 80% of each of these six elements being found in very narrow density ranges. Ta and Nb are borderline, with only ~60% of each found in the highest density range, and then ~30% found between 4600 and 5400 kg/m$^3$ and the remaining ~10% found between 3100 and 3500 kg/m$^3$.

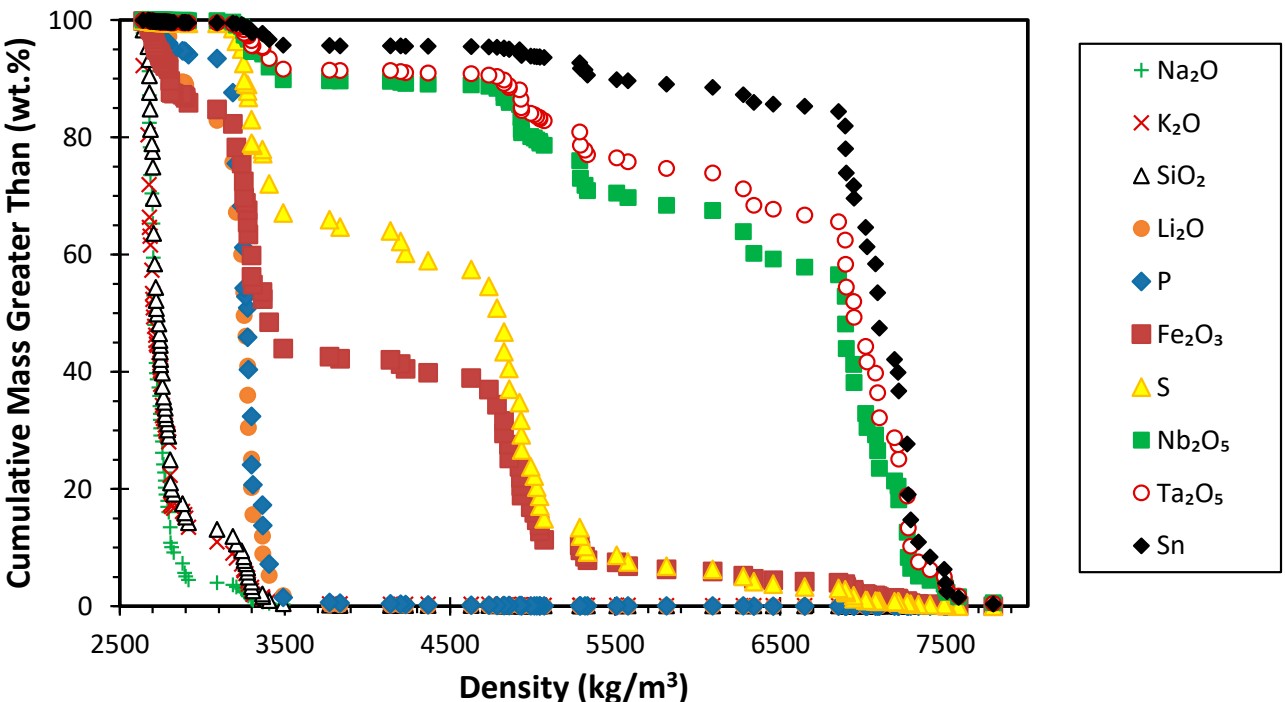

**Figure 10.** Cumulative mass plotted as a function of density distributions of selected species in the +0.045 mm wet-screened composite feed sample made by combining the product and reject data at a yield of 4.1%. Percentages are based on the total amount of that species in the sample, and so do not indicate amounts of the different species relative to each other.

The XRD data given in Appendix C suggest that the Na, Si and K peaks are likely due to the presence of albite ($NaAlSi_3O_8$), quartz ($SiO_2$) and microcline ($KAlSi_3O_8$), which were detected in the reject sample at levels of 29%, 38% and 5%, respectively, with densities around 2600–2700 kg/m$^3$. The Li and P peaks are likely due to the 10% of spodumene ($LiAlSi_2O_6$) and 5% of chlorapatite ($Ca_5(PO_4)_3Cl$) detected in the reject sample, which have densities of approximately 3100–3200 kg/m$^3$ and 3160–3220 kg/m$^3$, respectively [22,24]. The Sn is mainly present in the form of cassiterite ($SnO_2$) but is also present together with Ta and Nb as ixiolite $(TaFeSnNbMn)_4O_8$. Both are very dense minerals with densities around 7000 kg/m$^3$.

Unfortunately, Fe and S, which are the two elements of greatest potential interest for determining the shape of the partition curve near the density cut point, have a more complex distribution, with S and Fe being concentrated in different proportions in density bands around 5000 kg/m$^3$ and 3300 kg/m$^3$. Fe is also found in a third-density band down at 2700 kg/m$^3$. There is clearly a close association between Fe and S, which suggests that much of the iron is in the form of pyrite ($FeS_2$). This conclusion is supported by XRD data given in Appendix C, which indicate the presence of significant amounts of pyrite in the

product. However, the product also had some iron-bearing ixiolite at higher densities and small amounts of goethite (FeO(OH)) at lower densities. Meanwhile, the reject contained small amounts of hematite and iron-bearing clinochlore (Appendix C).

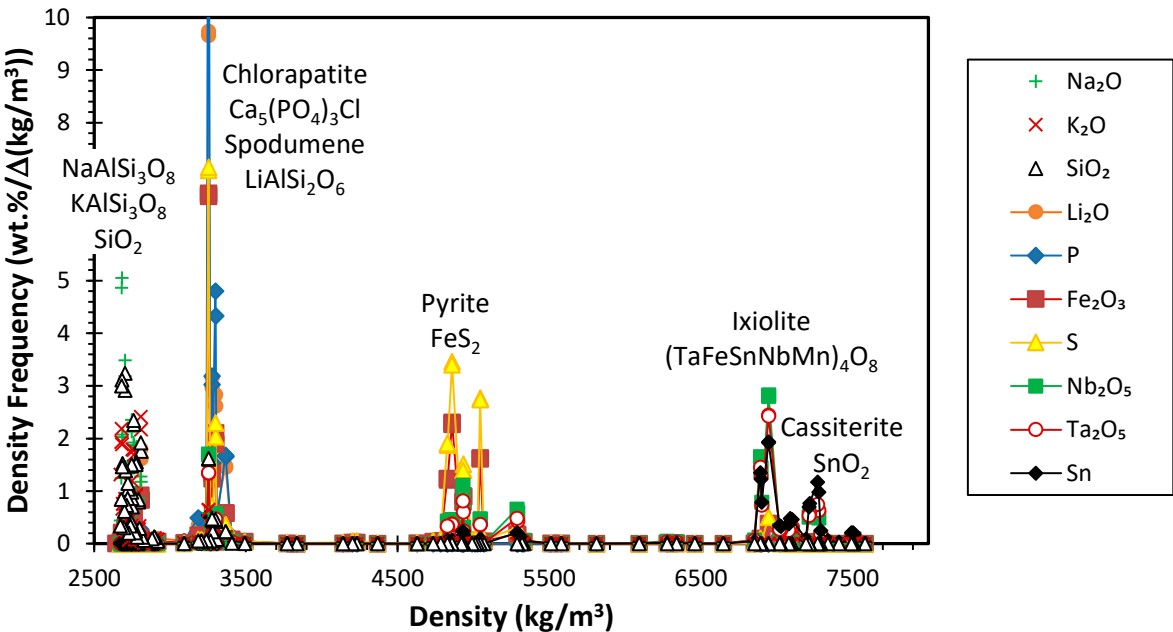

**Figure 11.** Density frequency mass distribution of selected species in the +0.045 mm wet-screened composite feed sample made by combining the product and reject data at a yield of 4.1%. Percentages are based on the total amount of that species in the sample, and so do not indicate amounts of the different species relative to each other. Labels indicate the mineral most likely responsible for the peak based on the XRD data (see Table 2).

Due to the noise and scatter in the density frequency data, to objectively calculate a density value to associate with each mineral band, the average density $\rho_{\mathrm{av},i}$ associated with species $i$ in a stream was found across the entire sample:

$$\rho_{\mathrm{av},i} = \frac{\sum_j m_j x_{i,j}}{\sum_j \frac{m_j x_{i,j}}{\rho_j}} \tag{7}$$

where $m_j$ and $\rho_j$ are the mass and average density of material in a given flow $\times$ size portion $j$ and $x_{i,j}$ is the assay of species $i$ in portion $j$ of a given stream. This average is not the density of any specific mineral but rather is a proxy density that incorporates the densities of all minerals associated with that species, and it is used here purely as an indicator for understanding the separation performance. Table 2 shows the average density associated with the 10 species of interest as calculated by Equation (7). For the high-density species (Sn, Ta and Nb), the average is based on the underflow product sample. For Fe and S, the density is calculated based on the reconstructed feed sample (Figures 9 and 10). For the other five low-density species, the average density is calculated using the reject overflow sample data. Also shown are the peak densities (with density frequency > 1 wt.%/$\Delta$(kg/m$^3$)) from Figure 10. There is reasonable agreement between at least one of the peak values and the literature density value. The calculated average density is offset somewhat because it includes the effects of other species.

Figure 12 shows the recoveries of different species as calculated by Equation (2) applied to the reconstituted assay data for each size fraction (Table B5) plotted versus the average density associated with the species (as calculated by Equation (7) and reported in Table 2). Note that any spurious recovery values less than zero are not included in this plot. This plot, which we herein call the "Tracer Species Recovery" method, gives an indication of what the partition functions may look like; it is, however, only indicative because none

of these species are found entirely at a single density. Even for one of the best cases, Sn, about 15% is found at densities less than 7000 kg/m$^3$ (Figure 10). So, there might have been a 100% recovery of particles with a density of 7000 kg/m$^3$, but this good performance is being masked by the lower recovery of the other 15% of the Sn that was found at lower densities, giving an overall recovery of "only" 95%. Similarly, some of the greater than zero recoveries of the low-density species may be due to some of those species being present in higher-density particles with higher partition values. These inaccuracies mean that the partition suggested by Figure 12 is likely to be less sharp (higher *Ep* value) than reality. Fitting partition curves through these data would also be problematic because Fe and S are the only two species with associated average densities near the density cut point, so their data would control the curve fit. However, these two species are both concentrated in more than one density interval (Figures 10 and 11), so they cannot be relied on to give an accurate partition value at a specific density. So, this approach does not conclusively demonstrate the true partition performance.

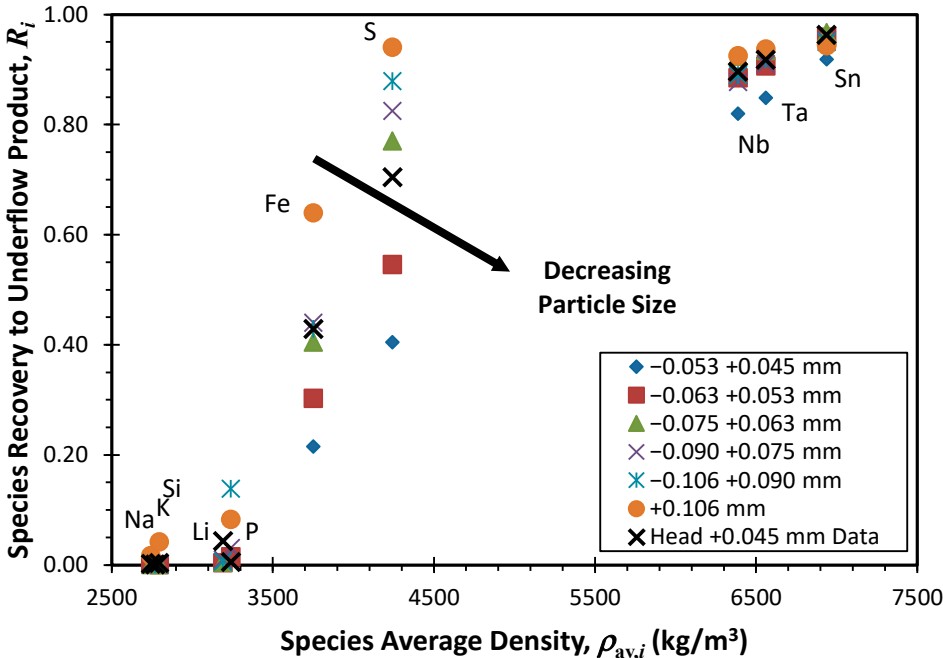

**Figure 12.** Recovery of selected tracer species calculated by applying Equation (2) to the reconstituted size interval assay data in Table B5 for the +0.045 mm wet-screened feed, product and reject samples plotted as a function of the average density associated with that species calculated using Equation (7) for the flow × size portion data (see Table 2).

The data in Figure 12 appear to be approaching 100% and 0% partition, respectively, at high and low densities with no signs of any tails (i.e., the curve has "good closure"). The data also suggest that as the particle size decreases, the density cut point shifts to higher densities and that the sharpness of separation worsens at finer sizes, which are trends observed previously in REFLUX™ Classifiers [15].

### 3.3. Partition Curve Estimation Using the "Full Species Distribution" Method

The weakness of the "Tracer Species Recovery" approach in Section 3.2 is that it assumes that each of the selected species is found only at a single density, but we know this is not true. Hence, an alternative approach to estimate the partition performance of the unit was developed, herein called the "Full Species Distribution" Method, which uses the full set of information available on how each of the 20 species is distributed in the three different streams as a function of the portion density.

Figure 12 justifies the assumption that the partition curve has full closure at both ends, so it is reasonable to assume that Equation (5) applies. Hence, only two parameters are required, $D_{50}$ and $Ep$. We cannot apply this partition equation directly to the measured feed density distribution because of the lack of resolution at the high densities. However, we can reconstruct the full feed density distribution by combining the product and reject density distributions based on a given mass yield to the product, $Y$. Let $y_{j,C}$ be the mass fraction of portion $j$ in the concentrate and $x_{i,j,C}$ be the assay of species $i$ in portion $j$ of the $j$ = 1–60 concentrate product flow × size portions, and similarly, let $y_{k,T}$ be the mass fraction of portion $k$ in the reject and $x_{i,k,T}$ be the assay of species $i$ in portion $k$ of the $k$ = 1–58 reject tailings flow × size portions. Then, the reconstituted feed consists of the yield $Y$ multiplied by the product of the mass fraction and assay of each of the 60 product portions combined with $(1 − Y)$ multiplied by each of the 58 reject portions, thus creating a reconstituted feed with 118 portions that cover the full range of density in the samples (the feed distributions of 10 of the species calculated in this manner is in fact already shown in Figure 11). Hence, the head +0.045 mm predicted (*) assay of species $i$ in this reconstituted feed is:

$$x_{i,F}^* = Y \sum_{j=1}^{60} y_{j,C} x_{i,j,C} + (1 - Y) \sum_{k=1}^{58} y_{k,T} x_{i,k,T} \tag{8}$$

We know the measured density $D$ of each of these 118 feed portions, and so for an assumed value of $D_{50}$ and $Ep$, we can use the partition function $P$, Equation (5), to predict the mass of that portion, which reports to the concentrate product and to the reject tailings streams. Multiplying that portion's mass by the concentration of species $i$ in that portion, summing over all portions and then dividing the sum by the total mass of each stream ($Y$ or $1 − Y$, respectively) gives the predicted +0.045 mm assay of species $i$ in the product concentrate (C) and reject tailings (T) streams:

$$x_{i,C}^* = \left[ Y \sum_{j=1}^{60} P_j y_{j,C} x_{i,j,C} + (1 - Y) \sum_{k=1}^{58} P_k y_{k,T} x_{i,k,T} \right] \frac{1}{Y} \tag{9}$$

$$x_{i,T}^* = \left[ Y \sum_{j=1}^{60} (1 - P_j) y_{j,C} x_{i,j,C} + (1 - Y) \sum_{k=1}^{58} (1 - P_k) y_{k,T} x_{i,k,T} \right] \frac{1}{1 - Y} \tag{10}$$

where $P_j$ is the partition probability calculated by Equation (5) for material in portion $j$ of known density $D_j$. This approach implicitly assumes that when material in a portion is partitioned, the compositions of the parts directed to the concentrate and tailings streams remain the same as in the feed stream. This is only accurate for portions with a very narrow range of particle densities, and so the reliability of this approach depends on how well the batch fractionation technique separates particles based only on their density. The narrowness of the density bands seen in Figure 7 suggests that this is a reasonable assumption.

So, given the values of just three parameters, $Y$, $D_{50}$ and $Ep$, the assays of all species in the three streams, $x_{i,F}{}^*$, $x_{i,C}{}^*$ and $x_{i,T}{}^*$, can be predicted based on knowledge of the densities and assays of the flow × size portions of the concentrate product and overflow tailing reject streams. The yield found from mass balancing the +0.045 assay data ($Y$ = 0.041) was then taken as fixed, and it was assumed that the best estimate of the true values of $D_{50}$ and $Ep$ will be those that minimise the sum of the square relative error (SSRE) between the mass-balanced (B) assays of the +0.045 mm wet-screened material for each stream (Table A1) and the assays of the reconstructed streams given by Equations (8)–(10):

$$SSRE = \sum_{i=1}^{20} \left( \frac{x_{i,F}^B - x_{i,F}^*}{x_{i,F}^B} \right)^2 + \sum_{i=1}^{20} \left( \frac{x_{i,C}^B - x_{i,C}^*}{x_{i,C}^B} \right)^2 + \sum_{i=1}^{20} \left( \frac{x_{i,T}^B - x_{i,T}^*}{x_{i,T}^B} \right)^2 \tag{11}$$

We note that taking $Y = 0.041$ as fixed means that the predicted $x_{i,\text{F}}{}^*$ values (Equation (8)) are fixed, and so the contribution of the feed term to the *SSRE* in Equation (11) is also fixed. The *SSRE* minimisation step was performed using the Solver function in MS Excel®, which found the minimum sum square error to be $SSRE_{\text{min}} = 1.29$ at $D_{50} = 3764$ kg/m³ and $Ep = 107$ kg/m³. Figure 13 shows the percentage differences between the assays found by the Full Species Distribution Method ($x_{i,\text{F}}{}^*, x_{i,\text{C}}{}^*, x_{i,\text{T}}{}^*$) and the balanced assay data ($x_{i,\text{F}}{}^{\text{B}}, x_{i,\text{C}}{}^{\text{B}}, x_{i,\text{T}}{}^{\text{B}}$) for each species in the feed, product and reject samples (Table A1). A total of 41 of the 60 values had less than a 10% relative difference, quite reasonable considering the trace levels of many of the assays. With these partition parameters applied to the reconstructed feed, the predicted output mass yield of the product was 4.0%, very close to the input value of 4.1%. Using this method, reconstructed feed, product and reject samples had average densities of 2890, 5768 and 2832 kg/m³, respectively, which are in reasonable agreement with the raw density measurements of 2810, 5485 and 2761 kg/m³ (Table 1).

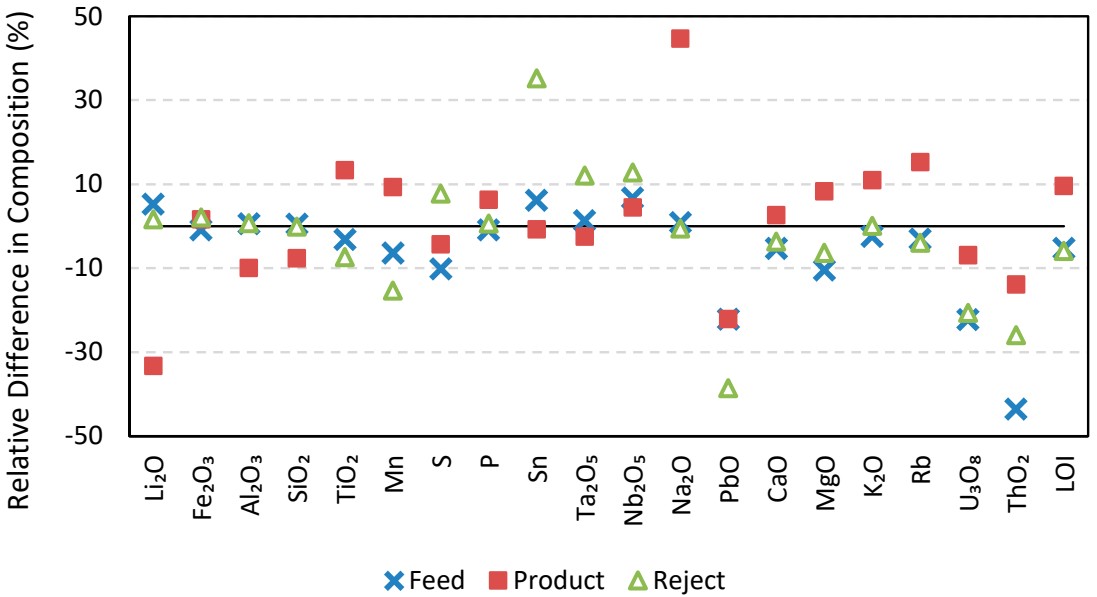

**Figure 13.** Relative difference between compositions fitted by the Full Species Distribution Method and the mass-balanced assay data for +0.045 mm wet-screened feed, product and reject samples (Table A1).

An *Ep* of 107 kg/m³ is quite low compared with published data for the RC™ system. It is noted that the addition of the narrow concentrator section at the base of the unit may have helped to improve its performance. However, another important factor to consider is the sensitivity of the result. The fitted $D_{50}$ value of 3764 kg/m³ lies at the lower end of a wide range of densities from 3400 to 4700 kg/m³ in which there is virtually no material (Figure 7). Given this absence of material, it is likely that a wide range of different partition curves would give very similar results. This is confirmed in Figure 14, which shows how the minimum *SSRE* varies when *Ep* is fixed, and only the $D_{50}$ is allowed to be varied. The $SSRE_{\text{min}}$ curve is very shallow with a wide range of *Ep* values from 250 kg/m³ down to 0 that arguably give equally good fits to the data (the range in which $SSRE_{\text{min}} < 2.0$).

The fitted $D_{50}$ value is strongly correlated with *Ep* (Figure 14). The range of results that give $SSRE_{\text{min}} < 2.0$ is shown in Figure 15, together with the optimum (global minimum *SSRE*) solution. The lower bound of plausible partition curves is anchored by the near-zero recovery of phosphorous (P) at $D \sim 3300$ kg/m³. Above that point, plausible curves can lie anywhere between a steeply rising (low *Ep*) partition curve with a low-density cut point (e.g., $Ep \sim 2$ kg/m³, $D_{50} \sim 3490$ kg/m³) or a slower rising (higher *Ep*) curve with a higher density cut point (e.g., $Ep \sim 250$ kg/m³, $D_{50} \sim 4380$ kg/m³).

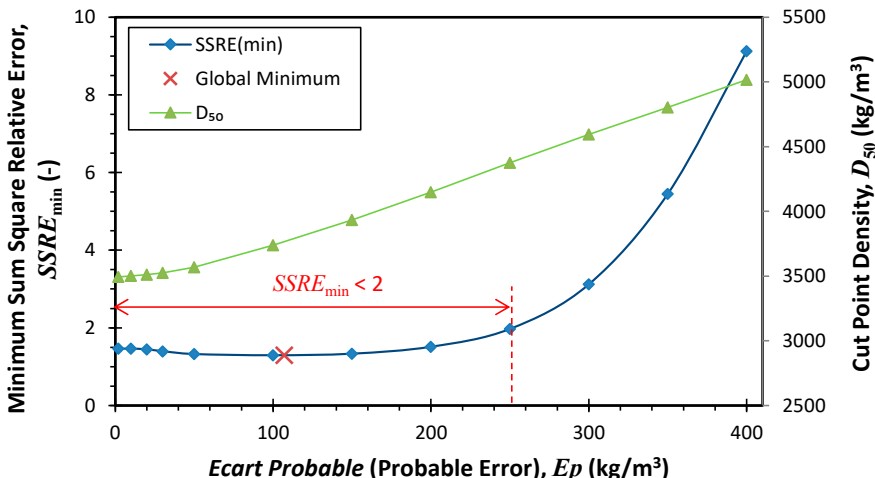

**Figure 14.** Results of the Full Species Distribution Method with $Y = 0.041$ when $Ep$ is fixed and $D_{50}$ is allowed to vary to minimise $SSRE$. The optimum $D_{50}$ is strongly correlated with $Ep$, and there is a wide range of $Ep$ values that give $SSRE_{min} < 2$.

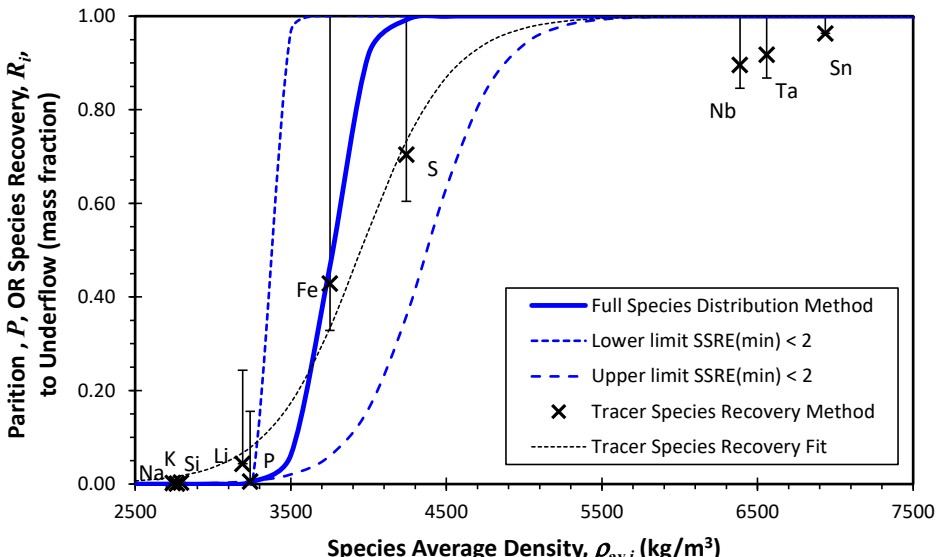

**Figure 15.** Partition curve of the wet-screened +0.045 mm material found by the Full Species Distribution Method with $Y = 0.041$. Dashed blue curves show range of results that give $SSRE_{min} < 2$ when $Ep$ is fixed and $D_{50}$ is allowed to vary. Also shown are the +0.045 mm recovery values plotted in Figure 12 and the best fit of Equation (5) through that data. Upper and lower error bars on those data show approximate fractions of the species found, respectively, below and above the narrow density range in which most of that species is found (see Figure 11).

This family of plausible curves is also in good agreement with the range of partition values suggested by the approximate Tracer Species Recovery method, which is also shown in Figure 15. These are the same head +0.045 mm species recovery data seen in Figure 12 with error bars added to indicate the fraction of the species found below and above the narrow density range in which most of that species is found (seen in Figure 11). The best fit of Equation (5) through this +0.045 mm species recovery data is $D_{50} = 3952 \text{ kg/m}^3$ and $Ep = 317 \text{ kg/m}^3$. However, we know that the overall recovery of a species is not an accurate indication of the true partition performance at that density because there were significant amounts of each of these tracer species found at other densities, and thus, this is likely to overestimate the $Ep$ value. Conversely, we also expect that there will be errors in the optimum Full Species Distribution Method fit of $D_{50} = 3764 \text{ kg/m}^3$ and $Ep = 107 \text{ kg/m}^3$ as it assumes the material in each portion is partitioned with equal concentrations into both the

product and reject streams; the accuracy of this assumption depends on how well the batch fractionation split the material into narrow density bands. Hence, the agreement between these two results is very pleasing, and the true partition curve likely lies somewhere in between.

The results of the two methods used in this study to estimate the partition performance suggest the $+0.045 -0.100$ mm $Ep$ value was somewhere in the range of 107–317 kg/m$^3$. A previous beneficiation study on a chromite ore (particle upper density ~4500 kg/m$^3$ and top size ~0.35 mm) in a standard RC™100 unit with 3 mm channels obtained $Ep$ values of the order of 300–400 kg/m$^3$ [15]. It is remarkable that the $Ep$ in this work appears to have been significantly sharper, especially given that the unit was operating at a higher cut-point density. However, the result is plausible given that this was a unit with a long, narrow 40 mm diameter concentrator section at the base, which is specially designed to minimise the entrainment of low-density material into the underflow product.

## 4. Conclusions

A laboratory-scale REFLUX™ Classifier was operated in continuous mode to beneficiate a nominal $-0.10$ mm tantalum ore with a head grade of 1.30 wt.% Sn, 0.56 wt.% Ta and 0.20 wt. % Nb. The yield to underflow product was 4.0 wt.%, and the product grade was 34.55% for Sn (26.7× upgrade), 13.30 wt.% for Ta (23.7× upgrade) and 4.36 wt.% for Nb (21.9× upgrade) at recoveries of 95.2% Sn, 88.3% Ta and 87.0% Nb. These results demonstrate the effectiveness of this technology for beneficiating high-density materials, even when the yield-to-product is quite low. This low yield is accommodated through the additional reduced diameter section in the lower vertical section.

The $+0.045$ mm samples of the feed, product and reject were then fractionated in a batch REFLUX™ Classifier unit using dense LST solution to form 10 flow fractions (plus a remains fraction). Each of these 11 flow fractions was then screened into seven size intervals, which were analysed by pycnometry and XRF. The analysis of these data revealed the majority of the material was concentrated into four relatively narrow density bands. A simple analysis based on the recovery of selected tracer elements showed that the partition curve had good closure at both ends and that the density cut point and $Ep$ both increased with decreasing particle size. For the composite $+0.045$ mm size range, the density cut point was estimated by this method to be around 3952 kg/m$^3$ with an $Ep$ of 317 kg/m$^3$, but it was expected that this approximate Tracer Species Recovery method would likely overestimate the $Ep$ value.

An alternative novel approach, here called the Full Species Distribution Method, was developed, which utilised the full composition and density distribution data of the feed, product and reject samples. This method gave a cut point of 3764 kg/m$^3$ and $Ep$ of 107 kg/m$^3$. However, sensitivity analysis indicated that due to the near total absence of any material in the density range of 3400–4700 kg/m$^3$, the true $Ep$ value could likely lie anywhere in a range from 0 to 250 kg/m$^3$.

**Author Contributions:** Experiments, methodology, N.B.; analysis, S.M.I. and K.P.G.; writing—original draft preparation, S.M.I. and K.P.G.; writing—review and editing, S.M.I., N.B. and K.P.G.; funding acquisition, N.B. and K.P.G. All authors have read and agreed to the published version of the manuscript.

**Funding:** The authors acknowledge the funding support from the Australian Research Council for grant number DP200102122 and other research support from FLSmidth.

**Data Availability Statement:** Data is contained within the article or Appendices A–C.

**Conflicts of Interest:** Kevin Galvin reports financial support was provided by the Australian Research Council. Kevin Galvin reports financial support and equipment were provided by FLSmidth. Kevin Galvin reports a relationship with FLSmidth that includes funding grants. Kevin Galvin has patents issued via his employer to the licensee and is a beneficiary of the University IP policy. Nicholas Boonzaier is an employee of FLSmidth, which is the licensee for the REFLUX™ Classifier technology.

## Appendix A. Mass-Balanced Assay Data

**Table A1.** Mass-balanced assays, yields, recoveries, upgrades and magnitude of the raw data relative adjustments for the head and 0.045 mm wet-screened samples. Each set of mass balancing is performed independently, with the objective function to minimise the sum square of the relative error between the predicted assay and the measured assay (Equation (6)). Bold highlights the three densest species of most commercial interest.

| Sample | $Li_2O$ (wt.%) | $Fe_2O_3$ (wt.%) | $Al_2O_3$ (wt.%) | $SiO_2$ (wt.%) | $TiO_2$ (wt.%) | Mn (wt.%) | S (wt.%) | P (wt.%) | Sn (wt.%) | $Ta_2O_5$ (wt.%) | $Nb_2O_5$ (wt.%) | $Na_2O$ (wt.%) | PbO (wt.%) | CaO (wt.%) | MgO (wt.%) | $K_2O$ (wt.%) | Rb (wt.%) | $U_3O_8$ (wt.%) | $ThO_2$ (wt.%) | LOI (wt.%) |
|---|---|---|---|---|---|---|---|---|---|---|---|---|---|---|---|---|---|---|---|---|
| Head assay mass balancing relative adjustment (%): | | | | | | | | | | | | | | | | | | | | |
| Feed | 0.4 | −0.7 | 0.3 | −0.2 | 1.0 | −3.3 | −4.0 | 0.1 | 5.7 | 3.1 | −0.2 | 0.6 | −10.4 | −1.3 | 1.5 | −0.7 | −2.2 | 12.3 | −2.2 | −1.9 |
| Product | 0.0 | 0.3 | 0.0 | 0.0 | −0.1 | 1.7 | 3.0 | 0.0 | −4.4 | −2.5 | 0.1 | 0.0 | 11.3 | 0.0 | 0.0 | 0.0 | 0.0 | −6.3 | 0.7 | 0.5 |
| Reject | −0.4 | 0.5 | −0.3 | 0.2 | −0.9 | 2.0 | 1.6 | −0.1 | −0.2 | −0.3 | 0.0 | −0.6 | 4.2 | 1.3 | −1.4 | 0.8 | 2.4 | −2.4 | 1.7 | 1.6 |
| Head assay mass balancing results: | | | | | | | | | | | | | | | | | | | | |
| Feed (wt.%) | 0.831 | 1.867 | 15.178 | 66.123 | 0.106 | 0.114 | 0.776 | 0.802 | **1.369** | 0.579 | **0.199** | 4.064 | 0.013 | 3.278 | 0.507 | 1.425 | 0.092 | 0.008 | 0.002 | 1.265 |
| Product (wt.%) | 0.255 | 16.442 | 1.750 | 4.280 | 0.324 | 1.349 | 12.621 | 0.222 | **33.025** | 12.974 | **4.365** | 0.160 | 0.238 | 1.200 | 0.140 | 0.141 | 0.006 | 0.153 | 0.014 | 7.818 |
| Reject (wt.%) | 0.855 | 1.266 | 15.731 | 68.671 | 0.097 | 0.063 | 0.288 | 0.826 | **0.065** | 0.068 | **0.027** | 4.225 | 0.004 | 3.364 | 0.523 | 1.478 | 0.096 | 0.002 | 0.001 | 0.995 |
| Yield (wt.%) | 4.0 | 4.0 | 4.0 | 4.0 | 4.0 | 4.0 | 4.0 | 4.0 | **4.0** | **4.0** | **4.0** | 4.0 | 4.0 | 4.0 | 4.0 | 4.0 | 4.0 | 4.0 | 4.0 | 4.0 |
| Recovery (%) | 1.2 | 34.9 | 0.5 | 0.3 | 12.1 | 46.8 | 64.4 | 1.1 | 95.5 | 88.7 | 86.9 | 0.2 | 70.2 | 1.4 | 1.1 | 0.4 | 0.3 | 73.7 | 28.3 | 24.5 |
| Upgrade (-) | 0.3 | 8.8 | 0.1 | 0.1 | 3.1 | 11.8 | 16.3 | 0.3 | 24.1 | 22.4 | 22.0 | 0.0 | 17.7 | 0.4 | 0.3 | 0.1 | 0.1 | 18.6 | 7.2 | 6.2 |
| +0.045 mm mass balancing relative adjustments (%): | | | | | | | | | | | | | | | | | | | | |
| Feed | 2.1 | −0.8 | −0.1 | 0.2 | 1.1 | −0.8 | −4.6 | −0.8 | 5.3 | 3.5 | 2.8 | 0.6 | 6.7 | −0.9 | −2.3 | −1.4 | 0.4 | −6.3 | −13.9 | −1.9 |
| Product | 0.0 | 0.4 | 0.0 | 0.0 | −0.1 | 0.4 | 3.7 | 0.0 | −4.2 | −2.9 | −2.2 | 0.0 | −3.7 | 0.0 | 0.0 | 0.0 | 0.0 | 6.4 | 4.8 | 0.6 |
| Reject | −1.9 | 0.5 | 0.1 | −0.2 | −0.9 | 0.4 | 1.7 | 0.9 | −0.2 | −0.3 | −0.3 | −0.6 | −1.8 | 0.9 | 2.5 | 1.4 | −0.4 | 1.6 | 20.9 | 1.4 |
| +0.045 mm mass balancing results: | | | | | | | | | | | | | | | | | | | | |
| Feed (wt.%) | 0.793 | 1.795 | 15.095 | 66.759 | 0.105 | 0.102 | 0.726 | 0.785 | **1.323** | 0.546 | **0.181** | 4.134 | 0.012 | 3.240 | 0.488 | 1.397 | 0.091 | 0.007 | 0.002 | 1.197 |
| Product (wt.%) | 0.327 | 18.224 | 2.110 | 5.280 | 0.338 | 1.210 | 11.966 | 0.266 | **30.829** | 12.130 | **3.927** | 0.190 | 0.193 | 1.350 | 0.170 | 0.153 | 0.006 | 0.140 | 0.012 | 8.702 |
| Reject (wt.%) | 0.813 | 1.085 | 15.656 | 69.416 | 0.095 | 0.054 | 0.240 | 0.808 | **0.048** | 0.045 | **0.019** | 4.305 | 0.004 | 3.321 | 0.502 | 1.450 | 0.095 | 0.002 | 0.002 | 0.872 |
| Yield (wt.%) | 4.1 | 4.1 | 4.1 | 4.1 | 4.1 | 4.1 | 4.1 | 4.1 | **4.1** | **4.1** | **4.1** | 4.1 | 4.1 | 4.1 | 4.1 | 4.1 | 4.1 | 4.1 | 4.1 | 4.1 |
| Recovery (%) | 1.7 | 42.1 | 0.6 | 0.3 | 13.3 | 49.1 | 68.3 | 1.4 | 96.5 | 92.1 | 90.0 | 0.2 | 67.9 | 1.7 | 1.4 | 0.5 | 0.3 | 78.8 | 23.5 | 30.1 |
| Upgrade (-) | 0.4 | 10.2 | 0.1 | 0.1 | 3.2 | 11.8 | 16.5 | 0.3 | 23.3 | 22.2 | 21.7 | 0.0 | 16.4 | 0.4 | 0.3 | 0.1 | 0.1 | 19.0 | 5.7 | 7.3 |
| −0.045 mm bass balancing relative adjustments (%): | | | | | | | | | | | | | | | | | | | | |
| Feed | 3.5 | −0.2 | 3.2 | −1.3 | 2.1 | −2.7 | −11.4 | 3.1 | 2.4 | −4.5 | −6.6 | −2.3 | 7.1 | 1.5 | 9.4 | 9.8 | 5.7 | 11.5 | 12.2 | 3.2 |
| Product | 0.0 | 0.0 | 0.0 | 0.0 | −0.2 | 1.1 | 2.1 | 0.0 | −1.7 | 3.5 | 5.1 | 0.0 | −2.5 | 0.0 | 0.0 | 0.0 | 0.0 | −4.8 | −2.6 | −0.1 |
| Reject | −3.0 | 0.2 | −2.8 | 1.4 | −1.8 | 1.8 | 17.2 | −2.7 | −0.5 | 1.7 | 3.1 | 2.5 | −3.3 | −1.4 | −6.8 | −7.0 | −4.6 | −3.7 | −6.1 | −2.7 |
| −0.045 mm mass balancing results: | | | | | | | | | | | | | | | | | | | | |
| Feed (wt.%) | 0.803 | 3.772 | 15.603 | 56.904 | 0.193 | 0.303 | 0.807 | 1.018 | **3.744** | 1.713 | **0.546** | 3.655 | 0.052 | 4.162 | 0.799 | 1.536 | 0.087 | 0.034 | 0.005 | 1.764 |
| Product (wt.%) | 0.014 | 4.631 | 0.310 | 1.210 | 0.260 | 1.966 | 1.848 | 0.023 | **49.813** | 18.904 | **5.613** | 0.050 | 0.375 | 0.470 | 0.030 | 0.080 | 0.003 | 0.325 | 0.024 | 1.358 |
| Reject (wt.%) | 0.853 | 3.717 | 16.577 | 60.454 | 0.189 | 0.197 | 0.741 | 1.082 | **0.808** | 0.617 | **0.223** | 3.885 | 0.032 | 4.397 | 0.848 | 1.629 | 0.092 | 0.016 | 0.004 | 1.790 |
| Yield (wt.%) | 6.0 | 6.0 | 6.0 | 6.0 | 6.0 | 6.0 | 6.0 | 6.0 | **6.0** | **6.0** | **6.0** | 6.0 | 6.0 | 6.0 | 6.0 | 6.0 | 6.0 | 6.0 | 6.0 | 6.0 |
| Recovery (%) | 0.1 | 7.4 | 0.1 | 0.1 | 8.1 | 38.9 | 13.7 | 0.1 | 79.7 | 66.1 | 61.6 | 0.1 | 42.8 | 0.7 | 0.2 | 0.3 | 0.2 | 56.8 | 28.4 | 4.6 |
| Upgrade (-) | 0.0 | 1.2 | 0.0 | 0.0 | 1.3 | 6.5 | 2.3 | 0.0 | 13.3 | 11.0 | 10.3 | 0.0 | 7.2 | 0.1 | 0.0 | 0.1 | 0.0 | 9.5 | 4.7 | 0.8 |

## Appendix B. Size × Flow Assay and Density Raw Data

**Table B1.** Assays, densities and mass fractions in each of the 11 flow fractions of the wet-screened +0.045 mm fraction of the three samples (feed, product and reject).

| Sample | Flow Fraction | Mass Fraction (wt.%) | Density (RD) | $Li_2O$ (wt.%) | $Fe_2O_3$ (wt.%) | $Al_2O_3$ (wt.%) | $SiO_2$ (wt.%) | $TiO_2$ (wt.%) | Mn (wt.%) | S (wt.%) | P (wt.%) | Sn (wt.%) | $Ta_2O_5$ (wt.%) | $Nb_2O_5$ (wt.%) | $Na_2O$ (wt.%) | PbO (wt.%) | CaO (wt.%) | MgO (wt.%) | $K_2O$ (wt.%) | Rb (wt.%) | $U_3O_8$ (wt.%) | $ThO_2$ (wt.%) | LOI (wt.%) |
|---|---|---|---|---|---|---|---|---|---|---|---|---|---|---|---|---|---|---|---|---|---|---|---|
| Feed | Flow 1 | 6.79 | 2.680 | 0.063 | 0.230 | 17.630 | 67.560 | 0.018 | 0.009 | 0.032 | 0.111 | 0.080 | 0.022 | 0.009 | 4.180 | 0.006 | 0.320 | 0.160 | 7.802 | 0.5326 | 0.0021 | 0.0004 | 0.61 |
| | Flow 2 | 7.59 | 2.701 | 0.028 | 0.160 | 16.970 | 71.510 | 0.015 | 0.006 | 0.005 | 0.063 | 0.017 | 0.005 | 0.003 | 7.820 | 0.002 | 0.360 | 0.130 | 2.264 | 0.1345 | 0.0007 | 0.0003 | 0.41 |
| | Flow 3 | 10.96 | 2.692 | 0.019 | 0.130 | 15.470 | 74.790 | 0.014 | 0.004 | 0.003 | 0.047 | 0.010 | 0.002 | 0.001 | 8.160 | 0.002 | 0.460 | 0.090 | 0.555 | 0.0195 | 0.0003 | 0.0002 | 0.30 |
| | Flow 4 | 8.59 | 2.679 | 0.028 | 0.140 | 14.290 | 76.630 | 0.017 | 0.004 | 0.002 | 0.041 | 0.008 | 0.002 | 0.001 | 7.260 | 0.002 | 0.580 | 0.090 | 0.565 | 0.0213 | 0.0004 | 0.0001 | 0.27 |
| | Flow 5 | 5.92 | 2.672 | 0.031 | 0.190 | 11.790 | 80.650 | 0.021 | 0.004 | 0.003 | 0.033 | 0.006 | 0.002 | 0.001 | 5.460 | 0.001 | 0.930 | 0.100 | 0.403 | 0.0112 | 0.0003 | 0.0001 | 0.32 |
| | Flow 6 | 10.94 | 2.703 | 0.038 | 0.280 | 11.020 | 81.800 | 0.028 | 0.007 | 0.019 | 0.033 | 0.048 | 0.018 | 0.006 | 4.430 | 0.001 | 1.360 | 0.130 | 0.426 | 0.0122 | 0.0004 | 0.0002 | 0.35 |
| | Flow 7 | 10.55 | 2.703 | 0.042 | 0.260 | 10.080 | 83.290 | 0.030 | 0.007 | 0.005 | 0.029 | 0.011 | 0.003 | 0.000 | 3.860 | 0.000 | 1.350 | 0.130 | 0.432 | 0.0128 | 0.0003 | 0.0002 | 0.37 |
| | Flow 8 | 9.59 | 2.733 | 0.085 | 0.500 | 11.590 | 80.220 | 0.055 | 0.011 | 0.008 | 0.041 | 0.024 | 0.006 | 0.003 | 3.180 | 0.001 | 2.030 | 0.270 | 0.835 | 0.0371 | 0.0003 | 0.0003 | 0.62 |
| | Flow 9 | 7.38 | 2.910 | 1.721 | 1.800 | 23.240 | 60.360 | 0.170 | 0.067 | 0.017 | 0.421 | 0.041 | 0.010 | 0.005 | 1.680 | 0.004 | 2.200 | 0.970 | 3.794 | 0.3138 | 0.0006 | 0.0006 | 2.13 |
| | Flow 10 | 11.91 | 3.207 | 3.901 | 3.860 | 22.690 | 44.980 | 0.309 | 0.156 | 1.210 | 3.191 | 0.106 | 0.136 | 0.050 | 0.680 | 0.004 | 10.030 | 1.490 | 0.912 | 0.0666 | 0.0043 | 0.0039 | 1.95 |
| | Flow 11 | 9.79 | 3.942 | 1.871 | 10.270 | 12.080 | 24.470 | 0.385 | 0.609 | 6.040 | 3.201 | 12.338 | 5.148 | 1.677 | 0.540 | 0.055 | 10.260 | 0.990 | 0.296 | 0.0167 | 0.0600 | 0.0090 | 3.72 |
| Underflow Product | Flow 1 | 10.17 | 3.550 | 2.915 | 11.630 | 16.260 | 35.400 | 0.495 | 0.214 | 9.428 | 2.272 | 0.365 | 0.484 | 0.206 | 0.750 | 0.037 | 7.230 | 1.120 | 0.397 | 0.0237 | 0.0307 | 0.0174 | 6.52 |
| | Flow 2 | 8.00 | 4.760 | 0.355 | 42.060 | 2.680 | 7.640 | 0.642 | 0.172 | 28.854 | 0.256 | 1.002 | 1.942 | 0.825 | 0.460 | 0.078 | 1.060 | 0.320 | 0.215 | 0.0070 | 0.0538 | 0.0124 | 23.19 |
| | Flow 3 | 8.99 | 4.940 | 0.143 | 46.330 | 1.470 | 4.440 | 0.490 | 0.297 | 28.815 | 0.094 | 2.245 | 4.017 | 1.789 | 0.370 | 0.093 | 0.780 | 0.230 | 0.207 | 0.0073 | 0.0702 | 0.0088 | 24.85 |
| | Flow 4 | 10.12 | 5.455 | 0.026 | 40.090 | 0.830 | 2.800 | 0.334 | 1.023 | 25.980 | 0.058 | 9.859 | 10.782 | 5.127 | 0.320 | 0.159 | 1.100 | 0.130 | 0.186 | 0.0059 | 0.1076 | 0.0094 | 19.92 |
| | Flow 5 | 9.62 | 5.983 | 0.016 | 26.560 | 0.610 | 1.910 | 0.302 | 1.726 | 17.190 | 0.038 | 23.412 | 15.884 | 7.086 | 0.260 | 0.189 | 1.070 | 0.070 | 0.168 | 0.0056 | 0.1139 | 0.0087 | 12.35 |
| | Flow 6 | 8.32 | 6.707 | 0.003 | 13.350 | 0.480 | 1.440 | 0.303 | 2.025 | 8.717 | 0.026 | 38.288 | 18.256 | 7.320 | 0.200 | 0.216 | 0.910 | 0.030 | 0.136 | 0.0041 | 0.1140 | 0.0081 | 5.70 |
| | Flow 7 | 7.11 | 7.143 | 0.009 | 8.220 | 0.430 | 1.270 | 0.304 | 2.019 | 4.808 | 0.023 | 44.877 | 18.359 | 6.813 | 0.180 | 0.229 | 0.810 | 0.020 | 0.121 | 0.0036 | 0.1166 | 0.0081 | 3.20 |
| | Flow 8 | 6.56 | 7.759 | 0.000 | 2.870 | 0.340 | 0.900 | 0.284 | 1.933 | 0.642 | 0.015 | 53.340 | 17.619 | 5.788 | 0.130 | 0.236 | 0.550 | 0.000 | 0.095 | 0.0026 | 0.1077 | 0.0073 | 0.57 |
| | Flow 9 | 14.45 | 7.340 | 0.014 | 3.400 | 0.450 | 1.410 | 0.271 | 1.661 | 1.088 | 0.024 | 55.020 | 15.614 | 4.508 | 0.150 | 0.220 | 0.440 | 0.010 | 0.088 | 0.0023 | 0.1074 | 0.0073 | 0.86 |
| | Flow 10 | 8.57 | 7.819 | 0.005 | 2.040 | 0.320 | 1.080 | 0.260 | 1.379 | 0.192 | 0.015 | 60.004 | 13.986 | 3.018 | 0.120 | 0.244 | 0.240 | 0.000 | 0.062 | 0.0021 | 0.1469 | 0.0097 | 0.29 |
| | Flow 11 | 8.09 | 7.646 | 0.015 | 2.340 | 0.290 | 0.890 | 0.259 | 1.361 | 0.443 | 0.017 | 59.653 | 14.337 | 2.605 | 0.100 | 0.290 | 0.220 | 0.000 | 0.057 | 0.0022 | 0.3956 | 0.0266 | 0.43 |
| Overflow Tailings Reject | Flow 1 | 9.80 | 2.650 | 0.047 | 0.160 | 17.390 | 68.780 | 0.013 | 0.008 | 0.008 | 0.096 | 0.012 | 0 | 0 | 5.440 | 0.005 | 0.310 | 0.110 | 6.236 | 0.4419 | 0.0005 | 0.0002 | 0.38 |
| | Flow 2 | 8.60 | 2.679 | 0.024 | 0.150 | 15.240 | 74.510 | 0.014 | 0.007 | 0.003 | 0.051 | 0.016 | 0.007 | 0.003 | 7.670 | 0.004 | 0.480 | 0.080 | 0.997 | 0.0530 | 0.0003 | 0.0001 | 0.24 |
| | Flow 3 | 9.63 | 2.701 | 0.032 | 0.180 | 13.760 | 77.290 | 0.023 | 0.005 | 0.003 | 0.044 | 0.014 | 0.004 | 0.000 | 6.690 | 0.003 | 0.710 | 0.100 | 0.778 | 0.0344 | 0.0002 | 0.0001 | 0.28 |
| | Flow 4 | 10.96 | 2.710 | 0.029 | 0.210 | 12.750 | 79.200 | 0.021 | 0.006 | 0.002 | 0.035 | 0.004 | 0.001 | 0 | 5.900 | 0.004 | 1.020 | 0.110 | 0.479 | 0.0162 | 0.0002 | 0.0001 | 0.20 |
| | Flow 5 | 5.49 | 2.705 | 0.028 | 0.230 | 11.760 | 80.340 | 0.027 | 0.005 | 0.002 | 0.033 | 0.004 | 0 | 0 | 5.120 | 0.003 | 1.170 | 0.120 | 0.443 | 0.0140 | 0.0001 | 0.0002 | 0.25 |
| | Flow 6 | 11.29 | 2.707 | 0.033 | 0.220 | 11.900 | 80.270 | 0.022 | 0.005 | 0.002 | 0.035 | 0.005 | 0.001 | 0 | 5.380 | 0.004 | 1.050 | 0.110 | 0.443 | 0.0141 | 0.0002 | 0.0001 | 0.26 |
| | Flow 7 | 9.79 | 2.708 | 0.058 | 0.350 | 11.610 | 80.880 | 0.040 | 0.009 | 0.004 | 0.038 | 0.008 | 0 | 0 | 4.100 | 0.003 | 1.590 | 0.180 | 0.634 | 0.0274 | 0.0002 | 0.0002 | 0.39 |
| | Flow 8 | 13.51 | 2.806 | 0.839 | 1.020 | 16.620 | 72.290 | 0.098 | 0.033 | 0.010 | 0.150 | 0.018 | 0.006 | 0.003 | 2.800 | 0.004 | 1.730 | 0.550 | 2.035 | 0.1617 | 0.0004 | 0.0004 | 1.10 |
| | Flow 9 | 4.57 | 3.129 | 4.314 | 2.110 | 26.260 | 53.960 | 0.194 | 0.097 | 0.020 | 1.242 | 0.038 | 0.007 | 0.003 | 0.740 | 0.002 | 4.090 | 1.290 | 1.798 | 0.1451 | 0.0007 | 0.0009 | 1.54 |
| | Flow 10 | 8.80 | 3.121 | 3.635 | 3.390 | 22.100 | 42.730 | 0.311 | 0.174 | 0.648 | 3.854 | 0.082 | 0.093 | 0.034 | 0.720 | 0.005 | 12.080 | 1.610 | 0.694 | 0.0530 | 0.0035 | 0.0042 | 1.61 |
| | Flow 11 | 7.55 | 3.306 | 2.901 | 5.890 | 17.810 | 36.930 | 0.401 | 0.230 | 2.455 | 4.706 | 0.671 | 0.569 | 0.205 | 0.830 | 0.012 | 14.680 | 1.400 | 0.403 | 0.0252 | 0.0094 | 0.0071 | 2.47 |

**Table B2.** Mass fraction, density and XRF assay data for each size interval of the 11 feed flow fractions. Size 1 = +0.106 mm, Size 2 = −0.106 + 0.090 mm, Size 3 = −0.090 + 0.075 mm, Size 4 = −0.075 + 0.063 mm, Size 5 = −0.063 + 0.053 mm, Size 6 = −0.053 + 0.045 mm, Size 7 = −0.045 mm. Where data are absent, this is because there was insufficient material in the size interval for the density and XRF assay analysis.

| Sample | Size Interval | Mass Fraction (wt.%) | Density (RD) | Li$_2$O (wt.%) | Fe$_2$O$_3$ (wt.%) | Al$_2$O$_3$ (wt.%) | SiO$_2$ (wt.%) | TiO$_2$ (wt.%) | Mn (wt.%) | S (wt.%) | P (wt.%) | Sn (wt.%) | Ta$_2$O$_5$ (wt.%) | Nb$_2$O$_5$ (wt.%) | Na$_2$O (wt.%) | PbO (wt.%) | CaO (wt.%) | MgO (wt.%) | K$_2$O (wt.%) | Rb (wt.%) | U$_3$O$_8$ (wt.%) | ThO$_2$ (wt.%) | LOI (wt.%) |
|---|---|---|---|---|---|---|---|---|---|---|---|---|---|---|---|---|---|---|---|---|---|---|---|
| Feed Flow 1 | 1 | 6.2 | | | | | | | | | | | | | | | | | | | | | |
| | 2 | 10.6 | 2.6359 | 0.05 | 0.23 | 17.67 | 67.11 | 0.028 | 0.009 | 0.022 | 0.113 | 0.037 | 0.009 | 0.003 | 3.95 | 0.004 | 0.21 | 0.13 | 8.58 | 0.572 | 0.0006 | 0.0001 | 0.84 |
| | 3 | 10.5 | 2.7095 | 0.04 | 0.21 | 17.63 | 67.78 | 0.002 | 0.008 | 0.019 | 0.11 | 0.035 | 0.008 | 0 | 4.05 | 0.004 | 0.24 | 0.14 | 8.23 | 0.582 | 0.0003 | 0.0001 | 0.62 |
| | 4 | 29.7 | 2.6323 | 0.04 | 0.21 | 17.53 | 67.82 | 0.004 | 0.008 | 0.024 | 0.106 | 0.05 | 0.017 | 0.004 | 4.23 | 0.002 | 0.31 | 0.15 | 7.81 | 0.535 | 0.0004 | 0.0001 | 0.60 |
| | 5 | 25.0 | 2.6507 | 0.06 | 0.23 | 17.74 | 67.90 | 0.007 | 0.01 | 0.033 | 0.108 | 0.082 | 0.03 | 0.009 | 4.33 | 0.004 | 0.37 | 0.16 | 7.50 | 0.511 | 0.0006 | 0.0002 | 0.63 |
| | 6 | 15.3 | 2.7150 | 0.06 | 0.30 | 17.74 | 67.53 | 0.017 | 0.014 | 0.04 | 0.11 | 0.133 | 0.052 | 0.016 | 4.46 | 0.004 | 0.42 | 0.18 | 7.22 | 0.486 | 0.0011 | 0.0002 | 0.83 |
| | 7 | 2.6 | | | | | | | | | | | | | | | | | | | | | |
| Feed Flow 2 | 1 | 1.6 | | | | | | | | | | | | | | | | | | | | | |
| | 2 | 3.9 | | | | | | | | | | | | | | | | | | | | | |
| | 3 | 6.1 | 2.6885 | 0.03 | 0.20 | 18.60 | 67.86 | 0.007 | 0.005 | 0.005 | 0.08 | 0.011 | 0.001 | 0 | 8.05 | 0 | 0.16 | 0.15 | 3.59 | 0.215 | 0.0003 | 0.0001 | 0.66 |
| | 4 | 29.7 | 2.6346 | 0.02 | 0.14 | 17.21 | 71.01 | 0.003 | 0.005 | 0.003 | 0.064 | 0.009 | 0.001 | 0 | 8.08 | 0 | 0.25 | 0.12 | 2.35 | 0.137 | 0.0002 | 0.0000 | 0.47 |
| | 5 | 31.3 | 2.6466 | 0.02 | 0.16 | 16.62 | 71.72 | 0.009 | 0.004 | 0.003 | 0.056 | 0.006 | 0.002 | 0 | 7.77 | 0 | 0.39 | 0.12 | 2.07 | 0.115 | 0.0002 | 0.0001 | 0.41 |
| | 6 | 23.3 | 2.6860 | 0.02 | 0.18 | 16.49 | 72.35 | 0.007 | 0.005 | 0.004 | 0.055 | 0.014 | 0.005 | 0 | 7.61 | 0 | 0.52 | 0.13 | 1.87 | 0.106 | 0.0002 | 0.0001 | 0.53 |
| | 7 | 4.1 | | | | | | | | | | | | | | | | | | | | | |
| Feed Flow 3 | 1 | 4.2 | | | | | | | | | | | | | | | | | | | | | |
| | 2 | 8.9 | 2.6783 | 0.02 | 0.10 | 17.16 | 71.75 | 0 | 0.003 | 0.002 | 0.061 | 0.002 | 0.002 | 0 | 9.57 | 0 | 0.13 | 0.07 | 0.59 | 0.020 | 0.0002 | 0.0001 | 0.36 |
| | 3 | 9.7 | 2.6842 | 0.02 | 0.10 | 16.20 | 73.40 | 0 | 0.003 | 0.002 | 0.055 | 0.002 | 0 | 0 | 8.86 | 0 | 0.19 | 0.08 | 0.55 | 0.019 | 0.0002 | 0.0001 | 0.28 |
| | 4 | 34.5 | 2.6677 | 0.02 | 0.12 | 15.23 | 75.11 | 0.005 | 0.003 | 0.002 | 0.046 | 0.002 | 0.002 | 0 | 8.11 | 0 | 0.41 | 0.08 | 0.52 | 0.017 | 0.0002 | 0.0001 | 0.25 |
| | 5 | 24.9 | 2.6724 | 0.02 | 0.14 | 15.07 | 75.27 | 0.007 | 0.003 | 0.002 | 0.042 | 0.004 | 0.001 | 0 | 7.76 | 0 | 0.60 | 0.09 | 0.56 | 0.018 | 0.0002 | 0.0001 | 0.31 |
| | 6 | 15.3 | 2.6821 | 0.02 | 0.18 | 15.04 | 75.31 | 0.036 | 0.005 | 0.003 | 0.041 | 0.007 | 0.004 | 0 | 7.46 | 0.001 | 0.75 | 0.10 | 0.60 | 0.021 | 0.0002 | 0.0001 | 0.29 |
| | 7 | 2.6 | | | | | | | | | | | | | | | | | | | | | |
| Feed Flow 4 | 1 | 3.6 | | | | | | | | | | | | | | | | | | | | | |
| | 2 | 7.2 | 2.7104 | 0.012 | 0.11 | 16.06 | 73.40 | 0.011 | 0.003 | 0.002 | 0.057 | 0.002 | 0.003 | 0 | 8.82 | 0 | 0.17 | 0.08 | 0.69 | 0.028 | 0.0002 | 0 | 0.43 |
| | 3 | 9.1 | 2.7293 | 0.017 | 0.11 | 15.04 | 75.22 | 0.003 | 0.004 | 0.002 | 0.049 | 0.002 | 0.002 | 0 | 8.05 | 0 | 0.26 | 0.08 | 0.609 | 0.025 | 0.0002 | 0.0001 | 0.44 |
| | 4 | 32.5 | 2.6806 | 0.014 | 0.13 | 14.14 | 76.83 | 0.005 | 0.003 | 0.003 | 0.043 | 0.001 | 0 | 0 | 7.31 | 0 | 0.50 | 0.08 | 0.536 | 0.019 | 0.0001 | 0.0001 | 0.26 |
| | 5 | 28.4 | 2.6713 | 0.014 | 0.15 | 13.88 | 77.26 | 0.008 | 0.003 | 0.002 | 0.038 | 0.003 | 0 | 0 | 6.89 | 0 | 0.71 | 0.09 | 0.539 | 0.018 | 0.0002 | 0.0001 | 0.24 |
| | 6 | 16.9 | 2.6932 | 0.024 | 0.22 | 13.87 | 77.27 | 0.014 | 0.004 | 0.003 | 0.035 | 0.004 | 0.001 | 0 | 6.53 | 0 | 0.91 | 0.11 | 0.571 | 0.018 | 0.0002 | 0.0002 | 0.34 |
| | 7 | 2.4 | | | | | | | | | | | | | | | | | | | | | |
| Feed Flow 5 | 1 | 3.9 | | | | | | | | | | | | | | | | | | | | | |
| | 2 | 7.8 | | | | | | | | | | | | | | | | | | | | | |
| | 3 | 9.0 | 2.7654 | 0.022 | 0.16 | 11.53 | 80.85 | 0.009 | 0.003 | 0.003 | 0.039 | 0.003 | 0.002 | 0.000 | 5.93 | 0 | 0.40 | 0.09 | 0.41 | 0.012 | 0.0002 | 0.0001 | 0.31 |
| | 4 | 34.4 | 2.7050 | 0.029 | 0.19 | 11.38 | 81.38 | 0.013 | 0.003 | 0.003 | 0.032 | 0.001 | 0.002 | 0.000 | 5.28 | 0 | 0.92 | 0.10 | 0.39 | 0.010 | 0.0002 | 0.0002 | 0.24 |
| | 5 | 28.5 | 2.7099 | 0.017 | 0.20 | 11.94 | 80.23 | 0.014 | 0.004 | 0.002 | 0.031 | 0.004 | 0.002 | 0.000 | 5.34 | 0 | 1.14 | 0.11 | 0.41 | 0.010 | 0.0002 | 0.0002 | 0.31 |
| | 6 | 14.9 | 2.7821 | 0.023 | 0.23 | 12.61 | 79.25 | 0.016 | 0.005 | 0.003 | 0.03 | 0.006 | 0.003 | 0.000 | 5.43 | 0 | 1.31 | 0.11 | 0.45 | 0.011 | 0.0002 | 0.0002 | 0.38 |
| | 7 | 1.5 | | | | | | | | | | | | | | | | | | | | | |
| Feed Flow 6 | 1 | 4.9 | | | | | | | | | | | | | | | | | | | | | |
| | 2 | 8.6 | 2.7086 | 0.025 | 0.24 | 9.98 | 83.17 | 0.013 | 0.004 | 0.024 | 0.039 | 0.019 | 0.008 | 0.003 | 4.88 | 0 | 0.49 | 0.11 | 0.39 | 0.011 | 0.0003 | 0.0001 | 0.29 |
| | 3 | 9.0 | 2.7644 | 0.033 | 0.27 | 10.2 | 83.07 | 0.019 | 0.006 | 0.016 | 0.038 | 0.028 | 0.009 | 0.000 | 4.47 | 0 | 0.87 | 0.13 | 0.43 | 0.012 | 0.0002 | 0.0002 | 0.27 |
| | 4 | 38.7 | 2.7065 | 0.027 | 0.27 | 10.72 | 82.18 | 0.022 | 0.006 | 0.014 | 0.031 | 0.028 | 0.013 | 0.003 | 4.28 | 0 | 1.38 | 0.12 | 0.41 | 0.011 | 0.0002 | 0.0002 | 0.31 |
| | 5 | 24.9 | 2.7245 | 0.026 | 0.29 | 11.54 | 80.73 | 0.023 | 0.007 | 0.025 | 0.03 | 0.062 | 0.023 | 0.005 | 4.35 | 0 | 1.69 | 0.13 | 0.44 | 0.012 | 0.0004 | 0.0002 | 0.37 |
| | 6 | 12.5 | 2.7192 | 0.035 | 0.35 | 12.37 | 79.38 | 0.026 | 0.01 | 0.035 | 0.033 | 0.13 | 0.054 | 0.013 | 4.52 | 0.001 | 1.91 | 0.15 | 0.50 | 0.015 | 0.0008 | 0.0003 | 0.41 |
| | 7 | 1.5 | | | | | | | | | | | | | | | | | | | | | |

**Table B2.** *Cont.*

| Sample | Size Interval | Mass Fraction (wt.%) | Density (RD) | Li$_2$O (wt.%) | Fe$_2$O$_3$ (wt.%) | Al$_2$O$_3$ (wt.%) | SiO$_2$ (wt.%) | TiO$_2$ (wt.%) | Mn (wt.%) | S (wt.%) | P (wt.%) | Sn (wt.%) | Ta$_2$O$_5$ (wt.%) | Nb$_2$O$_5$ (wt.%) | Na$_2$O (wt.%) | PbO (wt.%) | CaO (wt.%) | MgO (wt.%) | K$_2$O (wt.%) | Rb (wt.%) | U$_3$O$_8$ (wt.%) | ThO$_2$ (wt.%) | LOI (wt.%) |
|---|---|---|---|---|---|---|---|---|---|---|---|---|---|---|---|---|---|---|---|---|---|---|---|
| Feed Flow 7 | 1 | 7.5 | | | | | | | | | | | | | | | | | | | | | |
| | 2 | 10.8 | 2.7515 | 0.033 | 0.21 | 9.31 | 84.6 | 0.015 | 0.004 | 0.006 | 0.035 | 0.004 | 0.003 | 0.000 | 4.25 | 0 | 0.55 | 0.12 | 0.42 | 0.012 | 0.0002 | 0.0002 | 0.30 |
| | 3 | 8.7 | 2.7593 | 0.035 | 0.24 | 9.05 | 84.94 | 0.018 | 0.005 | 0.005 | 0.032 | 0.006 | 0.004 | 0.001 | 3.80 | 0 | 0.82 | 0.13 | 0.44 | 0.013 | 0.0002 | 0.0002 | 0.33 |
| | 4 | 42.2 | 2.7172 | 0.026 | 0.26 | 9.72 | 84.06 | 0.021 | 0.006 | 0.004 | 0.026 | 0.005 | 0.004 | 0.000 | 3.58 | 0 | 1.44 | 0.13 | 0.41 | 0.011 | 0.0001 | 0.0002 | 0.24 |
| | 5 | 18.7 | 2.7388 | 0.03 | 0.28 | 10.82 | 82.08 | 0.023 | 0.005 | 0.006 | 0.026 | 0.01 | 0.003 | 0.000 | 3.77 | 0.002 | 1.86 | 0.13 | 0.43 | 0.012 | 0.0001 | 0.0003 | 0.28 |
| | 6 | 10.7 | 2.7183 | 0.04 | 0.34 | 11.93 | 79.99 | 0.028 | 0.006 | 0.01 | 0.03 | 0.013 | 0.008 | 0.002 | 4.03 | 0.002 | 2.13 | 0.15 | 0.52 | 0.016 | 0.0002 | 0.0003 | 0.41 |
| | 7 | 1.5 | | | | | | | | | | | | | | | | | | | | | |
| Feed Flow 8 | 1 | 8.7 | | | | | | | | | | | | | | | | | | | | | |
| | 2 | 10.6 | 2.7720 | 0.068 | 0.52 | 8.43 | 85.50 | 0.045 | 0.01 | 0.007 | 0.046 | 0.011 | 0.005 | 0.002 | 2.35 | 0 | 1.14 | 0.29 | 0.777 | 0.032 | 0.0002 | 0.0003 | 0.58 |
| | 3 | 10.0 | 2.7867 | 0.086 | 0.53 | 9.83 | 83.43 | 0.048 | 0.01 | 0.008 | 0.046 | 0.012 | 0.006 | 0.000 | 2.6 | 0 | 1.51 | 0.30 | 0.877 | 0.038 | 0.0005 | 0.0004 | 0.62 |
| | 4 | 40.3 | 2.7049 | 0.073 | 0.47 | 11.93 | 80.15 | 0.045 | 0.01 | 0.006 | 0.037 | 0.017 | 0.007 | 0.002 | 3.29 | 0 | 2.24 | 0.24 | 0.772 | 0.033 | 0.0003 | 0.0003 | 0.49 |
| | 5 | 19.9 | 2.7297 | 0.061 | 0.47 | 13.49 | 77.54 | 0.046 | 0.011 | 0.010 | 0.035 | 0.028 | 0.011 | 0.004 | 3.76 | 0.002 | 2.64 | 0.24 | 0.801 | 0.034 | 0.0003 | 0.0003 | 0.58 |
| | 6 | 9.4 | 2.7839 | 0.106 | 0.61 | 15.14 | 74.42 | 0.055 | 0.015 | 0.016 | 0.041 | 0.06 | 0.022 | 0.007 | 3.95 | 0.001 | 2.85 | 0.31 | 1.070 | 0.053 | 0.0005 | 0.0004 | 0.74 |
| | 7 | 1.1 | | | | | | | | | | | | | | | | | | | | | |
| Feed Flow 9 | 1 | 10.9 | | | | | | | | | | | | | | | | | | | | | |
| | 2 | 10.0 | | | | | | | | | | | | | | | | | | | | | |
| | 3 | 9.8 | 2.8624 | 0.818 | 1.57 | 22.74 | 62.65 | 0.123 | 0.052 | 0.016 | 0.17 | 0.041 | 0.015 | 0.006 | 1.45 | 0.001 | 1.12 | 0.70 | 4.885 | 0.438 | 0.0005 | 0.0005 | 2.67 |
| | 4 | 28.2 | 2.9233 | 1.664 | 1.81 | 23.82 | 60.1 | 0.174 | 0.064 | 0.015 | 0.319 | 0.038 | 0.01 | 0.003 | 1.77 | 0.001 | 1.99 | 0.97 | 3.937 | 0.315 | 0.0005 | 0.0005 | 2.15 |
| | 5 | 22.0 | 2.9878 | 2.211 | 1.97 | 24.05 | 58.31 | 0.189 | 0.072 | 0.016 | 0.584 | 0.038 | 0.009 | 0.002 | 1.74 | 0 | 2.82 | 1.11 | 3.322 | 0.256 | 0.0006 | 0.0006 | 2.00 |
| | 6 | 15.6 | 3.0662 | 3.286 | 2.13 | 24.87 | 56 | 0.197 | 0.088 | 0.022 | 0.918 | 0.04 | 0.013 | 0.004 | 1.26 | 0.002 | 3.55 | 1.22 | 2.740 | 0.199 | 0.0007 | 0.0009 | 1.96 |
| | 7 | 3.6 | | | | | | | | | | | | | | | | | | | | | |
| Feed Flow 10 | 1 | 9.3 | 3.2228 | 3.890 | 4.08 | 25.11 | 50.51 | 0.190 | 0.137 | 0.988 | 1.456 | 0.063 | 0.069 | 0.034 | 0.67 | 0.002 | 4.65 | 1.10 | 1.744 | 0.157 | 0.0041 | 0.0027 | 2.19 |
| | 2 | 11.5 | 3.1856 | 3.933 | 2.59 | 25.13 | 50.14 | 0.211 | 0.143 | 0.675 | 2.108 | 0.068 | 0.066 | 0.034 | 0.70 | 0.002 | 6.51 | 1.32 | 1.363 | 0.117 | 0.0031 | 0.0028 | 1.88 |
| | 3 | 9.8 | 3.2415 | 3.935 | 2.85 | 24.43 | 49.06 | 0.241 | 0.144 | 0.813 | 2.380 | 0.066 | 0.077 | 0.035 | 0.69 | 0.004 | 7.43 | 1.39 | 1.113 | 0.089 | 0.0036 | 0.0031 | 1.93 |
| | 4 | 25.1 | 3.1956 | 3.769 | 3.56 | 22.53 | 44.87 | 0.314 | 0.161 | 1.016 | 3.433 | 0.074 | 0.093 | 0.048 | 0.78 | 0.004 | 10.61 | 1.56 | 0.800 | 0.054 | 0.0035 | 0.0034 | 1.91 |
| | 5 | 25.3 | 3.2399 | 3.500 | 3.95 | 21.59 | 42.77 | 0.351 | 0.165 | 1.298 | 3.971 | 0.093 | 0.123 | 0.059 | 0.72 | 0.005 | 12.26 | 1.61 | 0.638 | 0.039 | 0.0038 | 0.0039 | 2.06 |
| | 6 | 16.0 | 3.2231 | 3.613 | 4.27 | 21.9 | 43.75 | 0.343 | 0.159 | 1.571 | 3.617 | 0.206 | 0.237 | 0.111 | 0.70 | 0.007 | 11.10 | 1.61 | 0.549 | 0.028 | 0.0058 | 0.0048 | 2.27 |
| | 7 | 3.0 | | | | | | | | | | | | | | | | | | | | | |
| Feed Flow 11 | 1 | 11.1 | 3.7260 | 2.765 | 15.71 | 16.47 | 35.46 | 0.277 | 0.324 | 6.186 | 1.627 | 5.228 | 2.116 | 0.726 | 0.57 | 0.040 | 5.23 | 0.90 | 0.597 | 0.042 | 0.0388 | 0.0062 | 5.23 |
| | 2 | 8.9 | 3.6898 | 2.669 | 8.58 | 17.11 | 35.18 | 0.315 | 0.341 | 5.609 | 3.021 | 7.026 | 2.209 | 0.719 | 0.69 | 0.038 | 9.39 | 1.21 | 0.422 | 0.025 | 0.0325 | 0.0070 | 2.93 |
| | 3 | 8.6 | 3.7696 | 2.547 | 9.02 | 16.22 | 33.29 | 0.329 | 0.351 | 6.018 | 3.244 | 7.412 | 2.364 | 0.802 | 0.67 | 0.040 | 10.08 | 1.20 | 0.350 | 0.018 | 0.0343 | 0.0073 | 3.47 |
| | 4 | 24.2 | 3.8051 | 1.789 | 10.04 | 12.46 | 25.29 | 0.426 | 0.599 | 6.377 | 4.026 | 11.06 | 4.438 | 1.589 | 0.62 | 0.046 | 12.45 | 1.10 | 0.243 | 0.011 | 0.0464 | 0.0082 | 4.01 |
| | 5 | 25.7 | 4.0382 | 1.46 | 10.04 | 10.38 | 20.97 | 0.454 | 0.76 | 6.344 | 3.785 | 15.19 | 6.036 | 2.126 | 0.57 | 0.061 | 11.80 | 0.98 | 0.203 | 0.009 | 0.0686 | 0.0097 | 4.59 |
| | 6 | 17.5 | 3.3299 | 1.282 | 9.62 | 9.59 | 19.23 | 0.46 | 0.941 | 5.811 | 2.925 | 19.35 | 7.793 | 2.708 | 0.51 | 0.102 | 9.30 | 0.92 | 0.179 | 0.007 | 0.1103 | 0.0121 | 4.24 |
| | 7 | 4.1 | | | | | | | | | | | | | | | | | | | | | |

**Table B3.** Mass fraction, density and XRF assay data for each size interval of the 11 underflow product flow fractions. Size 1 = +0.106 mm, Size 2 = −0.106 + 0.090 mm, Size 3 = −0.090 + 0.075 mm, Size 4 = −0.075 + 0.063 mm, Size 5 = −0.063 + 0.053 mm, Size 6 = −0.053 + 0.045 mm, Size 7 = −0.045 mm. Where data are absent, this is because there was insufficient material in the size interval for the density and XRF assay analysis.

| Sample | Size Interval | Mass Fraction (wt.%) | Density (RD) | Li₂O (wt.%) | Fe₂O₃ (wt.%) | Al₂O₃ (wt.%) | SiO₂ (wt.%) | TiO₂ (wt.%) | Mn (wt.%) | S (wt.%) | P (wt.%) | Sn (wt.%) | Ta₂O₅ (wt.%) | Nb₂O₅ (wt.%) | Na₂O (wt.%) | PbO (wt.%) | CaO (wt.%) | MgO (wt.%) | K₂O (wt.%) | Rb (wt.%) | U₃O₈ (wt.%) | ThO₂ (wt.%) | LOI (wt.%) |
|---|---|---|---|---|---|---|---|---|---|---|---|---|---|---|---|---|---|---|---|---|---|---|---|
| Product Flow 1 | 1 | 62.0 | 3.2535 | 3.956 | 3.63 | 21.91 | 45.85 | 0.24 | 0.239 | 2.419 | 2.639 | 0.100 | 0.168 | 0.083 | 0.73 | 0.015 | 8.34 | 1.36 | 0.437 | 0.026 | 0.0193 | 0.0147 | 2.52 |
|  | 2 | 7.9 | 3.8356 | 1.014 | 9.82 | 9.79 | 19.44 | 0.75 | 0.313 | 14.129 | 3.764 | 0.444 | 0.602 | 0.267 | 1.15 | 0.047 | 11.61 | 1.11 | 0.339 | 0.017 | 0.0477 | 0.0371 | 8.28 |
|  | 3 | 5.1 | 4.1395 | 0.855 | 12.49 | 8.07 | 16.92 | 0.96 | 0.251 | 17.967 | 2.365 | 0.537 | 0.695 | 0.299 | 1.26 | 0.052 | 7.31 | 0.90 | 0.375 | 0.020 | 0.0422 | 0.0271 | 10.96 |
|  | 4 | 6.9 | 4.2311 | 0.736 | 25.08 | 6.94 | 17.89 | 1.56 | 0.187 | 22.725 | 1.757 | 0.760 | 0.990 | 0.413 | 0.97 | 0.063 | 5.48 | 0.74 | 0.381 | 0.019 | 0.0500 | 0.0192 | 14.37 |
|  | 5 | 10.5 | 4.3683 | 0.499 | 38.50 | 4.93 | 14.77 | 1.15 | 0.133 | 24.345 | 1.137 | 0.959 | 1.268 | 0.495 | 0.61 | 0.070 | 3.62 | 0.56 | 0.300 | 0.014 | 0.0469 | 0.0130 | 19.34 |
|  | 6 | 6.7 | 4.6285 | 0.538 | 38.75 | 5.06 | 15.08 | 0.67 | 0.170 | 27.394 | 0.976 | 1.509 | 1.813 | 0.767 | 0.63 | 0.120 | 3.21 | 0.54 | 0.275 | 0.012 | 0.0540 | 0.0121 | 20.03 |
|  | 7 | 0.9 |  |  |  |  |  |  |  |  |  |  |  |  |  |  |  |  |  |  |  |  |  |
| Product Flow 2 | 1 | 20.2 | 3.7710 | 1.689 | 12.28 | 9.98 | 21.61 | 0.46 | 0.263 | 18.536 | 0.942 | 0.396 | 0.859 | 0.371 | 1.20 | 0.048 | 3.13 | 0.77 | 0.319 | 0.012 | 0.0479 | 0.0207 | 11.04 |
|  | 2 | 7.2 | 4.9325 | 0.049 | 25.53 | 1.82 | 5.14 | 1.39 | 0.242 | 23.445 | 0.251 | 0.939 | 1.792 | 0.731 | 1.19 | 0.082 | 1.00 | 0.39 | 0.341 | 0.010 | 0.0596 | 0.0177 | 19.23 |
|  | 3 | 6.7 | 5.0438 | 0.039 | 33.74 | 1.64 | 5.03 | 1.52 | 0.186 | 30.135 | 0.157 | 1.039 | 2.053 | 0.843 | 0.88 | 0.088 | 0.76 | 0.39 | 0.341 | 0.010 | 0.0626 | 0.0140 | 23.17 |
|  | 4 | 19.5 | 4.7838 | 0.026 | 51.53 | 1.15 | 4.65 | 0.95 | 0.109 | 30.807 | 0.098 | 0.821 | 1.729 | 0.719 | 0.32 | 0.075 | 0.53 | 0.31 | 0.226 | 0.007 | 0.0455 | 0.0086 | 27.88 |
|  | 5 | 30.8 | 4.7349 | 0.017 | 55.27 | 0.81 | 3.72 | 0.44 | 0.106 | 29.351 | 0.073 | 0.869 | 1.800 | 0.759 | 0.24 | 0.070 | 0.45 | 0.24 | 0.150 | 0.004 | 0.0399 | 0.0068 | 29.37 |
|  | 6 | 14.7 | 5.0444 | 0.033 | 52.19 | 0.80 | 3.73 | 0.33 | 0.289 | 31.056 | 0.079 | 2.355 | 3.775 | 1.741 | 0.27 | 0.146 | 0.67 | 0.21 | 0.139 | 0.003 | 0.0608 | 0.0082 | 27.37 |
|  | 7 | 1.0 |  |  |  |  |  |  |  |  |  |  |  |  |  |  |  |  |  |  |  |  |  |
| Product Flow 3 | 1 | 20.2 | 4.2017 | 0.531 | 28.80 | 4.09 | 9.42 | 0.54 | 0.242 | 27.071 | 0.219 | 0.820 | 1.968 | 0.847 | 0.89 | 0.057 | 0.96 | 0.51 | 0.336 | 0.011 | 0.0603 | 0.0095 | 20.18 |
|  | 2 | 8.7 | 4.8284 | 0.027 | 45.08 | 1.53 | 4.35 | 1.08 | 0.175 | 31.258 | 0.117 | 1.352 | 2.842 | 1.141 | 0.48 | 0.084 | 0.74 | 0.38 | 0.330 | 0.010 | 0.0649 | 0.0091 | 26.42 |
|  | 3 | 8.1 | 4.9901 | 0.026 | 49.96 | 1.27 | 3.84 | 0.86 | 0.140 | 31.875 | 0.090 | 1.199 | 2.637 | 1.074 | 0.36 | 0.077 | 0.63 | 0.33 | 0.273 | 0.008 | 0.0556 | 0.0078 | 28.67 |
|  | 4 | 24.6 | 4.8576 | 0.007 | 55.79 | 0.81 | 2.76 | 0.42 | 0.143 | 30.678 | 0.055 | 1.205 | 2.593 | 1.126 | 0.25 | 0.065 | 0.53 | 0.24 | 0.158 | 0.005 | 0.0417 | 0.0054 | 29.74 |
|  | 5 | 25.8 | 4.9343 | 0.008 | 53.70 | 0.65 | 2.17 | 0.28 | 0.295 | 30.527 | 0.048 | 2.493 | 4.649 | 2.079 | 0.25 | 0.089 | 0.73 | 0.20 | 0.135 | 0.004 | 0.0588 | 0.0066 | 28.57 |
|  | 6 | 11.4 | 5.3193 | 0.020 | 42.56 | 0.76 | 2.23 | 0.32 | 0.944 | 27.576 | 0.056 | 7.880 | 10.771 | 5.121 | 0.31 | 0.266 | 1.25 | 0.17 | 0.169 | 0.006 | 0.1191 | 0.0114 | 21.94 |
|  | 7 | 1.2 |  |  |  |  |  |  |  |  |  |  |  |  |  |  |  |  |  |  |  |  |  |
| Product Flow 4 | 1 | 7.4 | 4.8586 | 0.202 | 44.87 | 2.42 | 6.05 | 0.47 | 0.215 | 31.155 | 0.162 | 1.105 | 2.749 | 1.166 | 0.46 | 0.061 | 0.89 | 0.43 | 0.327 | 0.012 | 0.0649 | 0.0075 | 26.17 |
|  | 2 | 7.1 | 4.8586 | 0.028 | 53.73 | 1.27 | 3.38 | 0.51 | 0.143 | 31.811 | 0.094 | 1.392 | 3.066 | 1.234 | 0.32 | 0.071 | 0.71 | 0.32 | 0.261 | 0.009 | 0.0539 | 0.0058 | 30.12 |
|  | 3 | 9.4 | 5.0705 | 0.025 | 54.82 | 0.97 | 2.70 | 0.38 | 0.140 | 31.632 | 0.067 | 1.347 | 3.093 | 1.258 | 0.30 | 0.072 | 0.64 | 0.27 | 0.204 | 0.007 | 0.0566 | 0.0061 | 30.72 |
|  | 4 | 26.7 | 4.8267 | 0.012 | 54.23 | 0.69 | 1.93 | 0.26 | 0.333 | 31.380 | 0.046 | 2.480 | 4.878 | 2.233 | 0.26 | 0.083 | 0.74 | 0.21 | 0.150 | 0.005 | 0.0606 | 0.0060 | 28.51 |
|  | 5 | 26.7 | 4.9309 | 0.020 | 38.86 | 0.72 | 1.49 | 0.32 | 1.285 | 25.297 | 0.048 | 10.221 | 13.775 | 6.642 | 0.30 | 0.161 | 1.45 | 0.15 | 0.183 | 0.007 | 0.1308 | 0.0113 | 19.30 |
|  | 6 | 20.0 | 5.2868 | 0.019 | 14.78 | 0.73 | 1.12 | 0.39 | 2.654 | 9.721 | 0.043 | 27.702 | 23.181 | 11.045 | 0.27 | 0.360 | 1.64 | 0.07 | 0.188 | 0.006 | 0.1890 | 0.0149 | 6.42 |
|  | 7 | 2.7 | 6.3402 |  |  |  |  |  |  |  |  |  |  |  |  |  |  |  |  |  |  |  |  |
| Product Flow 5 | 1 | 11.1 | 5.0269 | 0.029 | 54.01 | 1.07 | 2.50 | 0.23 | 0.239 | 32.790 | 0.075 | 1.715 | 4.036 | 1.761 | 0.30 | 0.060 | 0.78 | 0.28 | 0.231 | 0.007 | 0.0672 | 0.0053 | 30.03 |
|  | 2 | 8.8 | 5.0102 | 0.026 | 55.17 | 0.78 | 1.93 | 0.22 | 0.203 | 31.025 | 0.055 | 2.086 | 4.169 | 1.686 | 0.26 | 0.078 | 0.74 | 0.23 | 0.190 | 0.006 | 0.0554 | 0.0046 | 30.21 |
|  | 3 | 9.2 | 4.9215 | 0.017 | 53.11 | 0.71 | 1.80 | 0.22 | 0.273 | 31.863 | 0.049 | 2.711 | 5.178 | 2.133 | 0.27 | 0.088 | 0.84 | 0.21 | 0.179 | 0.006 | 0.0677 | 0.0057 | 29.71 |
|  | 4 | 9.8 | 5.2913 | 0.014 | 43.17 | 0.70 | 1.54 | 0.28 | 0.993 | 27.905 | 0.045 | 7.464 | 11.350 | 5.447 | 0.29 | 0.137 | 1.28 | 0.17 | 0.186 | 0.006 | 0.1035 | 0.0085 | 22.92 |
|  | 5 | 29.0 | 6.2765 | 0.024 | 14.80 | 0.66 | 0.94 | 0.39 | 2.689 | 9.609 | 0.037 | 28.115 | 23.301 | 11.002 | 0.25 | 0.208 | 1.57 | 0.06 | 0.189 | 0.007 | 0.1549 | 0.0118 | 6.42 |
|  | 6 | 28.5 | 6.9467 | 0.006 | 4.62 | 0.42 | 0.94 | 0.34 | 2.664 | 1.727 | 0.022 | 44.651 | 21.629 | 8.785 | 0.15 | 0.268 | 0.90 | 0.01 | 0.118 | 0.004 | 0.1381 | 0.0097 | 1.28 |
|  | 7 | 3.5 |  |  |  |  |  |  |  |  |  |  |  |  |  |  |  |  |  |  |  |  |  |
| Product Flow 6 | 1 | 6.7 | 5.5098 | 0.032 | 49.25 | 0.93 | 1.98 | 0.21 | 0.487 | 33.108 | 0.058 | 4.089 | 7.869 | 3.377 | 0.33 | 0.094 | 1.20 | 0.22 | 0.230 | 0.009 | 0.1202 | 0.0090 | 27.29 |
|  | 2 | 5.4 | 6.9456 | 0.019 | 48.05 | 0.77 | 1.67 | 0.22 | 0.481 | 28.943 | 0.049 | 5.386 | 8.581 | 3.447 | 0.32 | 0.128 | 1.24 | 0.19 | 0.216 | 0.008 | 0.1024 | 0.0076 | 26.48 |
|  | 3 | 5.4 | 6.0923 | 0.018 | 43.92 | 0.78 | 1.58 | 0.25 | 0.658 | 28.155 | 0.046 | 7.093 | 10.704 | 4.402 | 0.33 | 0.156 | 1.46 | 0.17 | 0.224 | 0.008 | 0.1216 | 0.0091 | 24.83 |
|  | 4 | 6.9 | 6.4589 | 0.015 | 25.75 | 0.76 | 1.15 | 0.37 | 2.051 | 17.409 | 0.042 | 17.693 | 20.229 | 9.663 | 0.30 | 0.224 | 1.80 | 0.10 | 0.228 | 0.008 | 0.1565 | 0.0116 | 12.98 |
|  | 5 | 34.4 | 6.8934 | 0.009 | 5.51 | 0.49 | 0.85 | 0.36 | 2.847 | 2.162 | 0.024 | 41.387 | 23.065 | 9.764 | 0.18 | 0.200 | 1.07 | 0.02 | 0.141 | 0.005 | 0.1316 | 0.0092 | 1.68 |
|  | 6 | 38.0 | 7.0179 | 0.003 | 2.58 | 0.31 | 1.35 | 0.31 | 2.142 | 0.389 | 0.014 | 53.008 | 17.842 | 5.975 | 0.11 | 0.213 | 0.48 | 0.00 | 0.081 | 0.003 | 0.1159 | 0.0078 | 0.35 |
|  | 7 | 3.3 |  |  |  |  |  |  |  |  |  |  |  |  |  |  |  |  |  |  |  |  |  |

**Table B3.** *Cont.*

| Sample | Size Interval | Mass Fraction (wt.%) | Density (RD) | Li₂O (wt.%) | Fe₂O₃ (wt.%) | Al₂O₃ (wt.%) | SiO₂ (wt.%) | TiO₂ (wt.%) | Mn (wt.%) | S (wt.%) | P (wt.%) | Sn (wt.%) | Ta₂O₅ (wt.%) | Nb₂O₅ (wt.%) | Na₂O (wt.%) | PbO (wt.%) | CaO (wt.%) | MgO (wt.%) | K₂O (wt.%) | Rb (wt.%) | U₃O₈ (wt.%) | ThO₂ (wt.%) | LOI (wt.%) |
|---|---|---|---|---|---|---|---|---|---|---|---|---|---|---|---|---|---|---|---|---|---|---|---|
| Product Flow 7 | 1 | 4.3 | | | | | | | | | | | | | | | | | | | | | |
| | 2 | 2.3 | 5.5792 | 0.032 | 36.15 | 1.06 | 2.06 | 0.30 | 1.070 | 21.513 | 0.061 | 11.149 | 15.550 | 6.207 | 0.37 | 0.223 | 1.94 | 0.15 | 0.297 | 0.012 | 0.1809 | 0.0131 | 18.84 |
| | 3 | 5.2 | 5.5792 | 0.032 | 36.15 | 1.06 | 2.06 | 0.30 | 1.070 | 21.513 | 0.061 | 11.149 | 15.550 | 6.207 | 0.37 | 0.223 | 1.94 | 0.15 | 0.297 | 0.012 | 0.1809 | 0.0131 | 18.84 |
| | 4 | 8.3 | 6.6485 | 0.014 | 14.35 | 0.72 | 0.98 | 0.39 | 2.687 | 9.259 | 0.038 | 27.589 | 23.886 | 11.228 | 0.29 | 0.252 | 1.67 | 0.07 | 0.211 | 0.008 | 0.1557 | 0.0110 | 6.38 |
| | 5 | 40.5 | 6.9009 | 0.006 | 3.89 | 0.40 | 1.12 | 0.33 | 2.473 | 1.148 | 0.021 | 47.415 | 20.398 | 7.920 | 0.15 | 0.185 | 0.77 | 0.01 | 0.115 | 0.004 | 0.1086 | 0.0073 | 0.95 |
| | 6 | 36.6 | 7.0895 | 0.006 | 2.17 | 0.28 | 1.60 | 0.30 | 1.853 | 0.218 | 0.013 | 55.791 | 15.966 | 5.120 | 0.10 | 0.182 | 0.36 | 0.00 | 0.069 | 0.002 | 0.1059 | 0.0070 | 0.23 |
| | 7 | 2.8 | | | | | | | | | | | | | | | | | | | | | |
| Product Flow 8 | 1 | 1.2 | | | | | | | | | | | | | | | | | | | | | |
| | 2 | 1.7 | 7.5103 | 0.031 | 11.42 | 1.24 | 2.59 | 0.41 | 2.182 | 7.170 | 0.058 | 29.136 | 24.450 | 9.988 | 0.41 | 0.360 | 2.38 | 0.08 | 0.354 | 0.014 | 0.2394 | 0.0167 | 5.30 |
| | 3 | 3.4 | 7.5103 | 0.031 | 11.42 | 1.24 | 2.59 | 0.41 | 2.182 | 7.170 | 0.058 | 29.136 | 24.450 | 9.988 | 0.41 | 0.360 | 2.38 | 0.08 | 0.354 | 0.014 | 0.2394 | 0.0167 | 5.30 |
| | 4 | 8.6 | 7.7905 | 0.012 | 3.93 | 0.58 | 1.10 | 0.38 | 2.890 | 0.983 | 0.025 | 41.487 | 23.811 | 10.431 | 0.20 | 0.268 | 1.21 | 0.02 | 0.176 | 0.006 | 0.1446 | 0.0096 | 0.90 |
| | 5 | 49.0 | 7.2183 | 0.004 | 2.29 | 0.30 | 1.41 | 0.30 | 2.073 | 0.182 | 0.014 | 53.671 | 17.486 | 5.885 | 0.10 | 0.170 | 0.47 | 0.00 | 0.082 | 0.003 | 0.0901 | 0.0059 | 0.27 |
| | 6 | 33.6 | 7.2769 | 0.002 | 1.81 | 0.20 | 1.60 | 0.28 | 1.617 | 0.112 | 0.009 | 58.571 | 14.987 | 3.619 | 0.08 | 0.175 | 0.23 | 0.00 | 0.054 | 0.002 | 0.1192 | 0.0078 | 0.13 |
| | 7 | 2.5 | | | | | | | | | | | | | | | | | | | | | |
| Product Flow 9 | 1 | 2.9 | | | | | | | | | | | | | | | | | | | | | |
| | 2 | 3.4 | 5.8116 | 0.208 | 11.80 | 2.24 | 7.25 | 0.38 | 1.835 | 5.146 | 0.164 | 33.138 | 17.739 | 6.980 | 0.34 | 0.234 | 1.77 | 0.13 | 0.291 | 0.014 | 0.1637 | 0.0115 | 3.85 |
| | 3 | 5.9 | 6.8529 | 0.024 | 5.39 | 0.80 | 2.85 | 0.36 | 1.817 | 2.588 | 0.043 | 47.467 | 17.219 | 6.149 | 0.21 | 0.283 | 1.09 | 0.04 | 0.199 | 0.008 | 0.1384 | 0.0091 | 2.06 |
| | 4 | 17.8 | 6.8961 | 0.010 | 3.57 | 0.45 | 2.08 | 0.32 | 1.957 | 1.319 | 0.021 | 52.202 | 16.382 | 5.924 | 0.15 | 0.167 | 0.51 | 0.01 | 0.106 | 0.004 | 0.0806 | 0.0052 | 1.01 |
| | 5 | 50.6 | 7.2703 | 0.006 | 2.49 | 0.32 | 2.06 | 0.29 | 1.628 | 0.661 | 0.015 | 57.104 | 14.468 | 4.241 | 0.11 | 0.131 | 0.29 | 0.00 | 0.069 | 0.002 | 0.0793 | 0.0052 | 0.49 |
| | 6 | 18.0 | 7.0785 | 0.010 | 2.45 | 0.31 | 1.98 | 0.29 | 1.625 | 0.544 | 0.019 | 57.038 | 15.353 | 3.516 | 0.12 | 0.176 | 0.27 | 0.00 | 0.061 | 0.002 | 0.1790 | 0.0118 | 0.43 |
| | 7 | 1.4 | | | | | | | | | | | | | | | | | | | | | |
| Product Flow 10 | 1 | 2.1 | | | | | | | | | | | | | | | | | | | | | |
| | 2 | 4.8 | 7.2144 | 0.074 | 4.66 | 1.04 | 5.68 | 0.31 | 1.240 | 0.461 | 0.054 | 54.240 | 11.838 | 3.499 | 0.16 | 0.250 | 0.59 | 0.03 | 0.145 | 0.006 | 0.0863 | 0.0056 | 0.71 |
| | 3 | 9.0 | 5.3330 | 0.005 | 1.68 | 0.42 | 2.96 | 0.29 | 1.063 | 0.258 | 0.018 | 61.651 | 10.584 | 2.873 | 0.12 | 0.235 | 0.34 | 0.00 | 0.100 | 0.004 | 0.0748 | 0.0047 | 0.34 |
| | 4 | 21.6 | 7.1930 | 0.000 | 1.82 | 0.29 | 2.18 | 0.29 | 1.410 | 0.194 | 0.012 | 59.491 | 13.333 | 3.335 | 0.11 | 0.172 | 0.23 | 0.00 | 0.071 | 0.002 | 0.0716 | 0.0044 | 0.20 |
| | 5 | 47.8 | 7.1013 | 0.004 | 1.94 | 0.25 | 1.72 | 0.28 | 1.614 | 0.186 | 0.012 | 57.822 | 15.602 | 3.216 | 0.09 | 0.141 | 0.20 | 0.00 | 0.054 | 0.002 | 0.1232 | 0.0079 | 0.17 |
| | 6 | 14.0 | 7.4931 | 0.008 | 2.08 | 0.28 | 1.70 | 0.30 | 1.668 | 0.217 | 0.018 | 57.316 | 16.510 | 2.973 | 0.11 | 0.192 | 0.21 | 0.00 | 0.048 | 0.001 | 0.3573 | 0.0245 | 0.24 |
| | 7 | 0.6 | | | | | | | | | | | | | | | | | | | | | |
| Product Flow 11 | 1 | 7.0 | 7.0266 | 0.099 | 4.40 | 0.96 | 4.59 | 0.30 | 1.076 | 0.738 | 0.054 | 57.089 | 10.854 | 2.577 | 0.12 | 0.448 | 0.42 | 0.03 | 0.106 | 0.004 | 0.1420 | 0.0093 | 0.64 |
| | 2 | 12.8 | 7.4092 | 0.003 | 1.60 | 0.31 | 2.71 | 0.27 | 0.940 | 0.281 | 0.015 | 63.251 | 10.095 | 2.103 | 0.09 | 0.223 | 0.21 | 0.00 | 0.075 | 0.004 | 0.1273 | 0.0079 | 0.27 |
| | 3 | 15.5 | 7.5031 | 0.003 | 1.66 | 0.28 | 2.64 | 0.27 | 0.905 | 0.338 | 0.012 | 63.686 | 9.877 | 1.958 | 0.09 | 0.232 | 0.17 | 0.00 | 0.068 | 0.002 | 0.1817 | 0.0113 | 0.30 |
| | 4 | 20.9 | 7.3380 | 0.000 | 2.13 | 0.21 | 1.88 | 0.28 | 1.347 | 0.441 | 0.011 | 59.978 | 13.926 | 2.497 | 0.08 | 0.212 | 0.17 | 0.00 | 0.051 | 0.002 | 0.2788 | 0.0178 | 0.43 |
| | 5 | 31.6 | 7.2911 | 0.003 | 2.50 | 0.22 | 1.29 | 0.29 | 1.777 | 0.516 | 0.014 | 55.674 | 17.789 | 3.088 | 0.10 | 0.193 | 0.20 | 0.00 | 0.048 | 0.001 | 0.5020 | 0.0353 | 0.39 |
| | 6 | 11.3 | 7.5821 | 0.007 | 2.57 | 0.24 | 1.13 | 0.31 | 1.935 | 0.463 | 0.018 | 54.477 | 19.466 | 3.099 | 0.12 | 0.288 | 0.25 | 0.00 | 0.047 | 0.001 | 0.9970 | 0.0703 | 0.40 |
| | 7 | 0.8 | | | | | | | | | | | | | | | | | | | | | |

**Table B4.** Mass fraction, density and XRF assay data for each size interval of the 11 overflow reject flow fractions. Size 1 = +0.106 mm, Size 2 = −0.106 + 0.090 mm, Size 3 = −0.090 + 0.075 mm, Size 4 = −0.075 + 0.063 mm, Size 5 = −0.063 + 0.053 mm, Size 6 = −0.053 + 0.045 mm, Size 7 = −0.045 mm. Where data are absent, this is because there was insufficient material in the size interval for the density and XRF assay analysis.

| Sample | Size Interval | Mass Fraction (wt.%) | Density (RD) | $Li_2O$ (wt.%) | $Fe_2O_3$ (wt.%) | $Al_2O_3$ (wt.%) | $SiO_2$ (wt.%) | $TiO_2$ (wt.%) | Mn (wt.%) | S (wt.%) | P (wt.%) | Sn (wt.%) | $Ta_2O_5$ (wt.%) | $Nb_2O_5$ (wt.%) | $Na_2O$ (wt.%) | PbO (wt.%) | CaO (wt.%) | MgO (wt.%) | $K_2O$ (wt.%) | Rb (wt.%) | $U_3O_8$ (wt.%) | $ThO_2$ (wt.%) | LOI (wt.%) |
|---|---|---|---|---|---|---|---|---|---|---|---|---|---|---|---|---|---|---|---|---|---|---|---|
| Reject Flow 1 | 1 | 3.8 | | | | | | | | | | | | | | | | | | | | | |
| | 2 | 7.0 | 2.6940 | 0.032 | 0.15 | 18.65 | 65.53 | 0.02 | 0.007 | 0.010 | 0.12 | 0.017 | 0.003 | 0.000 | 4.25 | 0.007 | 0.17 | 0.09 | 9.328 | 0.667 | 0.0003 | 0.0001 | 0.54 |
| | 3 | 8.0 | 2.7489 | 0.031 | 0.14 | 18.36 | 66.68 | 0.00 | 0.008 | 0.012 | 0.11 | 0.022 | 0.004 | 0.000 | 4.89 | 0.004 | 0.17 | 0.10 | 8.034 | 0.573 | 0.0003 | 0.0001 | 0.44 |
| | 4 | 35.3 | 2.6426 | 0.022 | 0.14 | 17.47 | 68.78 | 0.00 | 0.006 | 0.006 | 0.10 | 0.010 | 0.003 | 0.000 | 5.45 | 0.005 | 0.25 | 0.09 | 6.483 | 0.444 | 0.0002 | 0.0001 | 0.30 |
| | 5 | 23.5 | 2.6772 | 0.026 | 0.18 | 17.17 | 69.30 | 0.01 | 0.009 | 0.012 | 0.09 | 0.024 | 0.009 | 0.000 | 5.58 | 0.005 | 0.37 | 0.11 | 5.705 | 0.382 | 0.0002 | 0.0001 | 0.41 |
| | 6 | 18.8 | 2.6715 | 0.047 | 0.26 | 17.04 | 69.69 | 0.01 | 0.010 | 0.022 | 0.09 | 0.033 | 0.012 | 0.001 | 5.60 | 0.004 | 0.49 | 0.15 | 5.211 | 0.339 | 0.0010 | 0.0002 | 0.43 |
| | 7 | 3.6 | | | | | | | | | | | | | | | | | | | | | |
| Reject Flow 2 | 1 | 3.2 | | | | | | | | | | | | | | | | | | | | | |
| | 2 | 7.1 | 2.7448 | 0.007 | 0.10 | 16.80 | 71.79 | 0.00 | 0.004 | 0.002 | 0.07 | 0.005 | 0.003 | 0.000 | 8.89 | 0.002 | 0.15 | 0.06 | 1.300 | 0.074 | 0.0002 | 0.0000 | 0.29 |
| | 3 | 9.0 | 2.7717 | 0.008 | 0.12 | 16.01 | 73.15 | 0.00 | 0.004 | 0.002 | 0.06 | 0.005 | 0.000 | 0.000 | 8.46 | 0.000 | 0.20 | 0.08 | 1.145 | 0.063 | 0.0002 | 0.0001 | 0.28 |
| | 4 | 34.3 | 2.6798 | 0.006 | 0.13 | 15.00 | 75.00 | 0.00 | 0.005 | 0.002 | 0.05 | 0.004 | 0.002 | 0.000 | 7.77 | 0.001 | 0.40 | 0.07 | 0.956 | 0.049 | 0.0002 | 0.0001 | 0.24 |
| | 5 | 28.0 | 2.6806 | 0.012 | 0.16 | 14.91 | 75.47 | 0.00 | 0.005 | 0.002 | 0.05 | 0.006 | 0.004 | 0.000 | 7.36 | 0.000 | 0.60 | 0.09 | 0.939 | 0.045 | 0.0002 | 0.0001 | 0.29 |
| | 6 | 15.4 | 2.6985 | 0.023 | 0.25 | 15.03 | 75.12 | 0.01 | 0.009 | 0.010 | 0.05 | 0.013 | 0.005 | 0.000 | 7.08 | 0.001 | 0.76 | 0.13 | 0.977 | 0.045 | 0.0002 | 0.0002 | 0.37 |
| | 7 | 3.1 | | | | | | | | | | | | | | | | | | | | | |
| Reject Flow 3 | 1 | 2.4 | | | | | | | | | | | | | | | | | | | | | |
| | 2 | 5.3 | 2.7758 | 0.014 | 0.14 | 14.43 | 76.05 | 0.01 | 0.005 | 0.004 | 0.06 | 0.005 | 0.003 | 0.002 | 7.47 | 0.000 | 0.21 | 0.08 | 1.095 | 0.059 | 0.0003 | 0.0001 | 0.32 |
| | 3 | 6.9 | 2.7810 | 0.013 | 0.15 | 13.94 | 76.62 | 0.01 | 0.005 | 0.003 | 0.05 | 0.006 | 0.001 | 0.000 | 7.07 | 0.000 | 0.31 | 0.08 | 0.955 | 0.048 | 0.0002 | 0.0001 | 0.33 |
| | 4 | 35.8 | 2.6864 | 0.012 | 0.16 | 13.38 | 77.97 | 0.01 | 0.005 | 0.002 | 0.04 | 0.003 | 0.002 | 0.000 | 6.61 | 0.000 | 0.60 | 0.08 | 0.723 | 0.032 | 0.0002 | 0.0001 | 0.25 |
| | 5 | 29.0 | 2.6845 | 0.014 | 0.19 | 13.74 | 77.27 | 0.01 | 0.005 | 0.003 | 0.04 | 0.004 | 0.001 | 0.000 | 6.52 | 0.001 | 0.88 | 0.10 | 0.690 | 0.027 | 0.0001 | 0.0001 | 0.30 |
| | 6 | 16.3 | 2.7216 | 0.027 | 0.31 | 14.23 | 76.04 | 0.02 | 0.008 | 0.010 | 0.04 | 0.011 | 0.004 | 0.000 | 6.46 | 0.000 | 1.04 | 0.14 | 0.737 | 0.028 | 0.0002 | 0.0002 | 0.42 |
| | 7 | 4.3 | | | | | | | | | | | | | | | | | | | | | |
| Reject Flow 4 | 1 | 2.7 | | | | | | | | | | | | | | | | | | | | | |
| | 2 | 6.3 | 2.7477 | 0.010 | 0.13 | 13.41 | 77.60 | 0.01 | 0.004 | 0.001 | 0.05 | 0.002 | 0.002 | 0.000 | 7.21 | 0.002 | 0.26 | 0.07 | 0.554 | 0.021 | 0.0002 | 0.0001 | 0.31 |
| | 3 | 8.4 | 2.7659 | 0.014 | 0.16 | 12.44 | 79.53 | 0.01 | 0.004 | 0.002 | 0.04 | 0.001 | 0.002 | 0.000 | 6.31 | 0.000 | 0.46 | 0.09 | 0.493 | 0.016 | 0.0002 | 0.0001 | 0.32 |
| | 4 | 36.3 | 2.7065 | 0.009 | 0.19 | 12.26 | 79.60 | 0.01 | 0.004 | 0.001 | 0.03 | 0.000 | 0.000 | 0.000 | 5.80 | 0.000 | 0.92 | 0.09 | 0.415 | 0.012 | 0.0001 | 0.0002 | 0.19 |
| | 5 | 28.0 | 2.7016 | 0.014 | 0.22 | 12.88 | 78.54 | 0.02 | 0.005 | 0.002 | 0.03 | 0.000 | 0.002 | 0.000 | 5.72 | 0.000 | 1.27 | 0.11 | 0.469 | 0.014 | 0.0001 | 0.0002 | 0.25 |
| | 6 | 15.6 | 2.7193 | 0.031 | 0.33 | 13.70 | 77.08 | 0.03 | 0.007 | 0.006 | 0.03 | 0.009 | 0.005 | 0.000 | 5.71 | 0.001 | 1.46 | 0.16 | 0.593 | 0.020 | 0.0002 | 0.0002 | 0.46 |
| | 7 | 2.7 | | | | | | | | | | | | | | | | | | | | | |
| Reject Flow 5 | 1 | 3.8 | | | | | | | | | | | | | | | | | | | | | |
| | 2 | 8.0 | 2.7453 | 0.020 | 0.19 | 10.84 | 81.66 | 0.02 | 0.004 | 0.001 | 0.04 | 0.000 | 0.002 | 0.000 | 5.35 | 0.000 | 0.54 | 0.10 | 0.418 | 0.013 | 0.0002 | 0.0001 | 0.35 |
| | 3 | 9.0 | 2.7227 | 0.013 | 0.22 | 11.38 | 81.12 | 0.02 | 0.004 | 0.002 | 0.03 | 0.000 | 0.002 | 0.000 | 4.97 | 0.002 | 1.20 | 0.10 | 0.395 | 0.012 | 0.0001 | 0.0002 | 0.26 |
| | 4 | 39.4 | 2.7594 | 0.016 | 0.24 | 12.43 | 79.38 | 0.02 | 0.005 | 0.002 | 0.03 | 0.000 | 0.000 | 0.000 | 5.13 | 0.002 | 1.49 | 0.12 | 0.454 | 0.013 | 0.0001 | 0.0002 | 0.36 |
| | 5 | 24.8 | 2.8811 | 0.049 | 0.32 | 13.54 | 77.29 | 0.02 | 0.006 | 0.004 | 0.03 | 0.004 | 0.001 | 0.000 | 5.31 | 0.000 | 1.65 | 0.17 | 0.599 | 0.021 | 0.0008 | 0.0003 | 0.59 |
| | 6 | 12.7 | 2.8811 | 0.049 | 0.32 | 13.54 | 77.29 | 0.02 | 0.006 | 0.004 | 0.03 | 0.004 | 0.001 | 0.000 | 5.31 | 0.000 | 1.65 | 0.17 | 0.599 | 0.021 | 0.0008 | 0.0003 | 0.59 |
| | 7 | 2.3 | | | | | | | | | | | | | | | | | | | | | |
| Reject Flow 6 | 1 | 5.9 | 2.8268 | 0.010 | 0.14 | 12.41 | 79.55 | 0.01 | 0.003 | 0.002 | 0.04 | 0.000 | 0.002 | 0.000 | 6.62 | 0.000 | 0.24 | 0.08 | 0.419 | 0.015 | 0.0002 | 0.0001 | 0.38 |
| | 2 | 10.9 | 2.7421 | 0.017 | 0.18 | 11.61 | 80.82 | 0.01 | 0.005 | 0.002 | 0.04 | 0.004 | 0.004 | 0.000 | 5.90 | 0.002 | 0.44 | 0.10 | 0.417 | 0.013 | 0.0001 | 0.0001 | 0.31 |
| | 3 | 10.9 | 2.7271 | 0.022 | 0.20 | 11.46 | 80.83 | 0.02 | 0.005 | 0.001 | 0.04 | 0.000 | 0.002 | 0.000 | 5.55 | 0.001 | 0.63 | 0.11 | 0.443 | 0.014 | 0.0002 | 0.0001 | 0.35 |
| | 4 | 56.3 | 2.7030 | 0.021 | 0.22 | 11.78 | 80.72 | 0.02 | 0.005 | 0.002 | 0.03 | 0.005 | 0.003 | 0.000 | 5.13 | 0.001 | 1.18 | 0.11 | 0.425 | 0.013 | 0.0002 | 0.0002 | 0.26 |
| | 5 | 7.8 | 2.7910 | 0.033 | 0.26 | 12.35 | 79.26 | 0.02 | 0.005 | 0.002 | 0.03 | 0.002 | 0.000 | 0.000 | 5.01 | 0.001 | 1.55 | 0.12 | 0.461 | 0.013 | 0.0002 | 0.0002 | 0.39 |
| | 6 | 6.5 | 2.7921 | 0.058 | 0.34 | 13.29 | 78.04 | 0.03 | 0.009 | 0.004 | 0.03 | 0.005 | 0.004 | 0.000 | 5.09 | 0.002 | 1.71 | 0.17 | 0.588 | 0.022 | 0.0003 | 0.0002 | 0.49 |
| | 7 | 1.8 | | | | | | | | | | | | | | | | | | | | | |

**Table B4.** *Cont.*

| Sample | Size Interval | Mass Fraction (wt.%) | Density (RD) | Li₂O (wt.%) | Fe₂O₃ (wt.%) | Al₂O₃ (wt.%) | SiO₂ (wt.%) | TiO₂ (wt.%) | Mn (wt.%) | S (wt.%) | P (wt.%) | Sn (wt.%) | Ta₂O₅ (wt.%) | Nb₂O₅ (wt.%) | Na₂O (wt.%) | PbO (wt.%) | CaO (wt.%) | MgO (wt.%) | K₂O (wt.%) | Rb (wt.%) | U₃O₈ (wt.%) | ThO₂ (wt.%) | LOI (wt.%) |
|---|---|---|---|---|---|---|---|---|---|---|---|---|---|---|---|---|---|---|---|---|---|---|---|
| Reject Flow 7 | 1 | 5.8 | 2.8945 | 0.046 | 0.30 | 9.61 | 83.92 | 0.02 | 0.006 | 0.006 | 0.04 | 0.004 | 0.001 | 0.000 | 4.44 | 0.000 | 0.44 | 0.14 | 0.483 | 0.020 | 0.0002 | 0.0001 | 0.30 |
| | 2 | 9.3 | 2.8084 | 0.052 | 0.34 | 9.82 | 83.54 | 0.03 | 0.007 | 0.005 | 0.04 | 0.005 | 0.004 | 0.000 | 4.02 | 0.002 | 0.79 | 0.18 | 0.580 | 0.026 | 0.0002 | 0.0002 | 0.37 |
| | 3 | 8.6 | 2.8149 | 0.062 | 0.37 | 10.49 | 82.53 | 0.03 | 0.008 | 0.005 | 0.04 | 0.005 | 0.003 | 0.000 | 3.87 | 0.000 | 1.06 | 0.20 | 0.681 | 0.031 | 0.0002 | 0.0002 | 0.46 |
| | 4 | 51.8 | 2.7127 | 0.054 | 0.34 | 11.54 | 80.89 | 0.04 | 0.009 | 0.003 | 0.04 | 0.005 | 0.002 | 0.000 | 4.01 | 0.002 | 1.71 | 0.16 | 0.604 | 0.027 | 0.0002 | 0.0003 | 0.34 |
| | 5 | 12.9 | 2.7464 | 0.059 | 0.37 | 12.63 | 78.78 | 0.04 | 0.008 | 0.005 | 0.03 | 0.003 | 0.002 | 0.000 | 4.24 | 0.000 | 2.11 | 0.18 | 0.631 | 0.026 | 0.0002 | 0.0002 | 0.43 |
| | 6 | 9.6 | 2.7752 | 0.098 | 0.48 | 13.93 | 76.55 | 0.04 | 0.012 | 0.008 | 0.04 | 0.007 | 0.004 | 0.000 | 4.40 | 0.002 | 2.22 | 0.25 | 0.856 | 0.038 | 0.0003 | 0.0003 | 0.60 |
| | 7 | 2.0 | | | | | | | | | | | | | | | | | | | | | |
| Reject Flow 8 | 1 | 12.3 | 2.7381 | 0.150 | 0.64 | 12.17 | 79.87 | 0.04 | 0.019 | 0.009 | 0.07 | 0.016 | 0.007 | 0.002 | 3.03 | 0.000 | 0.62 | 0.30 | 1.776 | 0.146 | 0.0003 | 0.0002 | 0.99 |
| | 2 | 14.2 | 2.7585 | 0.235 | 0.73 | 13.58 | 77.39 | 0.06 | 0.022 | 0.007 | 0.08 | 0.015 | 0.007 | 0.001 | 2.71 | 0.000 | 1.03 | 0.37 | 2.080 | 0.166 | 0.0003 | 0.0003 | 1.12 |
| | 3 | 8.2 | 2.7969 | 0.326 | 0.83 | 15.22 | 74.99 | 0.07 | 0.026 | 0.008 | 0.09 | 0.017 | 0.006 | 0.003 | 2.67 | 0.000 | 1.33 | 0.44 | 2.321 | 0.186 | 0.0004 | 0.0003 | 1.25 |
| | 4 | 49.7 | 2.8066 | 0.935 | 1.05 | 17.65 | 70.97 | 0.10 | 0.035 | 0.010 | 0.13 | 0.018 | 0.007 | 0.000 | 2.86 | 0.003 | 1.95 | 0.55 | 2.051 | 0.153 | 0.0004 | 0.0004 | 1.16 |
| | 5 | 5.5 | 2.8997 | 0.891 | 1.18 | 18.17 | 69.42 | 0.12 | 0.036 | 0.011 | 0.20 | 0.018 | 0.006 | 0.001 | 3.11 | 0.002 | 2.45 | 0.65 | 1.940 | 0.137 | 0.0004 | 0.0005 | 1.17 |
| | 6 | 7.1 | 2.9171 | 1.931 | 1.74 | 21.28 | 63.17 | 0.16 | 0.058 | 0.017 | 0.45 | 0.027 | 0.005 | 0.000 | 2.35 | 0.000 | 2.79 | 0.96 | 2.248 | 0.152 | 0.0006 | 0.0007 | 1.51 |
| | 7 | 3.0 | | | | | | | | | | | | | | | | | | | | | |
| Reject Flow 9 | 1 | 9.4 | 3.0905 | 3.587 | 1.68 | 27.55 | 56.84 | 0.14 | 0.082 | 0.022 | 0.41 | 0.039 | 0.012 | 0.009 | 0.82 | 0.002 | 1.44 | 0.92 | 3.274 | 0.311 | 0.0008 | 0.0006 | 2.18 |
| | 2 | 8.9 | 3.0905 | 3.587 | 1.68 | 27.55 | 56.84 | 0.14 | 0.082 | 0.022 | 0.41 | 0.039 | 0.012 | 0.009 | 0.82 | 0.002 | 1.44 | 0.92 | 3.274 | 0.311 | 0.0008 | 0.0006 | 2.18 |
| | 3 | 8.9 | 3.0905 | 3.587 | 1.68 | 27.55 | 56.84 | 0.14 | 0.082 | 0.022 | 0.41 | 0.039 | 0.012 | 0.009 | 0.82 | 0.002 | 1.44 | 0.92 | 3.274 | 0.311 | 0.0008 | 0.0006 | 2.18 |
| | 4 | 25.8 | 3.2383 | 4.542 | 2.12 | 26.74 | 55.51 | 0.20 | 0.090 | 0.017 | 0.90 | 0.039 | 0.006 | 0.005 | 0.75 | 0.001 | 2.96 | 1.34 | 1.546 | 0.118 | 0.0005 | 0.0006 | 1.61 |
| | 5 | 23.0 | 3.2648 | 4.617 | 2.25 | 25.84 | 53.32 | 0.21 | 0.103 | 0.053 | 1.55 | 0.040 | 0.008 | 0.009 | 0.74 | 0.001 | 4.97 | 1.46 | 1.142 | 0.080 | 0.0006 | 0.0008 | 1.50 |
| | 6 | 19.6 | 3.2798 | 4.480 | 2.19 | 24.87 | 51.79 | 0.21 | 0.115 | 0.027 | 2.08 | 0.033 | 0.004 | 0.008 | 0.74 | 0.000 | 6.60 | 1.44 | 0.878 | 0.056 | 0.0008 | 0.0014 | 1.42 |
| | 7 | 4.4 | | | | | | | | | | | | | | | | | | | | | |
| Reject Flow 10 | 1 | 9.9 | 3.2752 | 4.423 | 2.82 | 25.29 | 52.32 | 0.19 | 0.137 | 0.153 | 1.71 | 0.043 | 0.029 | 0.017 | 0.71 | 0.001 | 5.42 | 1.24 | 1.104 | 0.096 | 0.0023 | 0.0022 | 1.62 |
| | 2 | 12.0 | 3.2997 | 3.941 | 2.56 | 24.42 | 48.13 | 0.22 | 0.162 | 0.249 | 2.71 | 0.055 | 0.039 | 0.024 | 0.74 | 0.000 | 8.49 | 1.50 | 0.969 | 0.085 | 0.0025 | 0.0029 | 1.67 |
| | 3 | 10.8 | 3.3069 | 3.881 | 2.58 | 24.24 | 47.71 | 0.24 | 0.159 | 0.351 | 2.85 | 0.056 | 0.048 | 0.028 | 0.72 | 0.003 | 8.84 | 1.51 | 0.963 | 0.080 | 0.0028 | 0.0032 | 1.72 |
| | 4 | 25.5 | 3.1856 | 3.416 | 3.30 | 22.33 | 43.05 | 0.32 | 0.175 | 0.591 | 4.01 | 0.069 | 0.071 | 0.041 | 0.76 | 0.005 | 12.34 | 1.71 | 0.629 | 0.047 | 0.0031 | 0.0038 | 1.77 |
| | 5 | 25.4 | 3.2063 | 3.107 | 3.64 | 20.82 | 40.03 | 0.36 | 0.184 | 0.864 | 4.85 | 0.095 | 0.105 | 0.057 | 0.72 | 0.006 | 14.87 | 1.72 | 0.499 | 0.032 | 0.0036 | 0.0044 | 1.78 |
| | 6 | 12.9 | 3.2766 | 3.152 | 3.92 | 20.91 | 40.22 | 0.37 | 0.177 | 1.099 | 4.68 | 0.169 | 0.170 | 0.083 | 0.73 | 0.005 | 14.29 | 1.77 | 0.416 | 0.022 | 0.0045 | 0.0057 | 1.95 |
| | 7 | 3.6 | | | | | | | | | | | | | | | | | | | | | |
| Reject Flow 11 | 1 | 8.5 | 3.4919 | 3.753 | 6.95 | 22.74 | 48.56 | 0.21 | 0.174 | 0.471 | 2.08 | 0.420 | 0.171 | 0.066 | 0.75 | 0.007 | 6.46 | 1.14 | 1.079 | 0.091 | 0.0057 | 0.0033 | 2.02 |
| | 2 | 10.1 | 3.3667 | 3.514 | 3.00 | 21.91 | 45.66 | 0.26 | 0.205 | 0.570 | 3.60 | 0.256 | 0.128 | 0.060 | 0.86 | 0.004 | 11.07 | 1.45 | 0.588 | 0.044 | 0.0052 | 0.0052 | 1.63 |
| | 3 | 9.8 | 3.3647 | 3.412 | 3.25 | 21.03 | 44.09 | 0.28 | 0.207 | 0.938 | 3.97 | 0.254 | 0.162 | 0.074 | 0.85 | 0.005 | 12.25 | 1.44 | 0.454 | 0.030 | 0.0055 | 0.0057 | 1.75 |
| | 4 | 27.1 | 3.2531 | 2.771 | 4.67 | 18.50 | 38.87 | 0.41 | 0.222 | 1.823 | 5.15 | 0.317 | 0.303 | 0.133 | 0.91 | 0.010 | 15.75 | 1.49 | 0.310 | 0.016 | 0.0071 | 0.0064 | 2.20 |
| | 5 | 23.9 | 3.2981 | 2.332 | 6.33 | 16.37 | 33.96 | 0.48 | 0.237 | 3.170 | 5.96 | 0.548 | 0.520 | 0.221 | 0.89 | 0.011 | 18.23 | 1.44 | 0.273 | 0.013 | 0.0093 | 0.0074 | 2.66 |
| | 6 | 16.7 | 3.4055 | 2.268 | 8.38 | 15.90 | 32.46 | 0.54 | 0.287 | 4.681 | 5.16 | 1.710 | 1.343 | 0.560 | 0.87 | 0.022 | 15.89 | 1.44 | 0.268 | 0.013 | 0.0181 | 0.0094 | 3.76 |
| | 7 | 3.9 | | | | | | | | | | | | | | | | | | | | | |

**Table B5.** Reconstituted mass fraction, density and XRF assay data for each size interval of the +0.045 mm wet-screened feed, product and reject samples. Note that the masses in Size Interval 7 were too small for pycnometry and XRF analysis and so were assumed to have the same properties as Size Interval 6 (hence the Interval 7 assays are shown in grey to indicate their higher uncertainty). Similarly, the Size Interval 1 data are based on many density and assay values, which were assumed to be the same as Size Interval 2 due to insufficient mass (see Tables B2–B4). Yields and recoveries calculated using Equations (1) and (2).

| Sample | Size Interval | Mass Fraction (wt.%) | Density (RD) | Li$_2$O (wt.%) | Fe$_2$O$_3$ (wt.%) | Al$_2$O$_3$ (wt.%) | SiO$_2$ (wt.%) | TiO$_2$ (wt.%) | Mn (wt.%) | S (wt.%) | P (wt.%) | Sn (wt.%) | Ta$_2$O$_5$ (wt.%) | Nb$_2$O$_5$ (wt.%) | Na$_2$O (wt.%) | PbO (wt.%) | CaO (wt.%) | MgO (wt.%) | K$_2$O (wt.%) | Rb (wt.%) | U$_3$O$_8$ (wt.%) | ThO$_2$ (wt.%) | LOI (wt.%) |
|---|---|---|---|---|---|---|---|---|---|---|---|---|---|---|---|---|---|---|---|---|---|---|---|
| Feed | 1 | 6.7 | 2.954 | 1.209 | 3.55 | 16.22 | 65.47 | 0.10 | 0.085 | 1.176 | 0.55 | 0.871 | 0.360 | 0.125 | 3.35 | 0.007 | 1.95 | 0.48 | 1.819 | 0.133 | 0.0072 | 0.0016 | 1.77 |
|  | 2 | 9.1 | 2.863 | 0.932 | 1.49 | 15.84 | 68.70 | 0.08 | 0.062 | 0.641 | 0.65 | 0.688 | 0.224 | 0.075 | 4.07 | 0.004 | 2.27 | 0.46 | 1.740 | 0.118 | 0.0037 | 0.0012 | 1.08 |
|  | 3 | 9.1 | 2.877 | 0.826 | 1.48 | 15.60 | 68.88 | 0.08 | 0.059 | 0.664 | 0.65 | 0.702 | 0.231 | 0.079 | 4.18 | 0.005 | 2.39 | 0.44 | 1.707 | 0.115 | 0.0038 | 0.0012 | 1.07 |
|  | 4 | 32.9 | 2.801 | 0.602 | 1.35 | 14.47 | 70.34 | 0.08 | 0.066 | 0.558 | 0.66 | 0.815 | 0.333 | 0.120 | 4.35 | 0.004 | 2.81 | 0.38 | 1.310 | 0.079 | 0.0038 | 0.0010 | 0.86 |
|  | 5 | 24.7 | 2.867 | 0.743 | 1.80 | 15.13 | 65.66 | 0.11 | 0.106 | 0.814 | 0.94 | 1.579 | 0.637 | 0.226 | 4.28 | 0.007 | 3.70 | 0.46 | 1.319 | 0.078 | 0.0077 | 0.0016 | 1.12 |
|  | 6 | 15.0 | 2.857 | 0.884 | 2.00 | 15.68 | 63.60 | 0.13 | 0.140 | 0.875 | 0.90 | 2.269 | 0.933 | 0.327 | 4.19 | 0.013 | 3.59 | 0.50 | 1.321 | 0.076 | 0.0136 | 0.0022 | 1.24 |
|  | 7 | 2.5 | 2.897 | 1.084 | 2.53 | 16.00 | 59.70 | 0.16 | 0.187 | 1.165 | 1.11 | 3.166 | 1.297 | 0.454 | 3.85 | 0.018 | 4.10 | 0.59 | 1.381 | 0.082 | 0.0188 | 0.0028 | 1.49 |
| Product | 1 | 13.7 | 3.892 | 2.125 | 17.11 | 12.25 | 26.15 | 0.33 | 0.377 | 13.232 | 1.39 | 5.079 | 2.863 | 1.096 | 0.69 | 0.068 | 4.65 | 0.85 | 0.353 | 0.018 | 0.0537 | 0.0129 | 10.64 |
|  | 2 | 6.4 | 5.347 | 0.165 | 28.43 | 2.21 | 5.64 | 0.56 | 0.598 | 18.608 | 0.55 | 18.199 | 6.678 | 2.282 | 0.47 | 0.133 | 2.20 | 0.32 | 0.241 | 0.009 | 0.0875 | 0.0123 | 16.25 |
|  | 3 | 7.5 | 5.581 | 0.077 | 26.98 | 1.32 | 3.79 | 0.49 | 0.715 | 17.485 | 0.21 | 24.857 | 8.242 | 2.769 | 0.35 | 0.162 | 1.24 | 0.20 | 0.205 | 0.008 | 0.1045 | 0.0101 | 15.39 |
|  | 4 | 16.0 | 5.709 | 0.042 | 28.19 | 0.89 | 2.95 | 0.44 | 1.051 | 16.499 | 0.11 | 25.945 | 10.183 | 3.681 | 0.24 | 0.137 | 0.86 | 0.15 | 0.150 | 0.005 | 0.0961 | 0.0083 | 14.62 |
|  | 5 | 34.3 | 6.301 | 0.024 | 15.03 | 0.58 | 2.11 | 0.35 | 1.665 | 8.208 | 0.06 | 39.087 | 15.030 | 5.184 | 0.18 | 0.150 | 0.74 | 0.07 | 0.113 | 0.004 | 0.1289 | 0.0098 | 7.09 |
|  | 6 | 20.3 | 6.519 | 0.027 | 9.97 | 0.56 | 2.06 | 0.33 | 1.833 | 5.498 | 0.06 | 44.245 | 16.309 | 5.412 | 0.17 | 0.220 | 0.70 | 0.05 | 0.100 | 0.003 | 0.1821 | 0.0133 | 4.35 |
|  | 7 | 1.9 | 6.556 | 0.037 | 10.99 | 0.67 | 2.21 | 0.34 | 1.908 | 6.307 | 0.07 | 41.615 | 16.780 | 5.993 | 0.18 | 0.232 | 0.85 | 0.06 | 0.113 | 0.004 | 0.1694 | 0.0127 | 4.85 |
| Reject | 1 | 6.2 | 2.917 | 1.305 | 1.48 | 16.65 | 70.11 | 0.07 | 0.050 | 0.076 | 0.53 | 0.058 | 0.025 | 0.010 | 3.60 | 0.002 | 1.81 | 0.47 | 1.752 | 0.134 | 0.0011 | 0.0008 | 1.00 |
|  | 2 | 9.2 | 2.865 | 0.961 | 0.87 | 15.99 | 70.47 | 0.07 | 0.047 | 0.080 | 0.67 | 0.035 | 0.019 | 0.008 | 4.05 | 0.002 | 2.38 | 0.47 | 1.753 | 0.126 | 0.0009 | 0.0009 | 0.85 |
|  | 3 | 8.9 | 2.873 | 0.917 | 0.85 | 16.01 | 70.25 | 0.07 | 0.045 | 0.120 | 0.70 | 0.034 | 0.021 | 0.010 | 4.27 | 0.001 | 2.55 | 0.45 | 1.650 | 0.115 | 0.0009 | 0.0009 | 0.81 |
|  | 4 | 39.6 | 2.778 | 0.646 | 0.82 | 15.09 | 72.15 | 0.07 | 0.034 | 0.132 | 0.57 | 0.027 | 0.022 | 0.009 | 4.43 | 0.002 | 2.62 | 0.38 | 1.358 | 0.086 | 0.0007 | 0.0007 | 0.65 |
|  | 5 | 19.9 | 2.842 | 0.854 | 1.30 | 15.84 | 66.85 | 0.11 | 0.053 | 0.391 | 1.21 | 0.068 | 0.062 | 0.027 | 4.50 | 0.003 | 4.42 | 0.51 | 1.259 | 0.073 | 0.0014 | 0.0013 | 0.81 |
|  | 6 | 13.1 | 2.873 | 0.970 | 1.64 | 16.52 | 65.49 | 0.13 | 0.061 | 0.556 | 1.11 | 0.193 | 0.149 | 0.062 | 4.29 | 0.004 | 4.23 | 0.57 | 1.415 | 0.082 | 0.0025 | 0.0017 | 1.05 |
|  | 7 | 3.1 | 2.893 | 1.121 | 1.78 | 16.94 | 64.34 | 0.14 | 0.067 | 0.577 | 1.21 | 0.197 | 0.152 | 0.064 | 4.02 | 0.004 | 4.54 | 0.64 | 1.377 | 0.080 | 0.0026 | 0.0018 | 1.14 |
| Mass Yield to Product (wt.%) | 1 |  |  | −11.8 | 13.3 | 9.7 | 10.6 | 11.0 | 10.6 | 8.4 | 3.3 | 16.2 | 11.8 | 10.5 | 8.4 | 8.4 | 4.8 | 2.4 | −4.8 | 0.1 | 11.5 | 6.7 | 7.9 |
|  | 2 |  |  | 3.7 | 2.2 | 1.1 | 2.7 | 2.2 | 2.7 | 3.0 | 16.5 | 3.6 | 3.1 | 2.9 | −0.4 | 2.0 | 59.3 | 3.4 | 0.9 | 6.8 | 3.3 | 2.7 | 1.5 |
|  | 3 |  |  | 10.9 | 2.4 | 2.8 | 2.0 | 2.2 | 2.1 | 3.1 | 9.5 | 2.7 | 2.6 | 2.5 | 2.1 | 2.0 | 12.1 | 2.7 | −3.9 | −0.7 | 2.8 | 2.8 | 1.8 |
|  | 4 |  |  | 7.2 | 1.9 | 4.4 | 2.6 | 3.0 | 3.2 | 2.6 | −18.3 | 3.0 | 3.1 | 3.0 | 1.9 | 1.1 | −10.8 | −1.2 | 4.0 | 8.4 | 3.3 | 4.3 | 1.5 |
|  | 5 |  |  | 13.4 | 3.6 | 4.7 | 1.8 | 1.9 | 3.3 | 5.4 | 23.2 | 3.9 | 3.8 | 3.9 | 5.1 | 3.3 | 19.6 | 10.5 | −5.3 | −6.7 | 4.9 | 3.5 | 4.9 |
|  | 6 |  |  | 9.1 | 4.3 | 5.3 | 3.0 | 0.8 | 4.4 | 6.4 | 20.3 | 4.7 | 4.9 | 5.0 | 2.3 | 4.5 | 18.1 | 12.7 | 7.2 | 6.6 | 6.2 | 4.4 | 5.9 |
|  | 7 |  |  | 3.4 | 8.1 | 5.8 | 7.5 | 8.4 | 6.6 | 10.2 | 8.9 | 7.2 | 6.9 | 6.6 | 4.5 | 6.4 | 11.8 | 8.9 | −0.3 | −2.7 | 9.7 | 9.4 | 9.5 |
| Recovery to Product (%) | 1 |  |  | −20.7 | 64.0 | 7.3 | 4.2 | 35.7 | 47.2 | 94.1 | 8.3 | 94.4 | 93.8 | 92.5 | 1.7 | 79.6 | 11.4 | 4.3 | −0.9 | 0.0 | 86.1 | 55.0 | 47.8 |
|  | 2 |  |  | 0.6 | 42.9 | 0.1 | 0.2 | 15.0 | 26.4 | 87.9 | 13.9 | 95.0 | 91.9 | 89.0 | 0.0 | 61.6 | 57.4 | 2.3 | 0.1 | 0.5 | 76.3 | 27.9 | 22.3 |
|  | 3 |  |  | 1.0 | 44.0 | 0.2 | 0.1 | 12.9 | 25.5 | 82.5 | 3.1 | 95.2 | 91.1 | 87.7 | 0.2 | 69.4 | 6.3 | 1.3 | −0.5 | 0.0 | 76.2 | 24.0 | 25.9 |
|  | 4 |  |  | 0.5 | 40.5 | 0.3 | 0.1 | 15.7 | 50.3 | 77.0 | −3.2 | 96.7 | 93.5 | 92.4 | 0.1 | 39.1 | −3.3 | −0.5 | 0.5 | 0.6 | 81.9 | 33.8 | 26.1 |
|  | 5 |  |  | 0.4 | 30.3 | 0.2 | 0.1 | 5.8 | 51.7 | 54.6 | 1.5 | 95.9 | 90.7 | 88.5 | 0.2 | 66.6 | 3.9 | 1.5 | −0.5 | −0.3 | 82.3 | 21.1 | 31.1 |
|  | 6 |  |  | 0.3 | 21.5 | 0.2 | 0.1 | 2.1 | 58.2 | 40.5 | 1.3 | 91.9 | 84.8 | 82.0 | 0.1 | 74.3 | 3.5 | 1.2 | 0.5 | 0.3 | 82.8 | 26.9 | 20.6 |
|  | 7 |  |  | 0.1 | 35.3 | 0.2 | 0.3 | 18.2 | 66.8 | 55.5 | 0.6 | 94.2 | 89.1 | 86.9 | 0.2 | 81.8 | 2.4 | 0.9 | 0.0 | −0.1 | 87.6 | 42.3 | 31.0 |

## Appendix C. XRD Data for Product and Reject

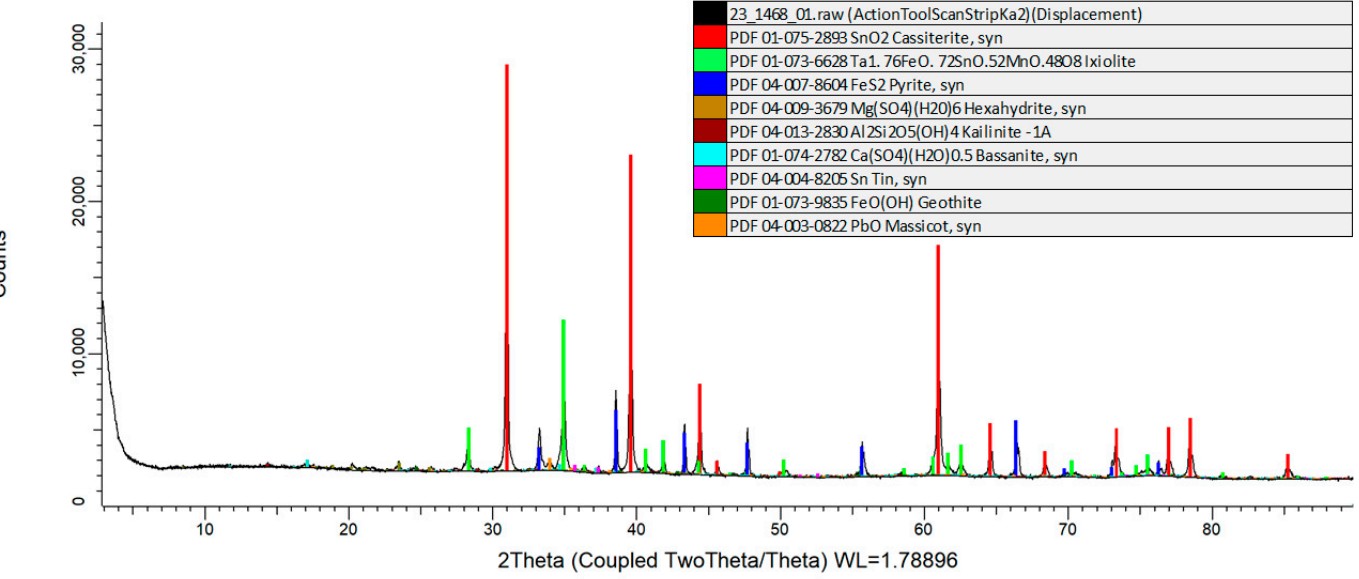

**Figure C1.** Product XRD scan.

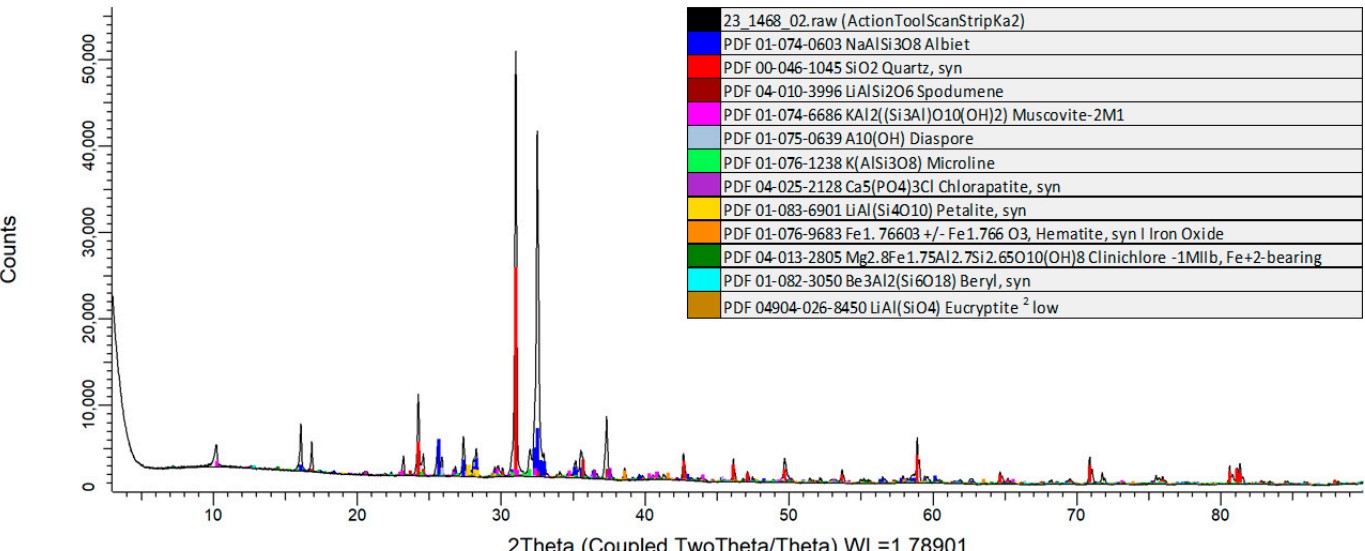

**Figure C2.** Reject XRD scan.

**Table C1.** Product and reject XRD interpretation. The ICDD match probability is reported as an indication of how well the peak positions and relative intensities match those in the published literature (www.icdd.org; accessed on 9 May 2023) for that compound.

| Crystalline Mineral Phase | Concentration (wt.%) | ICDD Match Probability |
|---|---|---|
| **Product Sample:** | | |
| Cassiterite, syn ($SnO_2$) | 44 | High |
| Ixiolite ($Ta_{1.76}Fe_{0.72}Sn_{0.52}Nb_{0.52}Mn_{0.48}O_8$) | 22 | High |
| Pyrite, syn ($FeS_2$) | 22 | High |
| Kaolinite-1A ($Al_2Si_2O_5(OH)_4$) | 4 | Low |
| Hexahydrite, syn ($Mg(SO_4)(H_2O)_6$) | 4 | Low |
| Bassanite, syn ($Ca(SO_4)(H_2O)_{0.5}$) | 3 | Low |
| Tin, syn (Sn) | 1 | Low |
| Goethite ($FeO(OH)$) | 1 | Low |
| Massicot, syn (PbO) | Trace | Low |

**Table C1.** *Cont.*

| Crystalline Mineral Phase | Concentration (wt.%) | ICDD Match Probability |
|---|---|---|
| **Reject Sample:** | | |
| Albite ($NaAlSi_3O_8$) | 38 | High |
| Quartz, syn ($SiO_2$) | 29 | High |
| Spodumene ($LiAlSi_2O_6$) | 10 | High |
| Muscovite-2M1 ($KAl_2((Si_3Al)O_{10}(OH)_2)$) | 6 | Medium |
| Microcline ($K(AlSi_3O_8)$) | 5 | Medim |
| Chlorapatite, syn ($Ca_5(PO_4)_3Cl$) | 5 | Low |
| Diaspore ($AlO(OH)$) | 3 | Low |
| Petalite, syn ($LiAl(Si_4O_{10})$) | 2 | Low |
| Hematite, syn ∣ Iron Oxide ($Fe_{1.766}O_3$) | 2 | Low |
| Clinochlore-1MIIb, Fe+2-bearing ($Mg_{2.8}Fe_{1.75}Al_{2.7}Si_{2.65}O_{10}(OH)_8$) | 1 | Low |
| Beryl, syn ($Be_3Al_2(Si_6O_{18})$) | Trace | Low |
| Eucryptite low ($LiAl(SiO_4)$) | Trace | Low |

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
