# Peer review of "Beneficiation of High-Density Tantalum Ore in the REFLUX™ Concentrating Classifier Analysed Using Batch Fractionation Assay and Density Data"

_minerals, doi:10.3390/min14020197_

Round 1
Reviewer 1 Report
Comments and Suggestions for Authors
Lots of great information and great efforts to prepare this paper. The only concern that I have is that it is a bit long.

Author Response
Please also see attached manuscript with changes shown in track changes mode.
Lots of great information and great efforts to prepare this paper. The only concern that I have is that it is a bit long.
RESPONSE: We realise the paper is quite lengthy, however none of the reviewers have suggested any parts of the paper that should be omitted or shortened, rather they have only suggested additions to the paper. In fact both Reviewers #1 and #3 have praised it for its comprehensiveness (“Lots of great information”, “describes in detail the experiment and provides a lot of data allowing any reader to take advantage of this work”). So we have concluded that the paper retain its current length. However, if the editor requests it, then we could move the raw data contained in Appendices B and C into a supplementary source to accompany the paper.
Reviewer #1 comments:
- Line 74-75: I suggest to change samples in “The batch RC™ 74 fractionation method does not yield three sets of samples to three sets of products. DONE: “samples” changed to “products”.
- Line 103-104: Samples of the feed, product, and reject from this experiment were fractionated into 11 flow fractions using a batch RC™ with LST solution. I I think it would be helpful to illustrate this arrangement with a schematic diagram. RESPONSE: A new Figure 2.1 has been added.
Figure 2.1: Schematic summary of the processing steps. The original continuous experiment furnished three samples – feed, product and reject. The +0.045 mm material from each of these samples was then batch fractioned into 11 flow fractions. Each of these fractions was then screened into 7 size intervals, thus resulting in a total of 3 × 11 × 7 = 231 portions.
- It's unclear why the option to reduce the number of size analyses to ensure sufficient mass in all fractions wasn't considered, opting instead for assays from adjacent fractions. RESPONSE: The experiments and sample analysis were all performed by a commercial laboratory as per an agreed set of instructions. So each flow fraction was screened into set size increments and then assays were performed on those portions with enough material in them. By the time we became aware that some portions were too small to perform assays on, it was too late to recombine them back into wider size intervals and limited funding also restricted out ability to perform further analysis.
- Instead of using "Flow11," you may consider using a more fitting term like "Underflow 10" to better represent the residue inside the unit at the conclusion of the testwork. RESPONSE: We did consider it, and indeed in some previous publications we refer to this material as the “Remains”. However, when discussing the analysis of such a large number of samples and presenting maths formula with summations over 11 terms, there is simplicity gained by being able to refer to “Flows 1-11”, rather than “Flows 1-10 and the Remains”. Hence we have decided to retain the present nomenclature.
- In Table B1, the numbers reported for Feed and Underflow Prod are identical. RESPONSE: This typo has been FIXED.
- Since the mass pull in Fraction 7 was too small for pycnometry and XRF analysis, you may consider combining that mass with Fraction 6. RESPONSE: As mentioned above, the screening and XRF analysis had already been done, so it was too late to recombine them. We chose to focus on the +0.045 mm material to avoid the noise associated with combing in fine material of uncertain composition.
- Conducting direct mineralogy analysis on the feed could enhance the paper by providing additional confirmation for some of the conclusions. RESPONSE: Given the extremely low yield to product, in most analysis where they can be compared directly the feed is very similar to the reject. XRD on the product and reject samples already gives a clear picture of what all the dominant mineral species in the system are. So we decided that there was little value in performing XRD on the feed sample.
- There seems to be a discrepancy in Fig. 3.4 where some species don't reach 100% at the lowest density. RESPONSE: The same is true also of Figures 3.2, 3.5 and 3.6, although less noticeable, and also the lower ends of these curves do not go all the way to 0 %. This is because the mass of each portion is assumed to be evenly split half above and half below the average density of that portion, leaving some mass at each end without a clear lower or upper density bound. So back at Line 329 where Figure 3.2 is first introduced, we have added the clarification:
“Hence reading from high to low density each curve starts slightly above 0 % (since half the mass of the densest portion is assumed to be denser than that portion’s average density, but the upper density limit is unknown) and rises to slightly below 100 % (since half the mass of the least dense portion is less dense than that portion’s average density and with an unknown lower density limit)."
- In Line 432, suggest replacing word silica in “albite (NaAlSi3O8), silica (SiO2) and microcline (KAlSi3O8)" with "quartz." DONE
- In Line 445, clarify that XRF does not specify iron as hematite; instead, it reports Fe as Fe2O3. To avoid confusion between assay values and mineral types, you may consider reporting elements as elemental assays rather than in oxide form. RESPONSE: The reference to Fe2O3 in this paragraph has been deleted. Because the XRF assay raw data were reported to us as wt.% of these oxides, we thought it best to retain reference to the oxides when quoting assays. Hopefully Table 3.2 makes it clear that the XRF label does not imply the actual presence of those oxides.
- You should address the negative numbers in Table B.5 for Yield and Recovery, since these numbers can not be negative. RESPONSE: The reviewer is correct that yields and recoveries cannot in reality be negative. However, the two-product formula can give negative values when the feed and reject compositions are similar and there is noise in the data. We have actually recently posted a pre-print on a comprehensive study in which we analyse how error in the raw assay data is amplified by the non-linear nature of the two-product formula, and how near the singularity the errors can grow dramatically: https://doi.org/10.26434/chemrxiv-2023-n9jqb
So across a set of 20 assays, many of which are for low density species that report to the tailings at almost the same concentration as they appear in the feed, it is unsurprising to see a few spurious negative results. This was already alluded to in Section 2.2 where we state: “Hence in situations like this study where there is a very low product yield resulting in a tailings composition that is often very similar to the feed composition, spurious zero or negative values can often occur.” However, we have added an additional paragraph discussing this to Section 3 where the negative yield and recovery values first appear in Table 3.1, including reference to this preprint and added the estimated error amplifications of the yield and recovery to Table 3.1.
- If feasible, conducting a comparative analysis of Reflux and alternative gravity methods such as Spiral, Wilfley Table, or their combination on a similar feed would offer valuable insights. Alternatively, providing a general comparison between these methods and Reflux, considering factors like footprint, capital investment, ease of operation and robustness, and separation performance, would enhance the understanding of the potential benefits of the Reflux unit for gravity separation.
RESPONSE: We agree that such comparative studies would be valuable, but such an extensive body of work is beyond the scope of the funding we have available. Comparing different technologies is also fraught with difficulty, as one has to demonstrate that each technology is being operated at its optimum performance point (it is easy to make something work poorly). We have, however, added a comment in the experimental section that the solids throughput in the continuous experiment was 17 tonnes per hour per square metre which may help readers to better evaluate the capacity of this technology compared to others.

Reviewer 2 Report
Comments and Suggestions for Authors
Thanks for a good and interesting article. I have only a few comments:
- The XRF abbreviation should be written first time in full
- XRF is elemental analysis method, XRF does not detect any minerals (line 148). Fe2O3 is just a way to express total iron content in a sample, and does not mean occurrence of hematite, so "inferred minerals" is wrong expression.
- Pyrite was identified by XRD, so the sentence on lines 444-446 is not relevant.
Author Response
Please also see changes shown in attached manuscript in track changes mode.
Response to Reviewer #2 Comments:
Thanks for a good and interesting article. I have only a few comments:
- The XRF abbreviation should be written first time in full
DONE in Section 2, and also the XRD abbreviation.
- XRF is elemental analysis method, XRF does not detect any minerals (line 148). Fe2O3 is just a way to express total iron content in a sample, and does not mean occurrence of hematite, so "inferred minerals" is wrong expression. RESPONSE: This has been changed to “reported as oxides” and explained more clearly.
- Pyrite was identified by XRD, so the sentence on lines 444-446 is not relevant. RESPONSE: The reference to hematite has been removed.

Reviewer 3 Report
Comments and Suggestions for Authors
See attached file

See attached file
Author Response
Thank you for the thoroughness you took in reviewing our manuscript, even picking up some typos in our reference list. Besides the comments below, please also see the revisions in track changes in attached manuscript.
Nevertheless, the reviewer would have liked to see more information on the sample masses used for characterisation, and on sampling and analysis errors that could have been used to reconcile data by material balance. Concerning this reconciliation, the use of different steps can generate some bias, but a one-pass data reconciliation is very difficult, if not impossible, to obtain with Excel. RESPONSE: We have included mention that the sample masses used for characterization were 35-40 g.
The following section of that review indicates some major revisions or suggestions to enhance the rigour of the talk. The last section lists detected mistakes or unnecessary notations which can easily be fixed.
Detailed major revisions
Lines 170-174, 264: I suggest you to use the split curve (??,? − ??,? as a function of ??,? − ??,?) for size intervals, some density fractions and chemical elements, and crossed compositions such as portions and size-by-size chemistry when available, to have a visual representation of this consistency.
RESPONSE: We have not added new graphs but have added extra discussion about the data consistency in Section 3.0:
"The average magnitude of the mass-balancing relative adjustments to the raw assay data was 1.8 % for the head assays, 2.1 % for the +0.045 mm assays, and 3.4 % for the -0.045 mm assays. This high level of consistency suggests that the system was indeed operating close to steady state, and that the data set is reliable."
We have worked extensively across similar experiments over the years and observed the consequences of inconsistencies. For example, certain size fractions can take longer to reach steady state due to the issue of hold-up in the separator. Premature sampling then leads to considerable mass balance errors. We have also seen the consequence of poor-quality sieving, which propagates errors across the whole data set. The low errors here provide confidence in the results.
In addition, global data reconciliation by material balance using all data in one calculation allows to have a unique value for the yield. Table 3.1: If the balanced yields for +45 µm and –45 µm are 4.1 % and 6.0 % respectively, the global balanced yield cannot be 4.0 %. It is why you must consider global data reconciliation to have more realistic values. Table A.1 must be adjusted consequently.
RESPONSE: The focus of this study was on the fractionation of the +0.045 mm material, representing 93 % of the feed. If we had included the -0.045 mm material, then a relatively small amount of material covering a relative wide size range would have resulted in a fractionation analysis involving vastly more sub-samples, and involving many more samples containing insufficient material for XRF analysis, and much more cost. With the focus entirely on the +0.045 mm material there was no point in including the -0.045 mm material in the yield calculation. If one did include that material, then you would be introducing additional uncertainty from dispersed ultrafine material that is not even a part of the analysis or fractionation.
In hindsight it might have been better to not refer to the yield of the -0.045 mm material, but we looked to present a complete description of the work. The reality is that each of the three yields carries its own uncertainty. As such the three yields are consistent in the context of their likely uncertainty, but we appreciate its best not to leave this technical inconsistency “hanging”. The final mass balance needs to remain based on the +0.045 mm material.
We have added a brief comment to acknowledge the fact that in addition to the +0.045 mm yield of 4.1 % (for 93% of the feed), an additional yield calculation was performed on the -0.045 mm material (7% of the feed) resulting in a yield of 6.0 %. We then state that the estimated overall yield based on these two yield calculations, for the whole feed, is therefore 4.2 %, which is in close agreement with the value of 4.0 % found by mass balancing the head assay data.
In section 3.2, when you define the “volume-weighted average density”, indicate clearly that this is a pseudo density which is not the density of the species used for this calculation, neither the density of the mineral bearing that species (even though it can be close in the case of only one mineral type is bearing the species). Mention it is just an indicator for the understanding of the separation performances. It is why I add some suggestions in the next section of that review to indicate this calculated density is associated to a species, not the density of the species.
RESPONSE: We have added the statement that: “This average is not the density of any specific mineral, but rather is a proxy density that incorporates the densities of all minerals associated with that species and it is used here purely as an indicator for understanding the separation performance.”
Why do you name this parameter “volume-weighted average density”? It is well known that the average density is the weighted harmonic mean where the weights are the masses. The evidence being you are using “Species Mass-Weighted Average Density” for Figure 3.11, also in line 472. Maybe, just using “average density associated to species” is sufficient.
RESPONSE: We have deleted reference to “volume-weighted”, corrected Line 472 and the axis title of Figure 3.11, and as recommended adjusted the text to refer to it as the average density “associated” with a particular species.
In different parts of the paper, you indicate “the lack of resolution at the high densities”, or something similar, for the feed (see lines 215, 410 and 509). That seems understandable as the main part of the feed (96 %) reports to the low-density tailings, but it is in contradiction with table B.1. it appears the values in the Density column for the Feed section are wrong (just a copy of the ones of the Underflow Product section). RESPONSE: Typo in Table B.1 has been corrected (thank you for picking it up).
Detailed minor revisions
Concerning the grades or proportions in the solid material, it is not necessary to indicate “wt.%” each time as it is well known it is relative to dry weight. Just use “%”. Similarly for yield.
RESPONSES: Percentages can be based on a moles, volumes or masses, so for clarity we have left all occurrences of wt.% in the abstract, conclusions and in the units shown in tables and figures, but elsewhere we have simplified it to just %.
You can use past tense for section 2 as it describes the experiments which are no longer current at the time of writing and reading the paper, but for section 3, it is preferable to use the present tense as the results are still valid at the time of reading. Except when you are referring to the experiment conditions.
RESPONSE: We have changed some statements in Section 3 to present tense.
If A is function of B, there is always a discussion on the way to describe the curve: “A versus B” or “B versus A”? As there is no rule, maybe it is preferable to use something like “A is plotted as a function of B” to avoid any confusion. DONE
Line 146: Indicate the table B1 is in Appendix B. DONE
Lines 147-148: In place of "inferred minerals" use "oxides" as it is not really minerals which are present in the sample, but a way to display contents using main oxides. It concerns also lines 384, 391. DONE
Line 250-251 and Figure 3.10: The currently used name for Ep is “Ecart Probable” (coming from French) or “Probable Error”, but not “Ecart Probable error”. DONE
Line 269: Add “for” before “Sn”, “Ta” and “Nb”. DONE
Lines 290-291: To be consistent with lines 254-259, use “…sum of square relative deviations between experimental and balanced assays over all species.” DONE: changed to “… sum of the squares of the relative differences between the experimental and balanced assays over all species.”
Line 312: “Ta” in place of “Tn”. DONE
Lines 397-398: Proposed to reformulate the sentence: “As most of species only reside in a very narrow range of density, then this potentially means that we can treat that species as if they are density tracer particles.” DONE Has been re-written as: “Species that exclusively reside in only a very narrow range of density can potentially be treated as if is they are a density tracer particle.”
Lines 402-403: Remove “the” before “Li”; “Fe” and “S”.
RESPONSE: The absolute amount of Li, Fe, S, etc. is not large in and of itself, but it is a large fraction of the total amount of that species in the sample. To clarify this, the sentence has been changed to: “There is also a band of material with densities in the range 3200 to 3400 kg/m3 that contains a large proportion of the total Li and P.”
Line 415: Second “Li” must be replaced by “Na”. DONE
Line 437: Replace “(Kelly & Spottiswood, 1995; www.webmineral.com)” by “[22,23]”. DONE
Line 439: Start sentence as “Both are very dense minerals…”. DONE
Lines 445-446: Remove “rather than the default form of hematite (Fe2O3) that was assumed in the XRF output” for the reason explained above. DONE
Line 450: Correct “clinochlore”. DONE
Line 452: As the volume-weighted average density is not the density of the species, but the density of mineral bearing the species, it is preferable to write “…the volume-weighted average density ???,? associated to species ? in a stream is estimated across the entire sample:”. DONE
Line 456: Replace “for the 10” by “associated to the 10”. DONE
Line 463: Replace “due to it including” by “because it includes”. DONE
Line 464: “Average density associated to the 10…”. DONE
Table 3.2: It seems it is “Figure 3.7” instead of “Figure 3.6” in the head of third column. DONE
Lines 471 and 492: It seems to be “Table B.5” in place of A.5. DONE
Line 472: Suggesting “the mass-weighted average density, associated to the species, as calculated by Equation 3.1”. DONE
Line 485: “…two species with associated average densities…” DONE
Line 493: Replace “was found” by “are associated,”. DONE
Line 495: “The data in Figure 3.8 suggest a clearer separation at low and high densities.” CHANGED TO: “The data in Figure 3.8 appear to be approaching 100 % and 0 % partition respectively at high and low densities with no signs of any tails (i.e. the curve has “good closure”).”
Line 498: Replace “both” by “the”. DELETED the word “both”.
Lines 501-502: “…single density, knowing this is not true.” CHANGED to “… but we know this is not true.”
Lines 506 and 630: The term closure is not well appropriated. RESPONSE: We have clarified what we mean by “good closure” through the revision made to the old Line 495 described above.
Lines 529-530: Add a coma between “that” and “when” and replace second “that” by a coma. DONE
Lines 550-551: The “average relative deviation” is not pertinent. It is preferable to indicate the number of species having such relative deviation in the range -10 % - +10 % compared to the total number of species. DONE
Lines 553-554: “Using this method, reconstructed feed, product and reject streams…” DONE
Figure 3.9 and Line 558: Prefer “Relative deviation” than “Relative error”. CHANGED instead to “Relative Difference” so as to avoid any confusion with the term standard deviation.
Figure 3.10: “Ecart Probable, Ep (kg/m3”. DONE
Line 620: Aff “for” before “Sn”, “Ta” and “Nb”. DONE
Table B.1: Wrong densities for Feed section DONE
Lines 677, 690 and 704: Replace “is due to there being” by “is because there is”. DONE
Line 718: “B.2-B.4” instead of “A.2-A.4”. DONE
Figures C.1 and C.2: the legends are too small. RESPONSE: The legend size has been increased.
Line 731: Remove the s from interpretations. DONE
Line 732: Remove the s from matches. DONE – assuming meant to remove “ed” from “matched”.
Table C.1: The “Clinochlore” line is slightly truncated. In the “Eucryptite line, there is an upper script 2 as for a reference, but without reference. DONE
Line 740: Remove “Classifier”. DONE
Line 741: Replace 2020 by 2021. DONE
Line 743: Replace “Lambers” by “Lambert”. DONE
Line 747: “lift” in place of “life” DONE
Line 769: Replace 2019 by 2020. DONE
